# A Near-Optimal Primal-Dual Method for Off-Policy Learning in CMDP

**Fan Chen**
School of Mathematics
Peking University
chern@pku.edu.cn

**Junyu Zhang**[*]
Department of Industrial Systems Engineering and Management
National University of Singapore
junyuz@nus.edu.sg

**Zaiwen Wen**
Beijing International Center for Mathematical Research, Center For Machine Learning Research
Peking University
wenzw@pku.edu.cn

## Abstract

As an important framework for safe Reinforcement Learning, the Constrained Markov Decision Process (CMDP) has been extensively studied in the recent literature. However, despite the rich results under various on-policy learning settings, there still lacks some essential understanding of the offline CMDP problems, in terms of both the algorithm design and the information theoretic sample complexity lower bound. In this paper, we focus on solving the CMDP problems where only offline data are available. By adopting the concept of the single-policy concentrability coefficient $C^*$, we establish an $\Omega\left(\frac{\min\{|\mathcal{S}||\mathcal{A}|,|\mathcal{S}|+I\}C^*}{(1-\gamma)^3\epsilon^2}\right)$ sample complexity lower bound for the offline CMDP problem, where $I$ stands for the number of constraints. By introducing a simple but novel deviation control mechanism, we propose a near-optimal primal-dual learning algorithm called DPDL. This algorithm provably guarantees zero constraint violation and its sample complexity matches the above lower bound except for an $\tilde{\mathcal{O}}((1-\gamma)^{-1})$ factor. Comprehensive discussion on how to deal with the unknown constant $C^*$ and the potential asynchronous structure on the offline dataset are also included.

## 1 Introduction

Reinforcement Learning (RL) is an important tool for modeling the real world tasks that involve sequential decision making. Such RL problems are often mathematically described as a Markov Decision Process (MDP) that maximizes a cumulative sum of rewards. The safe reinforcement learning, on the other hand, not only cares the reward maximization, but also attempts to ensure a reasonable system performance with respect to certain safety constraints. Such safety constrained RL problems are often formulated as the Constrained Markov Decision Process (CMDP) $\mathcal{M} = (\mathcal{S}, \mathcal{A}, \mathbb{P}, r, u, \gamma, \rho_0)$, where $\mathcal{S}$ is a finite state space, $\mathcal{A}$ is a finite action space, $\gamma \in (0, 1)$ is the discount factor, $\mathbb{P}(s' \mid s, a)$ stands for the transition probability from $s$ to $s'$ under the action $a$ for $\forall (s, a, s') \in \mathcal{S} \times \mathcal{A} \times \mathcal{S}$, and $r : \mathcal{S} \times \mathcal{A} \to [-1, 1]$ is the reward function, $(u_i : \mathcal{S} \times \mathcal{A} \to [-1, 1])_{i \in [I]}$ is a set of $I$ utility functions, $\rho_0$ is the initial state distribution over $\mathcal{S}$. The goal of CMDP is to find

---

[*]Corresponding author.

36th Conference on Neural Information Processing Systems (NeurIPS 2022).

an optimal policy $\pi$ to maximize the cumulative reward while satisfying a group of constraints:

$$\max_\pi \quad J(\pi) := \mathbb{E}\left[\sum_{t=0}^{+\infty} \gamma^t \cdot r\left(s_t, a_t\right) \,\Big|\, s_0 \sim \rho_0, \pi\right] \tag{1}$$

$$\text{s.t.} \quad J_i^u(\pi) := \mathbb{E}\left[\sum_{t=0}^{+\infty} \gamma^t \cdot u_i\left(s_t, a_t\right)\right] \geq 0, \text{ for } i \in [I] = \{1, 2, ..., I\}.$$

For the CMDP problem, there has been plenty of on-policy algorithms, see [7, 8, 20, etc.]. However, in real world applications such as training physical robots, where safety is an important measure of performance, the real time on-policy interaction with the environment may suffer from the potential damages to the robots. Besides, in many non-simulating environments, the on-policy data collection may also be time-consuming. Therefore, it is crucial to design an off-policy algorithm to solve the CDMP problems, where plenty of historical data are already accumulated while real time interactions are limited. To our best knowledge, offline CMDP algorithms are rare [12, 27, 29], and the sample complexity guarantees are limited. In particular, a strong uniform concentrability assumption is required in [12], and the model-based method [27] mainly considers the case an empirical model is known. Thus it is still not clear how to efficiently solve offline CMDPs with model-free approaches, and there lacks essential understanding of the information theoretic lower bound on the sample complexity of the offline CMDP.

In this paper, we propose a Deviation-controlled Primal-Dual Learning (DPDL) method to solve problem (1). We adopt the primal-dual strategy developed in [4, 16, 26, 35, etc.] as the main algorithmic framework while several non-trivial contributions have been made beyond the existing results. Unlike the aforementioned literatures that exclusively rely on the accessibility of a generative model, DPDL utilizes the offline data, where the distribution shift difficulties of the offline data is tackled by a novel and effective adaptive deviation control mechanism. If the considered CMDP instance has a finite (but potentially unknown) *concentrability coefficient*, DPDL provably finds a policy with $\mathcal{O}(\epsilon)$-optimal reward and zero constraint violation. An information theoretical lower bound on the sample complexity of offline CMDP is also derived in this paper, which indicates that our deviation control mechanism achieves a minimax optimal complexity dependence on $I, |\mathcal{S}|, |\mathcal{A}|, C^*$.

**Main Contribution.** We summarize the contributions in details as follows.

- We propose the DPDL algorithm to solve the CMDP problem (1). Suppose the CMDP instance satisfies the Slater's condition and certain prior knowledge on the concentrability coefficient $C^*$ is given, DPDL provably finds an $\epsilon$-optimal policy with zero constraint violation using $\tilde{\mathcal{O}}\left(\frac{\min\{|\mathcal{S}||\mathcal{A}|, |\mathcal{S}|+I\}C^*}{(1-\gamma)^4\epsilon^2}\right)$ offline samples.

- We establish an information theoretic sample complexity lower bound of $\Omega\left(\frac{\min\{|\mathcal{S}||\mathcal{A}|, |\mathcal{S}|+I\}C^*}{(1-\gamma)^3\epsilon^2}\right)$ for the offline CMDPs, indicating that DPDL is near optimal up to an $\tilde{\mathcal{O}}((1-\gamma)^{-1})$ factor. The necessity of the Slater's condition for achieving zero constraint violation is also established. The sharp dependence on the number of constraints is mainly captured by our careful construction of the correlated actions.

- In order to handle the practical situation where $C^*$ is unknown, an adaptive version of DPDL is designed with the same sample complexity as DPDL.

- Our analysis of DPDL also extends to the asynchronous case, where the offline dataset consists of a sample trajectory generated by certain behavior policy. In this situation, the sample complexity of DPDL is shown to be $\tilde{\mathcal{O}}\left(\frac{t_{\text{mix}}^2 \min\{|\mathcal{S}||\mathcal{A}|, |\mathcal{S}|+I\}C^*}{(1-\gamma)^4\epsilon^2}\right)$. Our handling of the correlated gradient estimators with large variance can also be beneficial to other algorithms under the asynchronous setting.

**Related Work.** Recently, considerable efforts have been devoted to the online learning of CMDP. Under the episodic and tabular setting, several works [7, 8, 20] have achieved the $\tilde{\mathcal{O}}\left(\sqrt{|\mathcal{S}|^2|\mathcal{A}|T}\right)$ regret and cumulative constraint violation, with different dependence on the episode length $H$ omitted. Under proper assumptions, zero or bounded cumulative constraint violation can be achieved [1, 17]. In terms of the number of constraints $I$, MOMA proposed in [34] achieves an $\tilde{\mathcal{O}}\left(\sqrt{\min\{|\mathcal{S}|, I\}I|\mathcal{S}||\mathcal{A}|/T}\right)$ convergence on both average reward gap and constraint violation.

Nevertheless, all the above results adopt the model-based approaches. Except for [34], they either consider the cases where $I = 1$ or completely ignore the influence of $I$ in the sample complexity. Therefore, both deriving an efficient model-free method and obtaining the optimal dependence on $I$ remain open.

Another approach closely related to our paper is the primal-dual method in RL, see [4, 11, 25, 26, 35, etc.]. Given the access to a generative model, the model-free primal-dual method developed in [4] achieves an $\tilde{\mathcal{O}}\big(\frac{I|\mathcal{S}||\mathcal{A}|}{(1-\gamma)^4\epsilon^2}\big)$ sample complexity to find an $\epsilon$-optimal safe policy. The deviation control mechanism we develop enables the primal-dual approach to extend beyond the generative model.

Finally, we mention a few related works in the offline RL and safe RL. Previous offline RL algorithms with sample efficiency guarantees typically assume the *uniform concentrability* [12, 18, etc.] or lower bounded *minimum visitation* $\mu_{\min}$ [32, 33, etc.]. Recently, under the less restrictive assumption of the *single-policy concentrability coefficient* $C^*$, a minimax optimal sample complexity lower bound of $\Omega\big(\frac{|\mathcal{S}|C^*}{(1-\gamma)^3\epsilon^2}\big)$ for discounted offline MDPs is derived in [21]. A similar $\Omega\big(\frac{H^3|\mathcal{S}|C^*}{\epsilon^2}\big)$ lower bound is also derived for the episodic setting in [28]. Under both settings, offline algorithms with $\tilde{\mathcal{O}}(|\mathcal{S}|C^*\epsilon^{-2})$ sample complexity (with different $(1-\gamma)^{-1}$ or $H$ factors omitted) have been discovered with either model-based [15, 21, 28, 31] or model-free approaches [22, 30]. In terms of the offline CMDP problem, the only existing results are [12, 27, 29], where [29] only provides asymptotic convergence, [12] relies on a much stronger uniform concentrability assumption, and [27] is a model based method that potentially suffers an $\mathcal{O}((C^*)^2)$ dependence. Compared to these works, our method is model-free and has an optimal $\mathcal{O}(C^*)$ dependence on the concentrability coefficient.

## 2 Problem setup

### 2.1 LP formulation of CMDP problem

For any policy $\pi$, the (unnormalized) state-action occupancy measure is defined as

$$\nu^\pi(s,a) := \sum_{t=0}^{+\infty} \gamma^t \cdot \mathbb{P}\left(s_t = s, a_t = a \mid s_0 \sim \rho_0, \pi\right), \quad \text{for} \ \ \forall (s,a) \in \mathcal{S} \times \mathcal{A}. \tag{2}$$

Given any occupancy measure $\nu^\pi$, the policy $\pi$ that generates $\nu^\pi$ can be recovered as

$$\pi(a|s) = \frac{\nu^\pi(s,a)}{\sum_{a'} \nu^\pi(s,a')}, \quad \forall (s,a) \in \mathcal{S} \times \mathcal{A}. \tag{3}$$

According to [2], it is well known that the set of all state-action occupancy measures form a polyhedron $\big\{\nu \in \mathbb{R}_{\geq 0}^{|\mathcal{S}| \times |\mathcal{A}|} : \sum_{a \in \mathcal{A}}(\mathbf{I} - \gamma\mathbb{P}_a)\nu_a = \rho_0\big\}$, where $\nu_a := (\nu(s,a))_{s \in \mathcal{S}}$ is an $|\mathcal{S}|$-dimensional column vector, $\mathbf{I}$ is the $|\mathcal{S}| \times |\mathcal{S}|$ identity matrix, and $\mathbb{P}_a := (\mathbb{P}(s'|s,a))_{s',s}$ is an $|\mathcal{S}| \times |\mathcal{S}|$ transition matrix, see also [26]. Therefore, combined with the fact that $J(\pi) = \langle \nu^\pi, r\rangle$ and $J_i^u(\pi) = \langle \nu^\pi, u_i\rangle$, the CMDP problem (1) can be reformulated as an LP problem with $|\mathcal{S}|+I$ constraints:

$$\max_{\nu \in \mathbb{R}_{\geq 0}^{|\mathcal{S}| \times |\mathcal{A}|}} \quad \langle \nu, r\rangle \quad \text{s.t.} \quad \sum_{a \in \mathcal{A}}(\mathbf{I} - \gamma\mathbb{P}_a)\nu_a = \rho_0, \quad \langle \nu, u_i\rangle \geq 0, \forall i \in [I]. \tag{4}$$

Due to the fundamental theorem of LP, see e.g. [5], problem (4) has an optimal basic feasible solution with at most $|\mathcal{S}| + I$ positive entries, which indicates the following proposition.

**Proposition 2.1.** *For the CMDP problem* (1) *with $I$ constraints, there is an optimal policy $\pi^*$ such that $|\mathrm{supp}(\nu^{\pi^*})| \leq \mathcal{N} := \min\{|\mathcal{S}|+I, |\mathcal{S}||\mathcal{A}|\}$, where $\mathrm{supp}(\cdot)$ denotes the support of a vector.*

This result captures the potential sparse structure of the optimal policy when $I$ is not as large as $|\mathcal{S}||\mathcal{A}|$, and is the key to deriving a tight complexity dependence on the number of constraints $I$.

### 2.2 Off-policy learning from demonstration

In this work, we consider the offline CMDP problems where the agent cannot interact with the environment. Instead, the optimization is conducted using a fixed offline dataset. To standardize the discussion, we make the following assumption on the offline dataset, see e.g. [21].

**Assumption 2.2** (Independent batch dataset)**.** *The batch dataset $\mathcal{D}$ consists of independent tuples $(s, a, s', r, \mathbf{u})$, such that $(s, a) \sim \mu$, $\mathbb{E}\left[r \,|\, s, a\right] = r(s, a)$, $\mathbb{E}\left[\mathbf{u}_i \,|\, s, a\right] = u_i(s, a)$, and $s' \sim \mathbb{P}(\cdot | s, a)$, where $\mu$ is called the reference distribution.*

To characterize the distribution shift of an arbitrary occupancy measure $\nu^\pi$ from the reference distribution $\mu$, we introduce the following notion of the deviation: $D^\pi := \max_{s,a} \frac{(1-\gamma)\nu^\pi(s,a)}{\mu(s,a)}$, where the $(1-\gamma)$-factor normalizes $\nu^\pi$ to be a distribution. In offline RL, it is natural to assume that the deviation $D^{\pi^*}$ of the optimal policy is finite. That is, the reference distribution $\mu$ fully covers $\mathrm{supp}(\pi^*)$. Otherwise, no optimality can be guaranteed. Combining the sparse nature of the optimal solution of (1), we introduce the following finite concentrability assumption for our problem.

**Assumption 2.3.** *For $\forall \psi \geq 1$, denote the $\psi$-deviated policy class as $\Pi(\psi) := \left\{\pi : \nu^\pi \in D(\psi)\right\}$ where*

$$D(\psi) := \left\{\nu \in \mathbb{R}_{\geq 0}^{|\mathcal{S}||\mathcal{A}|} : \max_{s,a} \frac{(1-\gamma)\nu(s,a)}{\mu(s,a)} \leq \psi, \sum_{s,a} \frac{(1-\gamma)\nu(s,a)}{\mu(s,a)} \leq \mathcal{N}\psi\right\}. \qquad (5)$$

*We assume there exists a finite $\psi$ such that some optimal policy $\pi^*$ is contained in $\Pi(\psi)$. Let $C^*$ be the minimum of such $\psi$. We call this constant $C^*$ the (single-policy) concentrability coefficient.*

The above assumption includes a sparsity induced constraint as a result of Proposition 2.1, its counterpart in the definition of single-policy concentrability of offline MDP [21] is the deterministic optimal policy. The explicit dependence on $\mathcal{N}$ in $D(\psi)$ facilitates the derivation of the information theoretic lower bound as well as a near-optimal algorithm.

A second remark is that if we know any upper bound $\psi$ of the coefficient $C^*$, then it will be sufficient to only consider the policies in $\Pi(\psi)$. When $C^*$ is unknown, $\psi$ control the risk of distribution shift. Consequently, in this paper, we propose to solve the LP formulation (4) with a tighter feasible region introduced by $D(\psi)$. This will allow us to properly control the variance of the off-policy sampling when some of $\mu(s, a)$ is extremely small or even zero. We call this strategy *deviation control*.

## 2.3 Conservatism toward constraints

We say policy $\pi$ is safe if it satisfies all constraints in (1), and we say $\pi$ is $\epsilon$-safe if $J_i^u(\pi) \geq -\epsilon$, for $\forall i \in [I]$. Most of the existing online CMDP algorithms guarantee $\mathcal{O}\left(1/\sqrt{T}\right)$ average safeness. To ensure the true safeness (zero constraint violation) in this work, we assume the Slater's condition to hold throughout this paper. In fact, in Section 5, we will show that the Slater's condition is the necessary condition for any offline CMDP algorithm to obtain zero constraint violation.

**Assumption 2.4.** *There exists $\varphi > 0$ and a policy $\pi$ such that $J_i^u(\pi) \geq \frac{\varphi}{1-\gamma}$, $\forall i \in [I]$.*

A prior knowledge of such a constant $\varphi$ is assumed throughout our discussion, and we also assume the Slater's condition holds for $\Pi' := \Pi(C^*)$. Given Assumption 2.4, we leverage the idea of conservative constraints proposed in [4]. Namely, instead of $J_i^u(\pi) \geq 0$, we consider the conservative constraints $J_i^u(\pi) \geq \kappa$ when solving the CMDP problem, where $\kappa > 0$ is a properly chosen parameter that controls the level of conservatism in the constraints. In order to keep the form of the constraints in problem (1), we adopt a shifted utility function $u_i^\kappa$ defined by $u_i^\kappa(s, a) := u^i(s, a) - (1-\gamma)\kappa$ for $\forall (s, a) \in \mathcal{S} \times \mathcal{A}$, $\forall i \in [I]$. Therefore, $J_i^u(\pi) \geq \kappa$ is then equivalent to $J_i^{u^\kappa}(\pi) \geq 0$. It can be shown that a properly selected $\kappa$ will facilitate a high probability of preserving zero constraint violation, while only introducing an extra $\mathcal{O}\left(\frac{\kappa}{\varphi}\right)$ sub-optimality gap in the reward.

## 3 The Deviation-controlled Primal Dual Learning (DPDL) algorithm

To solve CMDP with offline samples, we transform its LP formulation (4) to a saddle point form

$$\max_{\nu \in D(\psi)} \min_{\lambda \geq 0, V} \mathcal{L}(V, \lambda, \nu) := \langle r, \nu \rangle + \left\langle V, \rho_0 - \sum_a (\mathbf{I} - \gamma \mathbb{P}_a)\nu_a \right\rangle + \langle \lambda, U_\kappa \nu \rangle, \qquad (6)$$

where $D(\psi)$ is defined by (5), $V \in \mathbb{R}^{|\mathcal{S}|}$, $\lambda \in \mathbb{R}^I$ are Lagrangian multipliers, and the matrix $U_\kappa$ is defined as $U_\kappa := [u_1^\kappa, \cdots, u_I^\kappa]^\top \in \mathbb{R}^{I \times |\mathcal{S}||\mathcal{A}|}$ with $u_i^\kappa$ being the shifted utility defined in Section 2.3.

Given the reference distribution $\mu$, the objective function can be rewritten as an expectation:

$$\mathcal{L}(V,\lambda,\nu) = \mathbb{E}_{s_0 \sim \rho_0} \left[V(s_0)\right] + \mathbb{E}_{\substack{(s,a) \sim \mu \\ s' \sim \mathbb{P}(\cdot|s,a)}} \left[\frac{\nu(s,a)}{\mu(s,a)}\left(r(s,a) - (V(s) - \gamma V(s')) + \sum_i \lambda_i u_i^\kappa(s,a)\right)\right].$$

If the reference distribution $\mu$ is known, we can directly sample a stochastic gradient of $\mathcal{L}$. However, when the reference distribution $\mu$ is unknown in practice, then the importance sampling weight $\frac{\nu(s,a)}{\mu(s,a)}$ is also unknown. To tackle this issue, let $\hat{\mu}$ be a proper estimation of the reference distribution $\mu$, we introduce the weights $w(s,a) = \frac{\mu(s,a)}{\hat{\mu}(s,a)}$, and the diagonal matrix $W = \mathrm{diag}\,(w(s,a))$. Then we apply a change of variables $x = W^{-1}\nu$, in other words, we set $\frac{x(s,a)}{\hat{\mu}(s,a)} = \frac{\nu(s,a)}{\mu(s,a)}$ for $\forall s, a$ to enable sampling. From now on, we will focus on the following reweighted problem

$$\min_{\lambda \in \Lambda, V \in \mathcal{V}} \max_{x \in \mathcal{X}} \mathcal{L}_w(V,\lambda,x) := \mathcal{L}(V,\lambda,Wx), \tag{7}$$

where the feasible regions are defined as

$$\mathcal{X} := \left\{x \in \mathbb{R}_{\geq 0}^{|\mathcal{S}||\mathcal{A}|} : \max_{s,a} \frac{x(s,a)}{\hat{\mu}(s,a)} \leq \frac{\psi}{1-\gamma}, \sum_{s,a} \frac{x(s,a)}{\hat{\mu}(s,a)} \leq \frac{\mathcal{N}\psi}{1-\gamma}, \sum_{s,a} x(s,a) \leq \frac{4}{1-\gamma}\right\},$$

$$\mathcal{V} := \left\{V \in \mathbb{R}^{|\mathcal{S}|} : \|V\|_\infty \leq \frac{8}{1-\gamma}(1 + \frac{2}{\varphi})\right\} \quad \text{and} \quad \Lambda =: \left\{\lambda \in \mathbb{R}_{\geq 0}^I : \|\lambda\|_1 \leq \frac{8}{\varphi}\right\}. \tag{8}$$

The sets $\mathcal{X}$, $\mathcal{V}$ and $\Lambda$ are chosen to be large enough so that they contain the optimal solution of the problem (6), see detailed discussion in Appendix E. Given a sample $\zeta = (s_0, s, a, s', r, \mathbf{u}) \sim \rho_0 \times \mathcal{D}$, and a point $Z := (V, \lambda, x)$, we construct the unbiased gradient estimators for $\mathcal{L}_w(\cdot)$ as

$$\widehat{g}_V(Z;\zeta) := \mathbb{I}_{s_0} + \frac{x(s,a)}{\hat{\mu}(s,a)}\left(\gamma \mathbb{I}_{s'} - \mathbb{I}_s\right),$$

$$\widehat{g}_\lambda(Z;\zeta) := \frac{x(s,a)}{\hat{\mu}(s,a)}\mathbf{u}^\kappa, \tag{9}$$

$$\widehat{g}_x(Z;\zeta) := \frac{r + \gamma V(s) - V(s') + \langle \mathbf{u}^\kappa, \lambda\rangle}{\hat{\mu}(s,a)}\mathbb{I}_{s,a},$$

where $\mathbb{I}_s$ is the $|\mathcal{S}|$-dimensional unit vector with the $s$-th element being one, $\mathbb{I}_{s,a}$ is the $|\mathcal{S}||\mathcal{A}|$-dimensional unit vector with the $(s,a)$-th element being one, and $\mathbf{u}^\kappa = \mathbf{u} - \kappa(1-\gamma)\mathbf{1} \in \mathbb{R}^I$ is the shifted utility vector. Based on these estimators, we propose a stochastic mirror descent ascent approach to solve problem (7), as stated in Algorithm 1.

The algorithm starts from a feasible solution $Z^1$, which, for example, can be easily chosen as $V^1 = \mathbf{0}$, $\lambda^1 = \frac{\mathbf{1}}{\varphi I}$, $x^1 = \frac{\mathcal{N}}{|\mathcal{S}||\mathcal{A}|}\frac{\hat{\mu}}{1-\gamma}$. In each iteration, an offline sample $\zeta^t$ is used to construct the unbiased gradient estimators $g_V^t$, $g_\lambda^t$ and $g_x^t$. A stochastic mirror descent ascent step (11) is then used to update the solution $Z^t$, where $\mathrm{Proj}_\mathcal{V}(\cdot)$ denotes the Euclidean projection to the set $\mathcal{V}$, and $\mathrm{KL}(Y\|Y') := \sum_i Y_i \log \frac{Y_i}{Y_i'} - \sum_i Y_i + \sum_i Y_i'$ denotes the generalized KL divergence. Simple closed form solutions are available to the $V^{t+1}$ and $\lambda^{t+1}$ updates. By taking the advantage of the special structure of $g_x^t$ and the fact that $x^t \in \mathcal{X}$ is feasible, the $x^{t+1}$ subproblem can be reduced to the root finding of a 1-dimensional monotone function, which can be solved efficiently, see details in Appendix A.

Finally, it is worth noting that $\overline{x}$ is the approximate optimal solution to the reweighted problem. And $W\overline{x}$ will be the approximate solution to the original problem (6) before the change of variable. Therefore, ideally, we should have output the policy $\overline{\pi}_w(a|s) = \frac{w(s,a)\overline{x}(s,a)}{\sum_{a'} w(s,a')\overline{x}(s,a')}$, which is inaccessible in practice without knowing the reference distribution $\mu$. In order to overcome such dilemma, we show that by properly constructing the estimated distribution $\hat{\mu}$, the $\overline{\pi}$ output by Algorithm 1 will be close enough to the ideal output $\overline{\pi}_w$.

## 4 The sample complexity of DPDL

### 4.1 Main results of DPDL

For the DPDL algorithm, the convergence and performance guarantee of the output policy $\bar{\pi}$ are summarized as the following theorem.

**Algorithm 1:** Deviation-controlled Primal-Dual Learning algorithm (DPDL)

---

**input** : Tolerance $\epsilon > 0$, confidential level $\delta > 0$, conservatism level $\kappa > 0$, stepsize $\eta_t > 0$, constants $\alpha_V, \alpha_\lambda, \alpha_x, N_e, \varsigma > 0$, and initial feasible solution $Z^1 = [V^1; \lambda^1; x^1]$.

1 Obtain $N_e$ samples from $\mathcal{D}$, let $N(s,a)$ be the times that the pair $(s,a)$ appears. Compute

$$\hat{\mu}(s,a) = \max\left(\frac{N(s,a)}{N_e}, \varsigma\right), \quad \forall (s,a) \in \mathcal{S} \times \mathcal{A}. \tag{10}$$

**for** $t = 1, \cdots, T-1$ **do**

2      Sample $\zeta_t = (s_t^0, s_t, a_t, s_t', r_t, \mathbf{u}_t)$ from $\rho_0 \times \mathcal{D}$;

3      Compute stochastic gradients $g_V^t := \hat{g}_V(Z^t; \zeta^t)$, $g_\lambda^t := \hat{g}_\lambda(Z^t; \zeta^t)$, and $g_x^t := \hat{g}_x(Z^t; \zeta^t)$;

4      Compute the stochastic mirror descent ascent update

$$V^{t+1} = \text{Proj}_\mathcal{V}\left(V^t - \eta_t \alpha_V^{-1} g_V^t\right),$$

$$\lambda^{t+1} = \underset{\lambda \in \Lambda}{\arg\min}\left(\langle g_\lambda^t, \lambda - \lambda^t\rangle + \frac{\alpha_\lambda}{\eta_t} \text{KL}(\lambda \parallel \lambda^t)\right), \tag{11}$$

$$x^{t+1} = \underset{x \in \mathcal{X}}{\arg\min}\left(-\langle g_x^t, x - x^t\rangle + \frac{\alpha_x}{\eta_t} \text{KL}(x \parallel x^t)\right),$$

5 Compute the average iterate $\overline{x} = \frac{1}{T}\sum_{t=1}^T x^t, \overline{V} = \frac{1}{T}\sum_{t=1}^T V^t, \overline{\lambda} = \frac{1}{T}\sum_{t=1}^T \lambda^t$;

6 Compute $\overline{\pi}(a|s) = \frac{\overline{x}(s,a)}{\sum_{a'} \overline{x}(s,a')}$, for all $(s,a)$;

**output:** Policy $\overline{\pi}$ and the approximate solution $\overline{x}$.

---

**Theorem 4.1.** *Suppose that Algorithm 1 runs with $\eta_t \equiv \frac{1}{\sqrt{T}}$, $\kappa = 5\varphi\epsilon$, $\alpha_\lambda = \frac{1}{1-\gamma}\sqrt{\frac{\psi}{\log I}}$, $\alpha_V = \varphi\sqrt{\frac{\psi}{|\mathcal{S}|}}$, $\alpha_x = \frac{1}{\varphi(1-\gamma)}\sqrt{\frac{\mathcal{N}\psi}{\log\psi}}$, and $\psi \geq C^*$. Then for any fixed $\epsilon \in \left(0, \frac{1}{10(1-\gamma)}\right]$, and $T \geq c_o \frac{\mathcal{N}\psi\iota}{\varphi^2(1-\gamma)^4\epsilon^2}$, where $\iota = \log\left(\frac{\psi|\mathcal{S}||\mathcal{A}|I}{\delta}\right)$ and $c_o$ is a universal constant, the output policy $\overline{\pi}$ of DPDL satisfies the following with probability at least $1 - \delta$*

$$J(\pi^*) - J(\overline{\pi}) \leq \mathcal{O}(\epsilon), \quad \text{and} \quad J_i^u(\overline{\pi}) \geq 0, \forall i \in [I].$$

*When $\psi = \mathcal{O}(C^*)$, DPDL needs at most $\tilde{\mathcal{O}}\left(\frac{\mathcal{N}C^*}{\varphi^2(1-\gamma)^4\epsilon^2}\right)$ samples to find a safe $\mathcal{O}(\epsilon)$-optimal policy.*

**Remark 4.2.** *When the prior knowledge of $C^*$ is not available, and the selected parameter $\psi < C^*$ but the Slater's condition for $\Pi(\psi)$ still holds, the output policy $\overline{\pi}$ of DPDL satisfies that*

$$J(\overline{\pi}) \geq \max_{\pi \in \Pi(\psi) \cap \mathfrak{S}} J(\pi) - \mathcal{O}(\epsilon) \quad \text{and} \quad J_i^u(\overline{\pi}) \geq -\epsilon_{\text{approx}}, \forall i \in [I],$$

*where $\mathfrak{S}$ denotes the set of safe policies, and $\epsilon_{\text{approx}}(\psi) := J(\pi^*) - \max_{\pi \in \Pi(\psi) \cap \mathfrak{S}} J(\pi)$ in some sense measures the "sub-optimality" of the policy class $\Pi(\psi)$. In case a fixed sub-optimality gap $\epsilon$ is given, such difficulty of unknown $C^*$ also appears in the guarantees provided in previous works [15, 21, 22, 28, 30, 31].*

A simple approach to resolve the difficulty of an unknown $C^*$ is discussed later in Section 6.

### 4.2 The analysis of DPDL

We break down the analysis of Theorem 4.1 into the following steps. First of all, we provide a proper choice of $N_e$ and $\varsigma$ so that $\hat{\mu}$ is close enough to $\mu$. See proof in Appendix B.

**Proposition 4.3.** *Denote $\epsilon_e = \frac{\epsilon}{100}$, and let $\varsigma = \frac{\varphi(1-\gamma)^2\epsilon_e}{2\mathcal{N}\psi}$, and $N_e \geq \frac{512\mathcal{N}\psi}{\varphi^2(1-\gamma)^4\epsilon_e^2} \cdot \log\left(\frac{6|\mathcal{S}||\mathcal{A}|}{\delta}\right)$. Then with probability at least $1 - \delta/3$, the estimated reference distribution $\hat{\mu}$ defined by (10) satisfies the following properties simultaneously: (1). $\frac{\mu(s,a)}{\hat{\mu}(s,a)} \leq 2$, and $\hat{\mu}(s,a) \geq \varsigma$, for all $s, a$; (2). For any $\pi \in \Pi(\psi)$, $W^{-1}\nu^\pi \in \mathcal{X}$; (3). For any $x \in \mathcal{X}$, $\|Wx - x\|_1 \leq \varphi(1-\gamma)\epsilon_e$.*

All the rest of our analyses are all conditioning on the success of Proposition 4.3. It is worth noting that in Proposition 4.3, (3) clarifies the validity of constructing the output policy $\overline{\pi}$ with $\overline{x}$ instead of $W\overline{x}$; (2) explains why the feasible region $\mathcal{X}$ is defined as (8); and (1), combined with the carefully specified feasible domains, provides the proper upper bounds on the magnitude and variance of the unbiased gradient estimators in (9). A very detailed discussion is provided in Appendix C. In particular, for the $\widehat{g}_x(\cdot)$ estimator, an explicit $\mathcal{O}(\mathcal{N})$ dependence has been established for both the magnitude and variance, which plays a crucial role in deriving the optimal $\mathcal{O}(\min\{|\mathcal{S}||\mathcal{A}|, |\mathcal{S}|+I\})$ dependence on $|\mathcal{S}|$, $|\mathcal{A}|$ and $I$. Let us define the following gap to measure the performance of the output $\overline{x}$ w.r.t. problem (7):

$$\mathrm{Gap}(\overline{x}) := \max_{x \in \mathcal{X}} \min_{V \in \mathcal{V}, \lambda \in \Lambda} \mathcal{L}_w(V, \lambda, x) - \min_{V \in \mathcal{V}, \lambda \in \Lambda} \mathcal{L}_w(V, \lambda, \overline{x}). \tag{12}$$

Based on the properly bounded gradient estimators, a high probability bound for $\mathrm{Gap}(\overline{x})$ is established in the following theorem. Its proof is detailed in Appendix D.

**Theorem 4.4.** *Suppose the constants $\eta_t$, $\alpha_V$, $\alpha_\lambda$, $\alpha_x$ and $\kappa$ are chosen the same as Theorem 4.1. Then there is a universal constant $c_o$ such that, as long as $T \geq c_o \frac{\mathcal{N}\psi\iota}{\varphi^2(1-\gamma)^4\epsilon^2}$, the output $\overline{x}$ satisfies $\mathrm{Gap}(\overline{x}) \leq \frac{\epsilon}{2}$ with probability at least $1 - \delta/3$.*

Given Theorem 4.4, we finalize the proof of Theorem 4.1 by properly transforming the bound on $\mathrm{Gap}(\overline{x})$ to the expected reward gap and the constraint violation on the original CMDP problem (1), which is discussed in details in Appendix E.

## 4.3 Extension to asynchronous setting

In some situations, an independent dataset that satisfies Assumption 2.2 may not be available. Instead, the dataset may have the following asynchronous structure.

**Assumption 4.5.** *The asynchronous dataset $\mathcal{D}_{async}$ is a single sample trajectory generated by some behavior policy $\pi_b$. Namely, what we observe is a sequence $\{s_t, a_t, r_t, \mathbf{u}_t\}_{t \geq 1}$ generated under $\pi_b$. We assume the Markov Chain $\{(s_t, a_t)\}_{t \geq 1}$ is irreducible, aperiodic and uniformly ergodic, with the stationary distribution $\mu$ and the mixing time $t_{\mathrm{mix}} < +\infty$.*

The asynchronous data structure introduced here is frequently considered in RL, for example, the asynchronous Q-learning [14]. However, to our best knowledge, this type of offline data has yet been considered under the assumption of a finite single-policy concentrability. In this situation, we set $\zeta_t = (s_t^0, s_t, a_t, s_{t+1}, r_t, \mathbf{u}_t)$ in the DPDL method (Algorithm 1), where $s_t^0 \sim \rho_0$ and $(s_t, a_t, s_{t+1}, r_t, \mathbf{u}_t)$ is the tuple in the $t$-th time step of the asynchronous dataset. The sample complexity of the DPDL Algorithm under Assumption 4.5 is established as follows.

**Theorem 4.6.** *Under Assumption 4.5, we follow the choice of constants in Theorem 4.1. Then given any fixed $\epsilon \in \left(0, \frac{1}{10(1-\gamma)}\right]$, $\psi \geq C^*$, and $T \geq c_o' \frac{t_{\mathrm{mix}}^2 \mathcal{N}\psi\iota^3}{\varphi^2(1-\gamma)^4\epsilon^2}$, the output policy $\overline{\pi}$ of DPDL satisfies the following with probability at least $1 - \delta$*

$$J(\pi^*) - J(\overline{\pi}) \leq \epsilon \qquad and \qquad J_i^u(\overline{\pi}) \geq 0, \forall i \in [I].$$

*Here $\iota = \log(T|\mathcal{S}||\mathcal{A}|I/\delta)$ and $c_o'$ is a universal constant. Therefore, when $\psi = \mathcal{O}(C^*)$, DPDL needs at most $\tilde{\mathcal{O}}\left(\frac{t_{\mathrm{mix}}^2 \mathcal{N} C^*}{\varphi^2(1-\gamma)^4\epsilon^2}\right)$ samples to find a safe $\epsilon$-optimal policy.*

The main framework for proving Theorem 4.6 is similar to that in Section 4.2, thus we present the proof in the Appendix H. However, compared to the synchronous setting, a key difficulty here is that the gradient estimators $\widehat{g}_V(Z^t; \zeta_t)$, $\widehat{g}_\lambda(Z^t; \zeta_t)$, and $\widehat{g}_x(Z^t; \zeta_t)$ are no longer unbiased, because the samples $\{\zeta_t\}_{t=1}^T$ are obtained from a sample path. This brings further difficulties in the analysis because the variance of the estimators can be amplified by the correlation between samples.

The basic idea to deal with this difficulty is to leverage the mixing property of the uniformly ergodic Markov chain. Take the $\widehat{g}_x(\cdot)$ estimator for example, the bias can be well controlled as long as $T$ is selected larger than the mixing time $t_{\mathrm{mix}}$ of the sample path, which can be illustrated by the following decomposition

$$\widehat{g}_x(Z^t; \zeta_t) - \nabla_x \mathcal{L}_w(Z^t) = \underbrace{\widehat{g}_x(Z^t; \zeta_t) - \widehat{g}_x(Z^{t-\tau}; \zeta_t) + \nabla_x \mathcal{L}_w(Z^{t-\tau}) - \nabla_x \mathcal{L}_w(Z^t)}_{\text{order } \mathcal{O}(\tau\eta)}$$

$$+ \underbrace{\widehat{g}_x(Z^{t-\tau}; \zeta_t) - \mathbb{E}\left[\widehat{g}_x(Z^{t-\tau}; \zeta_t) \big| Z^{t-\tau}\right]}_{\text{zero mean}} + \underbrace{\mathbb{E}\left[\widehat{g}_x(Z^{t-\tau}; \zeta_t) \big| Z^{t-\tau}\right] - \nabla_x \mathcal{L}_w(Z^{t-\tau})}_{\text{order } \mathcal{O}(\exp(-\tau/t_{\mathrm{mix}}))}. \tag{13}$$

When $t = \tilde{\Omega}(t_{\mathrm{mix}})$, one can bound the bias of $\hat{g}_x(Z^t; \zeta_t)$ by $\tilde{\mathcal{O}}(t_{\mathrm{mix}}\eta)$ with suitably chosen $\tau$.

# 5 Lower Bound of Sample Complexity for Learning CMDP

In this section we will discuss whether the DPDL Algorithm is the near-optimal and whether the Slater's condition (Assumption 2.4) is necessary in achieving zero constraint violation. We answer these questions affirmatively by establishing the following theorems.

**Theorem 5.1.** *Suppose $S \geq 4$, $A \geq 3$, $I \geq 8$, $C \geq 2$, $\gamma \in [\frac{1}{2}, 1)$, $N \geq 1$. For any learning algorithm $\mathfrak{A}$, there exists a CMDP $\mathcal{M} = (\mathcal{S}, \mathcal{A}, \mathbb{P}, r, (u_i)_{i \in [I]}, \gamma, \rho_0)$ and a reference distribution $\mu$, such that the following hold true.*

*(1) $|\mathcal{S}| \leq 4S + 1$, $|\mathcal{A}| \leq A$, and the concentrability coefficient $C^*$ for $\mathcal{M}$ and $\mu$ satisfies $C^* \leq C$.*

*(2) Let $\hat{\pi}$ be the policy output by $\mathfrak{A}$ given $N$ offline samples from $\mu$, and let $\pi^*$ be the optimal policy, then at least one of the following two inequalities hold true:*

$$\mathbb{E}_{\mathcal{M}, \mathfrak{A}}[J(\pi^*) - J(\hat{\pi})] \gtrsim \min\left\{\frac{1}{1-\gamma}, \sqrt{\frac{\min\{SA, S+I\}C}{(1-\gamma)^3 N}}\right\}, \quad \text{and} \quad \mathbb{E}_{\mathcal{M}, \mathfrak{A}}[\mathrm{violation}(\hat{\pi})] \gtrsim 1,$$

*where $\mathrm{violation}(\hat{\pi}) := \sum_{i=1}^I [J_i^u(\hat{\pi})]_-$, and $J_i^u$ is the utility w.r.t. the constraints $J_i^u \geq 0, \forall i \in [I]$.*

For DPDL, the constraint violation is guaranteed to be zero with high probability, then only the first inequality is valid for our method, which indicates an $\Omega\left(\frac{\mathcal{N}C^*}{(1-\gamma)^3\epsilon^2}\right)$ sample complexity lower bound. Therefore, the complexity of DPDL is nearly optimal up to an $\tilde{\mathcal{O}}\left(\frac{1}{1-\gamma}\right)$ factor. Besides the lower bound, we also establish the necessity of the Slater's condition in ensuring zero violation.

**Theorem 5.2.** *Let $S, A, C, \gamma$ be the same as Theorem 5.1. For any algorithm $\mathfrak{A}$, there exists a CMDP $\mathcal{M} = (\mathcal{S}, \mathcal{A}, \mathbb{P}, r, (u_i)_{i \in [I]}, \gamma, \rho_0)$ with $I = 1$, $|\mathcal{S}| \leq S$, $|\mathcal{A}| \leq A$ and a reference distribution $\mu$ with $C^* \leq C$, such that $\mathbb{E}_{\mathcal{M}, \mathfrak{A}}[\mathrm{violation}(\hat{\pi})] \gtrsim \min\left\{\frac{1}{1-\gamma}, \sqrt{\frac{SC}{(1-\gamma)^3 N}}\right\}$, where $\hat{\pi}$ is the output policy of $\mathfrak{A}$ given $N$ samples from $\mu$.*

Theorem 5.2 is obtained by utilizing the same idea as Theorem 5.1. Thus we only discuss the derivation of Theorem 5.1, while moving all the details to Appendix F.

For offline CMDPs, the fixed data distribution $\mu$ fully dominates the frequency of exploring the state-action pairs. Therefore, intuitively, the hard CMDP instances will be the ones with a large support $\mathrm{supp}(\nu^{\pi^*})$ that widely spreads across the less frequently visited station-action pairs of $\mu$. Based on this intuition, we design a basic block of CMDP presented in Fig. 1, which is essentially a constrained bandit with $2K + 1$ arms. The instance $\mathcal{M}$ will be $S$ replicas of the basic blocks, plus an extra "null" state $s_{-1}$ to control $C^*$. In this discussion, we only consider the case where $I \simeq KS$, the more general construction that cover full range of $I$ is presented in the appendix.

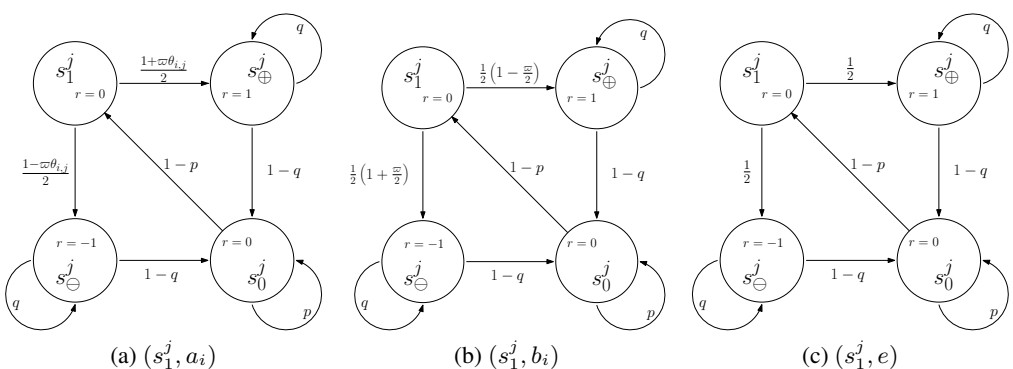

Figure 1: Transition dynamics of the $j$th replica under different actions, $i \in [K]$.

**State, action and transition**. At the states $s_\oplus^j, s_\ominus^j, s_0^j$, there is no action to be taken. At each state $s_1^j$, there are $2K + 1$ actions $a_1, b_1, \cdots, a_K, b_K, e$. The transition dynamics of the $j$th replica under different actions are illustrated in Fig. 1 where the directed arcs and the numbers associated with

them are the transitions and the corresponding probabilities, where $p = \frac{1}{2-\gamma}$ and $q = 2 - \frac{1}{\gamma}$ are some constants, while $\varpi$ and $\theta_{i,j} \in \{-1, 1\}$, $\forall i, j$ are parameters to be designed.

**Constraints and Reward**. By carefully selecting the $u_i$'s, one can construct a set of $I = 2SK$ constraints that indicate $\pi(a_i|s_1^j) \leq \pi(b_i|s_1^j) \leq \frac{1}{4K}$, $\forall i, j$. For the reward, we set $r(s_1^j) = r(s_0^j) = 0$, $r(s_\oplus^j) = 1$, and $r(s_\ominus^j) = -1$, regardless of the actions. At any replica $j$, we can view $a_i$, $b_i$, and $e$ as bandit arms with (cumulative) reward $c\varpi\theta_{i,j}$, $-\frac{c\varpi}{2}$, and 0 respectively, for some $c > 0$. When $\theta_{i,j} = -1$, one would rather pick $e$. But when $\theta_{i,j} = 1$, due to the constraint $\pi(a_i|s_1^j) \leq \pi(b_i|s_1^j) \leq \frac{1}{4K}$, picking $a_i$ and $b_i$ with equal probability $\frac{1}{4K}$ will be optimal. In fact, this $\frac{1}{4K}$ upper bound forces the support of the optimal policy to widely spread across the $(i, j)$'s where $\theta_{i,j} = 1$, and the task of learning is essentially determining whether $\theta_{i,j} = 1$ for each $(i, j)$.

**Optimal policy**. Based on the above discussion, it is not hard to see that the unique optimal policy is $\pi^{*,\theta}(a_i|s_1^j) = \pi^{*,\theta}(b_i|s_1^j) = \frac{\mathbb{I}\{\theta_{i,j}=1\}}{4K}$ and $\pi^{*,\theta}(e|s_1^j) = 1 - \frac{1}{2K}\sum_{i=1}^{K}\mathbb{I}\{\theta_{i,j}=1\}$.

Finally, with the above $\pi^{*,\theta}$ and a proper initial distribution $\rho_0$, the occupancy measure can be explicitly computed and a reference distribution $\mu$ with concentrability coefficient $C^* \leq C$ can be designed. Moreover, for any policy $\hat{\pi}$, we consider $\hat{\theta}_{i,j}(\hat{\pi}) := 8K\hat{\pi}(a_i|s_1^j) - 1$, then

$$\mathcal{L}(\hat{\pi}; \theta) := \left[J(\pi^{*,\theta}; \theta) - J(\hat{\pi}; \theta)\right]_+ + \frac{\gamma\varpi}{1-\gamma}\text{violation}(\hat{\pi}; \theta) \geq \frac{\gamma^2\varpi\|\hat{\theta}(\hat{\pi}) - \theta\|_1}{64KS(1-\gamma)}.$$

Namely, if $\hat{\theta}(\hat{\pi})$ is not close enough to the underlying parameter $\theta$, the policy $\hat{\pi}$ will incur a considerable reward gap or constraint violation. By setting $\varpi = \min\left\{\sqrt{\frac{(SK-3)C}{16(1-\gamma)N}}, \frac{1}{2}\right\}$ to be a small enough number, any two CMDP instances with different $\theta$ parameters will be non-distinguishable, given $N$ samples from $\mu$. According to [9] and [24], there exists a subset $\Theta \subseteq \{-1, 1\}^{SK}$ such that $|\Theta| \geq \exp(SK/8)$, and $\|\theta - \theta'\|_1 \geq \frac{SK}{2}$ for any pair of different $\theta, \theta' \in \Theta$. In other words, there will be at least $\exp(SK/8)$ CMDP instances with different enough $\theta$ parameters while being non-distinguishable under $N$ samples. Then the rest of the arguments will follow by applying the generalized Fano's inequality [3]. A detailed proof is provided in Appendix F.

## 6  Adaptive deviation-control framework of DPDL

We should notice that in both Theorems 4.1 and 4.6, it has been explicitly emphasized that a prior belief $\psi \geq C^*$ is required. Otherwise, both the reward and the constraints will suffer an extra loss of $\epsilon_{\text{approx}}(\psi)$. In this section, we propose an adaptive deviation-control framework (Algorithm 2) to handle the practical situation where no such prior knowledge is available.

---

**Algorithm 2:** The Adaptive-DPDL framework

**input** : Sub-optimality $\epsilon$, confidence level $\delta$.

1 Initialize $\psi_1$, default $J^K \equiv -\infty$, for $K = 0, 1, 2, ...$;
2 **for** $K = 1, 2, \cdots$ **do**
3      Call DPDL with $\psi = \psi_K$, obtain an approximate solution $x^{(K)}$ and the policy $\pi^{(K)}$;
4      **if** *VERIFY*$\left(x^{(K)}; \epsilon, \delta\right)$ == *TRUE* **then**
5          Compute $\widehat{J}(\pi^{(K)})$ as an $\mathcal{O}(\epsilon)$-accurate estimator of $J(\pi^{(K)})$, set $J^K = \widehat{J}(\pi^{(K)})$;
6      **if** $-\infty < J^K \leq J^{K-1} + \mathcal{O}(\epsilon)$ **then Terminate**;
7      Set $\psi_{K+1} = 2\psi_K$;

**output:** Policy $\pi^{(K)}$.

---

At a high-level, Algorithm 2 consists of the following steps.

**Verification** For the output $\overline{x}$ of the DPDL, we develop a verification method VERIFY$(\overline{x}; \epsilon, \delta)$ that, with probability at least $1 - \delta$, returns TRUE only when following two statements hold: (1). The vector $\overline{\nu} := W\overline{x}$ satisfies $\|\sum_a(\mathbf{I} - \gamma\mathbb{P}_a)\overline{\nu}_a - \rho_0\|_1 = \mathcal{O}(\epsilon)$, which essentially checks whether $\overline{\nu}$ is approximately a valid occupancy measure; (2). The policy $\overline{\pi}$ induced by $\overline{x}$ is safe. At step $K$, if any one of the two statements does not hold, we immediately know $\psi_K < C^*$ due to the analysis of Theorem 4.1. Consequently, we to double the coefficient $\psi_{K+1} \leftarrow 2\psi_K$ in the next iteration.

**Certifying performance improvement** When VERIFY$(\overline{x}; \epsilon, \delta)$ returns TRUE, then it holds that $j_0(\psi) = J(\pi^{(K)}) + \mathcal{O}(\epsilon)$, where $j_0(\psi)$ denotes the optimal value of problem (7) with $\kappa = 0$. That is, one can estimate $j_0(\psi)$ with $\widehat{J}(\pi^{(K)})$ if VERIFY$(\overline{x}; \epsilon, \delta) = $ TRUE. As long as VERIFY returns TRUE for two consecutive runs, and the performance improvement is small, i.e., $j_0(\psi_K) - j_0(\psi_{K-1}) = \mathcal{O}(\epsilon)$, then Lemma 6.1 guarantees that the safe policy $\pi^{(K)}$ is $\mathcal{O}(\frac{C^*}{\psi_K}\epsilon)$-optimal.

**Lemma 6.1.** *The function $j_0(\cdot)$ is strictly increasing in the range $\psi \in [1, C^*]$, and $j_0(\psi) = J(\pi^*)$ for $\psi \geq C^*$. For any $\psi < \psi' \leq C^*$, it holds that*

$$J(\pi^*) - j_0(\psi') \leq \frac{C^* - \psi}{\psi' - \psi}\left(j_0(\psi') - j_0(\psi)\right).$$

Detailed descriptions of VERIFY and Adaptive-DPDL are presented in Appendix G, and so does the proof of the following theorem.

**Theorem 6.2.** *Fixed $\epsilon \in \left(0, \frac{1}{10(1-\gamma)}\right], \delta \in (0, 1)$. Then with probability at least $1 - \delta$, Adaptive-DPDL stops at step $K$ such that $\psi_K \leq 4C^*$ and outputs the safe policy $\pi^{(K)}$ with sub-optimality gap $J(\pi^*) - J(\pi^{(K)}) \leq \mathcal{O}\left(\frac{C^*}{\psi_K}\epsilon\right)$. Moreover, there exists a (problem dependent) constant $\epsilon_0(\mathcal{M})$ such that, if $\epsilon \leq \epsilon_0(\mathcal{M})$, then it must hold that $\psi_K \in [C^*, 2C^*)$ and $\pi^{(K)}$ is $\epsilon$-optimal.*

Intuitively, the Adaptive-DPDL will quickly terminate within $\mathcal{O}(\log_2 C^*)$ calls of DPDL, resulting in a total samples complexity of $\tilde{\mathcal{O}}\left(\frac{\mathcal{N}C^*}{(1-\gamma)^4\epsilon^2}\right)$.

## Acknowledgement

J. Zhang is supported by Ministry of Education, Singapore, grant R-266-000-158-133. Z. Wen is supported in part by the NSFC grant 11831002. F. Chen is supported in part by the elite undergraduate training program of School of Mathematical Sciences in Peking University.

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
