# A  Efficiently solving the subproblems of DPDL

In this section, we describe how to efficiently solve the subproblems (11) in the DPDL Algorithm. In the following discussion, at most $\tilde{\mathcal{O}}(|\mathcal{S}||\mathcal{A}| + I)$ flops are needed to compute the update.

## A.1  Closed form solution for the $V$-update

The dual variable $V$ is updated by the formula $V^{t+1} = \text{Proj}_{\mathcal{V}}\left(V^t - \eta_t \alpha_V^{-1} g_V^t\right)$, where $\mathcal{V}$ is an $\ell_\infty$ normal ball defined as $\mathcal{V} := \left\{V \in \mathbb{R}^{|\mathcal{S}|} : \|V\|_\infty \leq R_{\mathcal{V}}\right\}$, $R_{\mathcal{V}} = \frac{8}{1-\gamma}\left(1 + \frac{2}{\varphi}\right)$. For any vector $V \in \mathbb{R}^{|\mathcal{S}|}$, the Euclidean projection $V_+ = \text{Proj}_{\mathcal{V}}(V)$ can be written as a simple truncation

$$
V_+(s) = \begin{cases} -R_{\mathcal{V}}, & \text{if } V(s) < -R_{\mathcal{V}}, \\ V(s), & \text{if } -R_{\mathcal{V}} \leq V(s) \leq +R_{\mathcal{V}}, \\ +R_{\mathcal{V}}, & \text{if } V(s) > +R_{\mathcal{V}}, \end{cases} \qquad \text{for} \qquad \forall s \in \mathcal{S}.
$$

This update will need $\mathcal{O}(1)$ flops due to the special structure of $g_V^t$.

## A.2  Closed form solution for the $\lambda$-update

The dual variable $\lambda$ is updated by the formula $\lambda^{t+1} = \arg\min_{\lambda \in \Lambda}\left(\langle g_\lambda^t, \lambda - \lambda^t\rangle + \frac{\alpha_\lambda}{\eta_t}\text{KL}(\lambda\|\lambda^t)\right)$, where $\Lambda$ is the nonnegative part of an $\ell_1$ norm ball $\Lambda = \left\{\lambda \in \mathbb{R}_{\geq 0}^I : \|\lambda\|_1 \leq R_\Lambda\right\}$, $R_\Lambda = \frac{8}{\varphi}$. The solution to this subproblem has the following closed form formula

$$
\lambda^{t+1} = \lambda^{t+\frac{1}{2}} \min\left\{\frac{R_\Lambda}{\left\|\lambda^{t+\frac{1}{2}}\right\|_1}, 1\right\},
$$

where $\lambda^{t+\frac{1}{2}} = \lambda^t \exp(-\frac{\eta_t}{\alpha_\lambda} g_\lambda^t)$ is an intermediate point. This update will need $\mathcal{O}(I)$ flops.

## A.3  Efficient implementation of the $x$-update

Compared to the previous two updates, the subproblem for $x$-update does not have a closed form solution. By carefully discussing the KKT condition of the problem and utilizing the special structure of $\mathcal{X}$ and $g_x^t$, we reduce the problem to finding the root of a monotonically decreasing 1-dimensional function. If the bisection method is applied to find the root, then in total $\tilde{\mathcal{O}}(|\mathcal{S}||\mathcal{A}|)$ flops are needed. We present the details as follows. For notational simplicity, we rewrite the subproblem as follows.

**Problem.** *Given a set $\mathcal{Y}$ defined by the linear constraints*

$$
\mathcal{Y} := \left\{y \in \mathbb{R}^n : 0 \leq y_i \leq a_i, \sum_{i=1}^n y_i \leq B_1, \sum_{i=1}^n c_i y_i \leq B_2\right\},
$$

*where $B_1, B_2 > 0$, and $c_i > 0$ are some constants. Let $y^0 \in \mathcal{Y}$, $y^0 > 0$, and let $g \in \mathbb{R}^n$ be a vector that has at most 1 non-zero entry. Then the goal is to solve*

$$
y^* = \arg\min_{y \in \mathcal{Y}}\left(\langle y, g\rangle + \text{KL}(y \parallel y^0)\right). \tag{14}
$$

Without loss of generality, we assume $g_2 = \cdots = g_n = 0$. For problem (14), we introduce two Lagrangian multipliers to the coupling constraints $\sum_{i=1}^n y_i \leq B_1, \sum_{i=1}^n c_i y_i \leq B_2$, while remaining the coordinately separable constraints $0 \leq y_i \leq a_i$ in the problem. Thus we get the following Lagrangian function:

$$
L(y, \alpha, \beta) := y_1 g_1 + \text{KL}(y\|y^0) + \alpha\left(\sum_i y_i - B_1\right) + \beta\left(\sum_i c_i y_i - B_2\right). \tag{15}
$$

By the strong convexity of KL divergence, there is a unique KKT point $(y^*, \alpha^*, \beta^*)$ of problem (14). Note that $y^* = \arg\min_{y_i \in [0,a_i], \forall i} L(y, \alpha^*, \beta^*)$. Because $y_0^i > 0$, we know

$$
\lim_{y_i \to 0+} \nabla_{y_i} L(y, \alpha^*, \beta^*) = \lim_{y_i \to 0+} g_1 \cdot \mathbb{I}\{i = 1\} + \alpha^* + c_i \beta^* + \log y_i - \log y_i^0 = -\infty,
$$

we know $y_i^*$ will not be 0. Thus we can write the KKT condition for problem (14) as

$$\begin{cases} \nabla_{y_i} L(y^*, \alpha^*, \beta^*) \leq 0, & \text{if } y_i^* = a_i, \quad \forall i \in [n], \\ \nabla_{y_i} L(y^*, \alpha^*, \beta^*) = 0, & \text{if } y_i^* \in (0, a_i), \quad \forall i \in [n], \\ \alpha^* \big( \sum_i y_i^* - B_1 \big) = 0, & \beta^* \big( \sum_i c_i y_i^* - B_2 \big) = 0, \\ y^* \in \mathcal{Y}, \alpha^* \geq 0, \beta^* \geq 0. \end{cases} \tag{16}$$

For $i = 2, ..., n$, the condition $\nabla_{y_i} L(y^*, \alpha^*, \beta^*) \leq 0$ implies that $y_i^* \leq y_i^0 \exp(-\alpha^* - c_i \beta^*)$. Note that $\alpha^*, \beta^* \geq 0, c_i > 0, y_i^0 \leq a_i$. If $y_i^* < a_i$, then $\nabla_{y_i} L(y^*, \alpha^*, \beta^*) = 0$ indicates that $y_i^* = y_i^0 \exp(-\alpha^* - c_i \beta^*)$. If $y_i^* = a_i$, then the only possibility is $y_i^0 = a_i$ happen to hold and $\alpha^* = \beta^* = 0$, in this case, we still have $y_i^* = y_i^0 \exp(-\alpha^* - c_i \beta^*)$. A similar formula can also be derived for $y_1^*$. Therefore, utilizing the feasibility of the point $y^0$, we solve the first two rows of the KKT condition and get

$$\begin{cases} y_1^*(\alpha^*, \beta^*) = \min \left\{ y_1^0 \exp(-g_1 - \alpha^* - c_1 \beta^*), a_1 \right\}, \\ y_i^*(\alpha^*, \beta^*) = y_i^0 \cdot \exp\{-\alpha^* - c_i \beta^*\}, \quad \text{for } i = 2, ..., n. \end{cases} \tag{17}$$

Here, we write $y_i^*$ as functions of $\alpha^*, \beta^*$ for the ease of later discussion. Next, we solve the third row of the KKT condition (16) by considering the following cases.

**Case 1:** $\beta^* = 0, \alpha^* = 0$**.** In this case, if $y^*(0, 0), \alpha^* = 0, \beta^* = 0$ satisfies (16), then $y^*(0, 0)$ is the solution to (14). Otherwise we conclude that $\alpha^* = \beta^* = 0$ is not true.

**Case 2:** $\beta^* = 0, \alpha^* > 0$**.** In this case, the KKT condition tells us that $\sum_i y_i^* = B_1$. Together with (17), we have the following two possible solutions to $\alpha^*$

$$\begin{cases} \alpha_1 = \ln \left( \frac{y_2^0 + \cdots + y_n^0}{B_1 - a_1} \right), & \text{corresponds to } y_1^* = a_1, \\ \alpha_2 = \ln \left( \frac{e^{-g_1} \cdot y_1^0 + y_2^0 + \cdots + y_n^0}{B_1} \right), & \text{corresponds to } y_1^* = y_1^0 \exp(-g_1 - \alpha^*). \end{cases}$$

Then if $y^*(\alpha_1, 0), \alpha^* = \alpha_1, \beta^* = 0$ satisfies (16), we conclude that $y^*(\alpha_1, 0)$ is the solution to (14). If $y^*(\alpha_2, 0) \in \mathcal{Y}, \alpha^* = \alpha_2, \beta^* = 0$ satisfies (16), we conclude that $y^*(\alpha_2, 0)$ is the solution to (14). Otherwise, we know $\alpha^* > 0, \beta^* = 0$ is not possible.

**Case 3:** $\beta^* > 0, \alpha^* = 0$**.** In this case, the KKT condition tells us that $\sum_i c_i y_i^* = B_2$. Denote $\hat{y}_1^0 = y_1^0 \exp(-g_1), \hat{y}_i^0 = y_i^0, i = 2, ..., n$. In this case, depending on the value of $y_1^*$ we set

$$\begin{cases} \beta_1 = \text{Root}_{\beta > 0} \big\{ \sum_{i=2}^n c_i \hat{y}_i^0 \exp(-c_i \beta) = B_2 - c_1 a_1 \big\}, \\ \beta_2 = \text{Root}_{\beta > 0} \big\{ \sum_{i=1}^n c_i \hat{y}_i^0 \exp(-c_i \beta) = B_2 \big\}. \end{cases}$$

Note that in both cases, the problem is finding the positive root of a 1-dimensional monotonically decreasing function, which can be solved efficiently. These equations should either have one unique positive solution or no positive solution at all. If there is no positive root, then $\text{Root}_{\beta > 0}$ will return FALSE. One can easily determine whether there is a positive solution. For example, due to the monotonicity, the first equation will have a positive solution if and only if $\sum_{i=2}^n \hat{c}_i y_i^0 > B_2 - c_1 a_1$.

Similar to case 2, we check the feasibility of $\{y^*(0, \beta_1), \alpha^* = 0, \beta^* = \beta_1\}$ and $\{y^*(0, \beta_2), \alpha^* = 0, \beta^* = \beta_2\}$ w.r.t. (16). If any one of them is feasible to the KKT condition, then it will be the solution to (14). Otherwise, we know $\alpha^* = 0, \beta^* > 0$ is not possible.

**Case 4:** $\beta^* > 0, \alpha^* > 0$**.** In this case, the KKT condition implies that $\sum_i c_i y_i^* = B_1, \sum_i c_i y_i^* = B_2$. Let us inherit the $\hat{y}$ notation from Case 3. Then we need to solve the following group of equations

$$\begin{cases} \sum_{i=2}^n \hat{y}_i^0 \exp(-\alpha_3 - c_i \beta_3) = B_1 - a_1, \\ \sum_{i=2}^n c_i \hat{y}_i^0 \exp(-\alpha_3 - c_i \beta_3) = B_2 - c_1 a_1 \end{cases} \quad \text{or} \quad \begin{cases} \sum_{i=1}^n \hat{y}_i^0 \exp(-\alpha_4 - c_i \beta_4) = B_1, \\ \sum_{i=1}^n c_i \hat{y}_i^0 \exp(-\alpha_4 - c_i \beta_4) = B_2 \end{cases}$$

We should notice that in both cases, as soon as we determine the value of $\beta$, then $\alpha$ will have a closed form formula given $\beta$. To demonstrate how to determine $\beta$, let us take the second group of equations for example. Taking the quotient between the two equations cancels $\alpha_4$, we get the following equation of $\beta_4$

$$f(\beta_4) := \frac{\sum_{i=1}^n c_i \hat{y}_i^0 \exp(-c_i \beta_4)}{\sum_{i=1}^n \hat{y}_i^0 \exp(-c_i \beta_4)} = \frac{B_2}{B_1}. \tag{18}$$

By Cauchy's inequality, we know $f'(\beta) < 0$ holds for $\forall \beta \in \mathbb{R}$ if $c_i \neq c_j$ for some $i, j$. In details

$$f'(\beta) = \frac{\left(\sum_{i=1}^n c_i \hat{y}_i^0 \exp(-c_i \beta)\right)^2 - \left(\sum_{i=1}^n \hat{y}_i^0 \exp(-c_i \beta)\right)\left(\sum_{i=1}^n c_i^2 \hat{y}_i^0 \exp(-c_i \beta)\right)}{\left(\sum_{i=1}^n \hat{y}_i^0 \exp(-c_i \beta)\right)^2} < 0.$$

Hence, $f$ is again a monotonically decreasing function, and finding its positive root can be implemented efficiently. After finding $\beta_4$, one immediately know $\alpha_4 = \ln\left(\frac{\sum_{i=1}^n \hat{y}_i^0 \exp(-c_i \beta_4)}{B_1}\right)$.

Finally, we need to check the feasibility of $\{y^*(\alpha_3, \beta_3), \alpha^* = \alpha_3, \beta^* = \beta_3\}$ and $\{y^*(\alpha_4, \beta_4), \alpha^* = \alpha_4, \beta^* = \beta_4\}$ w.r.t. (16). If any one of them is feasible to the KKT condition, then it will be the solution to (14). Otherwise, we know $\alpha^* > 0, \beta^* > 0$ is not possible. Due to the existence of a KKT pair, at least one of the 4 cases will return us a solution.

## B    Proof of Proposition 4.3

For the analysis of Proposition 4.3 and later results, let us first introduce a vector version of the Bernstein's inequality, which is a direct specification of the Freedman's inequality of matrix martingale [23]. To prove the current proposition, we only need the scalar case of the following lemma.

**Lemma B.1** (Vector Bernstein Inequality). *Assume that $\{x_i\}_{i=1}^n$ is a sequence of random vectors in $\mathbb{R}^d$, and it forms a martingale difference sequence with respect to $(\mathcal{F}_t)$ (i.e. $\mathbb{E}\left[x_t | \mathcal{F}_{t-1}\right] = 0$ and $x_t$ is $\mathcal{F}_t$-measurable). If $\mathbb{E}\left[\|x_t\|^2 | \mathcal{F}_{t-1}\right] \leq \sigma^2$ and $\|x_t\| \leq M$ a.s., then with probability at least $1 - \delta$,*

$$\left\|\sum_{i=1}^n x^i\right\| \leq 2\sigma\sqrt{n\log\left(\frac{d+1}{\delta}\right)} + 2M\log\left(\frac{d+1}{\delta}\right).$$

*When the $\ell_2$ norm is replaced by the $\ell_\infty$ norm, i.e., $\{x_i\}_{i=1}^n$ satisfies $\mathbb{E}\left[\|x_t\|_\infty^2 | \mathcal{F}_{t-1}\right] \leq \sigma^2$ and $\|x_t\|_\infty \leq M$,*

$$\left\|\sum_{i=1}^n x^i\right\|_\infty \leq 2\sigma\sqrt{n\log\left(\frac{2d}{\delta}\right)} + 2M\log\left(\frac{2d}{\delta}\right)$$

*holds with probability at least $1 - \delta$.*

To prove Proposition 4.3, we consider $\hat{\mu}_0(s, a) = \frac{N(s,a)}{N_e}$, then it is clear that $\hat{\mu}(s, a) = \max(\hat{\mu}_0(s, a), \varsigma)$. Now, according to the Bernstein's inequality, we construct the "failure event"

$$\Omega := \bigcup_{s,a}\left\{|\mu(s, a) - \hat{\mu}_0(s, a)| > \sqrt{\mu(s, a)\frac{\ell}{N_e}} + \frac{\ell}{N_e}\right\},$$

where $\ell \geq 4\log\left(\frac{6|\mathcal{S}||\mathcal{A}|}{\delta}\right)$ is a mild logarithmic term. We next prove the three properties listed in Proposition 4.3 one by one.

**Proof of Proposition 4.3 (1).** In fact, we only need to show that $\mathbb{P}(\Omega) \leq \frac{\delta}{3}$, and the event $\Omega^c$ implies that $\mu(s, a) \leq 2\hat{\mu}(s, a), \forall s, a$, as long as our choice of batch size satisfies $N_e \geq \frac{128\mathcal{N}\psi\ell}{\varphi^2(1-\gamma)^4\epsilon_e^2} \geq \frac{32\ell\mathcal{N}\psi}{\varphi(1-\gamma)^2\epsilon_e} = \frac{32\ell}{\varsigma}$.

By Bernstein's inequality, it holds that

$$\mathbb{P}\left(|\mu(s, a) - \hat{\mu}_0(s, a)| > \sqrt{\mu(s, a)\frac{\ell}{N_e}} + \frac{\ell}{N_e}\right) \leq \frac{\delta}{3|\mathcal{S}||\mathcal{A}|}, \quad \forall (s, a) \in \mathcal{S} \times \mathcal{A}.$$

Then $\mathbb{P}(\Omega) \leq \frac{\delta}{3}$ follows directly from the union bound. Conditioning on $\Omega^c$, we have

$$|\mu(s, a) - \hat{\mu}_0(s, a)| \leq \sqrt{\mu(s, a)\frac{\ell}{N_e}} + \frac{\ell}{N_e} \leq \sqrt{\mu(s, a)\frac{\varsigma}{32}} + \frac{\varsigma}{32} \leq \frac{\mu(s, a)}{4} + \frac{\varsigma}{16}. \tag{19}$$

Hence, it holds that

$$\mu(s,a) \le \frac{4}{3}\hat{\mu}_0(s,a) + \frac{\varsigma}{12} \le \frac{3}{2}\max(\hat{\mu}_0(s,a),\varsigma) \le 2\hat{\mu}(s,a).$$

From now on, the argument is all conditioning on $\Omega^c$.

**Proof of Proposition 4.3 (2).** Given a $\pi \in \Pi(\psi)$, we have to prove that $W^{-1}\nu^\pi \in \mathcal{X}$.

Let $\nu = \nu^\pi$, $x = W^{-1}\nu$. Then due to $\pi \in \Pi(\psi)$, we have

$$\max_{s,a}\frac{x(s,a)}{\hat{\mu}(s,a)} = \max_{s,a}\frac{\nu(s,a)}{\mu(s,a)} \le \frac{\psi}{1-\gamma},$$

$$\sum_{s,a}\frac{x(s,a)}{\hat{\mu}(s,a)} = \sum_{s,a}\frac{\nu(s,a)}{\mu(s,a)} \le \frac{\mathcal{N}\psi}{1-\gamma}.$$

Now it remains to show $\sum_{s,a} x(s,a) \le \frac{4}{1-\gamma}$. Note that (19) also implies

$$\hat{\mu}_0(s,a) \le \frac{5}{4}\mu(s,a) + \frac{\varsigma}{16}.$$

Hence if $\mu(s,a) \le \frac{1}{2}\hat{\mu}(s,a)$, then it must hold that $\hat{\mu}_0(s,a) < \hat{\mu}(s,a) \Rightarrow \hat{\mu}_0(s,a) < \varsigma, \hat{\mu}(s,a) = \varsigma$. We define $\mathfrak{S} := \{(s,a) \in \mathcal{S} \times \mathcal{A} : \hat{\mu}(s,a) = \varsigma\}$, then for $(s,a) \notin \mathfrak{S}$, it holds that $\mu(s,a) \ge \frac{1}{2}\hat{\mu}(s,a)$. Thus, we have

$$\sum_{s,a} x(s,a) = \sum_{(s,a)\in\mathfrak{S}} \hat{\mu}(s,a)\frac{\nu(s,a)}{\mu(s,a)} + \sum_{(s,a)\notin\mathfrak{S}} \frac{\hat{\mu}(s,a)}{\mu(s,a)}\nu(s,a)$$

$$\le \varsigma\frac{\mathcal{N}\psi}{1-\gamma} + \sum_{(s,a)\notin\mathfrak{S}} 2\nu(s,a)$$

$$\le \frac{3}{1-\gamma}.$$

The last inequality holds as long as $\varsigma \le \frac{1}{\mathcal{N}\psi}$.

**Proof of Proposition 4.3 (3).** We decompose the quantity $\|Wx - x\|_1$ as

$$\|Wx - x\|_1 = \sum_{s,a} |\mu(s,a) - \hat{\mu}(s,a)|\frac{x(s,a)}{\hat{\mu}(s,a)}$$

$$= \sum_{(s,a)\in\mathfrak{S}} |\mu(s,a) - \hat{\mu}(s,a)|\frac{x(s,a)}{\hat{\mu}(s,a)} + \sum_{(s,a)\notin\mathfrak{S}} |\mu(s,a) - \hat{\mu}(s,a)|\frac{x(s,a)}{\hat{\mu}(s,a)}.$$

From our definition of $\mathfrak{S}$, we see if $(s,a) \in \mathfrak{S}$, then $\hat{\mu}(s,a) = \varsigma \ge \hat{\mu}_0(s,a)$, and from (19) we have $\mu(s,a) \le 2\varsigma \Rightarrow |\mu(s,a) - \hat{\mu}(s,a)| \le \varsigma$. Thus, the first part can be bounded as

$$\sum_{(s,a)\in\mathfrak{S}} |\mu(s,a) - \hat{\mu}(s,a)|\frac{x(s,a)}{\hat{\mu}(s,a)} \le \sum_{s,a} \varsigma\frac{x(s,a)}{\hat{\mu}(s,a)} \le \varsigma\frac{\mathcal{N}\psi}{1-\gamma}.$$

As for the second part, we have

$$\sum_{(s,a)\notin\mathfrak{S}} |\mu(s,a) - \hat{\mu}(s,a)|\frac{x(s,a)}{\hat{\mu}(s,a)}$$

$$= \sum_{(s,a)\notin\mathfrak{S}} |\mu(s,a) - \hat{\mu}_0(s,a)|\frac{x(s,a)}{\hat{\mu}(s,a)}$$

$$\le \sum_{(s,a)\notin\mathfrak{S}} \left(\sqrt{\mu(s,a)\frac{\ell}{N_e}} + \frac{\ell}{N_e}\right)\frac{x(s,a)}{\hat{\mu}(s,a)}$$

$$= \sqrt{\frac{\ell}{N_e}}\sum_{(s,a)\notin\mathfrak{S}} \sqrt{\frac{\mu(s,a)}{\hat{\mu}(s,a)}}\sqrt{x(s,a)\cdot\frac{x(s,a)}{\hat{\mu}(s,a)}} + \frac{\ell}{N_e}\sum_{(s,a)\notin\mathfrak{S}} \frac{x(s,a)}{\hat{\mu}(s,a)}$$

$$\overset{(a)}{\leq} \sqrt{\frac{2\ell}{N_e}} \sum_{s,a} \sqrt{x(s,a) \cdot \frac{x(s,a)}{\hat{\mu}(s,a)}} + \frac{\ell}{N_e} \sum_{s,a} \frac{x(s,a)}{\hat{\mu}(s,a)}$$

$$\overset{(b)}{\leq} \sqrt{\frac{2\ell}{N_e}} \sqrt{\sum_{s,a} x(s,a) \sum_{s,a} \frac{x(s,a)}{\hat{\mu}(s,a)}} + \frac{\ell}{N_e} \sum_{s,a} \frac{x(s,a)}{\hat{\mu}(s,a)}$$

$$\overset{(c)}{\leq} \frac{2}{1-\gamma} \sqrt{\frac{2\mathcal{N}\psi\ell}{N_e}} + \frac{\mathcal{N}\psi\ell}{(1-\gamma)N_e},$$

where the inequality (a) comes from the fact $\frac{\mu(s,a)}{\hat{\mu}(s,a)} \leq 2$; (b) is due to Cauchy's inequality, and (c) is due to $\sum_{s,a} \frac{x(s,a)}{\hat{\mu}(s,a)} \leq \frac{\mathcal{N}\psi}{1-\gamma}$ and $\sum_{s,a} x(s,a) \leq \frac{4}{1-\gamma}$. Therefore, because we set $\varsigma = \frac{\varphi(1-\gamma)^2\epsilon_e}{2\mathcal{N}\psi}$, and $N_e \geq \frac{128\mathcal{N}\psi\ell}{\varphi^2(1-\gamma)^4\epsilon_e^2}$, we have $\|Wx - x\|_1 \leq \varphi(1-\gamma)\epsilon_e, \forall x \in \mathcal{X}$.

## C   The magnitude and variance of the gradient estimators

**Proposition C.1.** *For any sample $\zeta \sim \rho_0 \times \mathcal{D}$, and any feasible solution $Z = [V; \lambda; x]$, the stochastic gradient estimators constructed in* (9) *are unbiased, and they satisfy the following bounds:*[2]

$$\begin{cases} \mathbb{E}[\widehat{g}_V(Z;\zeta)] = \nabla_V\mathcal{L}_w(Z) \\ \|\widehat{g}_V(Z;\zeta)\| \leq \mathcal{O}(\frac{\psi}{1-\gamma}) \\ \mathbb{E}[\|\widehat{g}_V(Z;\zeta)\|^2] \leq \mathcal{O}(\frac{\psi}{(1-\gamma)^2}) \end{cases} \begin{cases} \mathbb{E}[\widehat{g}_\lambda(Z;\zeta)] = \nabla_\lambda\mathcal{L}_w(Z) \\ \|\widehat{g}_\lambda(Z;\zeta)\|_\infty \leq \mathcal{O}(\frac{\psi}{1-\gamma}) \\ \mathbb{E}[\|\widehat{g}_\lambda(Z;\zeta)\|_\infty^2] \leq \mathcal{O}(\frac{\psi}{(1-\gamma)^2}) \end{cases} \begin{cases} \mathbb{E}[\widehat{g}_x(Z;\zeta)] = \nabla_x\mathcal{L}_w(Z) \\ \|\widehat{g}_x(Z;\zeta)\|_{x'}^2 \leq \mathcal{O}(\frac{\psi^2\mathcal{N}}{\varphi^3(1-\gamma)^5\epsilon_e}) \\ \mathbb{E}[\|\widehat{g}_x(Z;\zeta)\|_{x'}^2] \leq \mathcal{O}(\frac{\mathcal{N}\psi}{\varphi^2(1-\gamma)^3}) \end{cases}$$

*where $x' \in \mathcal{X}$ is an arbitrary vector.*

For any sample $\zeta = (s_0, s, a, s', r, \mathbf{u}) \sim \rho_0 \times \mathcal{D}$, it is not hard to see that the estimators constructed in (9) are unbiased. Next, we provide the bound on the norm and variance of these estimators.

For the estimator $\widehat{g}_V(Z;\zeta) := \mathbb{I}_{s_0} + \frac{x(s,a)}{\hat{\mu}(s,a)}(\gamma\mathbb{I}_{s'} - \mathbb{I}_s)$, we have

$$\|\widehat{g}_V(Z;\zeta)\| \leq 1 + \frac{x(s,a)}{\hat{\mu}(s,a)}(1+\gamma) \overset{(a)}{\leq} 1 + \frac{2\psi}{1-\gamma},$$

$$\begin{aligned} \mathbb{E}[\|g_V(Z;\zeta)\|^2] &\leq \sum_{s,a} \mu(s,a) \cdot 2\left(1 + 4 \cdot \frac{x(s,a)^2}{\hat{\mu}(s,a)^2}\right), \\ &\leq 2 + 8 \cdot \sum_{s,a} \frac{\mu(s,a)}{\hat{\mu}(s,a)} \frac{x(s,a)}{\hat{\mu}(s,a)} x(s,a) \overset{(b)}{\leq} 2 + \frac{64\psi}{(1-\gamma)^2}. \end{aligned}$$

Here (a) is due to $x \in \mathcal{X}$, which indicates that $\frac{x(s,a)}{\hat{\mu}(s,a)} \leq \frac{\psi}{1-\gamma}$ for all $(s,a)$. The inequality (b) is due to $\frac{\mu(s,a)}{\hat{\mu}(s,a)} \leq 2$ established in Proposition 4.3, and $\sum_{s,a} x(s,a) \leq \frac{4}{1-\gamma}$.

Similarly, for the estimator $\widehat{g}_\lambda(Z;\zeta) := \frac{x(s,a)}{\hat{\mu}(s,a)}\mathbf{u}^\kappa$, we have

$$\|\widehat{g}_\lambda(Z;\zeta)\|_\infty \leq \left\|\frac{x(s,a)}{\hat{\mu}(s,a)}\mathbf{u}^\kappa\right\|_\infty \overset{(a)}{\leq} \frac{x(s,a)}{\hat{\mu}(s,a)}(1 + (1-\gamma)\kappa) \overset{(b)}{\leq} \frac{2\psi}{1-\gamma},$$

$$\begin{aligned} \mathbb{E}[\|g_\lambda(Z;\zeta)\|_\infty^2] &\leq \sum_{s,a} \mu(s,a) \cdot 4\frac{x(s,a)^2}{\hat{\mu}(s,a)^2}, \\ &= 4\sum_{s,a} \frac{\mu(s,a)}{\hat{\mu}(s,a)} \frac{x(s,a)}{\hat{\mu}(s,a)} x(s,a) \overset{(c)}{\leq} \frac{32\psi}{(1-\gamma)^2}. \end{aligned}$$

Here (a) follows from $\|\mathbf{u}^\kappa\|_\infty \leq \|\mathbf{u}\|_\infty + (1-\gamma)\kappa$, and (b) is due to $(1-\gamma)\kappa = 5\varphi\epsilon(1-\gamma) < 1$, and (c) is similar to the argument of the bound on $\mathbb{E}[\|g_V(Z;\zeta)\|^2]$.

---

[2]For vectors $u, v \in \mathbb{R}^n$, we write $\|u\|_v^2 := \sum_{i=1}^n v_i u_i^2$ for simplicity.

Finally, for the estimator $\widehat{g}_x(Z;\zeta) := \frac{r+\gamma V(s)-V(s')+\langle \mathbf{u}^\kappa,\lambda\rangle}{\hat{\mu}(s,a)}\mathbb{I}_{s,a}$, we have

$$
\begin{aligned}
\|\widehat{g}_x(Z;\zeta)\|_{x'}^2 &= \frac{x'(s,a)}{\hat{\mu}(s,a)^2}\cdot|r+\gamma V(s)-V(s')+\langle \mathbf{u}^\kappa,\lambda\rangle|^2 \\
&\leq \frac{x'(s,a)}{\hat{\mu}(s,a)^2}\left(1+\frac{16}{1-\gamma}(1+\frac{2}{\varphi})+\frac{8(1+\kappa)}{\varphi}\right)^2 \\
&\leq \frac{\psi}{(1-\gamma)\varsigma}\cdot\frac{64^2}{\varphi^2(1-\gamma)^2} \\
&= \mathcal{O}\left(\frac{\psi^2\mathcal{N}}{\varphi^3(1-\gamma)^5\epsilon_e}\right),
\end{aligned}
$$

and as long as $\zeta$ is independent of $x'\in\mathcal{X}$,

$$
\begin{aligned}
\mathbb{E}\left[\|\widehat{g}_x(Z;\zeta)\|_{x'}^2\right] &\leq \sum_{s,a}\frac{\mu(s,a)x'(s,a)}{\hat{\mu}(s,a)^2}\cdot\left(1+\frac{16}{1-\gamma}(1+\frac{2}{\varphi})+\frac{8(1+\kappa)}{\varphi}\right)^2 \\
&\leq \sum_{s,a}\frac{\mu(s,a)}{\hat{\mu}(s,a)}\frac{x'(s,a)}{\hat{\mu}(s,a)}\cdot\frac{64^2}{\varphi^2(1-\gamma)^2} \\
&\leq \mathcal{O}\left(\frac{\mathcal{N}\psi}{\varphi^2(1-\gamma)^3}\right).
\end{aligned}
$$

This completes the proof of Proposition C.1.

**A few notational definitions.** We should notice that the above bounds on the gradient estimators are notationally very complicated. Therefore, Let us conveniently write the above bounds as

$$
\begin{cases}
\|g_V(Z^t;\zeta_t)\|\leq M_V, \\
\|g_\lambda(Z^t;\zeta_t)\|_\infty\leq M_\lambda, \\
\|g_x(Z^t;\zeta_t)\|_{x'}\leq M_x\sqrt{D_{x,1}},
\end{cases}
\quad\text{and}\quad
\begin{cases}
\mathbb{E}\left[\|g_V(Z;\zeta)\|^2\right]\leq\sigma_V^2, \\
\mathbb{E}\left[\|g_\lambda(Z;\zeta)\|_\infty^2\right]\leq\sigma_\lambda^2, \\
\mathbb{E}\left[\|g_x(Z^t;\zeta_t)\|_{x'}^2\right]\leq\sigma_x^2 D_{x,1},
\end{cases}
$$

where the constants $\sigma_V,\sigma_\lambda,\sigma_x$ and $M_V,M_\lambda,M_x$ are

$$
\sigma_V^2=\Theta\left(\frac{\psi}{(1-\gamma)^2}\right), \qquad \sigma_\lambda^2=\Theta\left(\frac{\psi}{(1-\gamma)^2}\right), \qquad \sigma_x^2=\Theta\left(\frac{\mathcal{N}\psi}{\varphi^2(1-\gamma)^2}\right), \qquad (20)
$$

$$
M_V=\Theta\left(\frac{\psi}{1-\gamma}\right), \qquad M_\lambda=\Theta\left(\frac{\psi}{1-\gamma}\right), \qquad M_x=\Theta\left(\frac{\psi}{\varphi(1-\gamma)^2}\sqrt{\frac{\mathcal{N}}{\varphi\epsilon_e}}\right), \qquad (21)
$$

and $D_{x,1}$ is a suitable upper bound on the diameter of $\mathcal{X}$, namely we choose $D_{x,1}=\Theta\left(\frac{1}{1-\gamma}\right)$ such that $D_{x,1}\geq\sup_{x,x'\in\mathcal{X}}\|x'-x\|_1$. Similarly, we define $D_{\lambda,1}:=\sup_{\lambda,\lambda'\in\Lambda}\|\lambda'-\lambda\|_1=\Theta\left(\frac{1}{\varphi}\right)$.

Furthermore, we also introduce the diameters of the feasible domains w.r.t. the initial solution $V^1,\lambda^1,x^1$. Recall that the initial point of Algorithm 1 is chosen as

$$
V^1=\mathbf{0}\in\mathcal{V}, \qquad \lambda^1=\frac{\mathbf{1}}{\varphi I}\in\Lambda, \qquad x^1=\frac{c_x\hat{\mu}}{1-\gamma}\in\mathcal{X},
$$

where $c_x=\frac{\mathcal{N}}{|\mathcal{S}||\mathcal{A}|}$ ensures that $x^1\in\mathcal{X}$. Then, we can take $D_V,D_\lambda,D_x$ as

$$
D_V^2:=\sup_{V'\in\mathcal{V}}\|V'-V^1\|^2=\Theta\left(\frac{|\mathcal{S}|}{\varphi^2(1-\gamma)^2}\right),
$$

$$
D_\lambda:=\sup_{\lambda'\in\Lambda}\mathrm{KL}(\lambda'\,\|\,\lambda^1)=\Theta\left(\frac{\log I}{\varphi}\right),
$$

$$
D_x\geq\sup_{x'\in\mathcal{X}}\mathrm{KL}(x'\,\|\,x^1), \qquad D_x=\Theta\left(\frac{\log\psi}{1-\gamma}\right).
$$

**Remark C.2.** *It is worth noting that, Proposition C.1 directly implies* $\mathbb{E}\left[\left.\left\|\widehat{g}_V(Z^t;\zeta_t)\right\|^2\right| Z^t\right] \le \sigma_V^2$, $\mathbb{E}\left[\left.\left\|\widehat{g}_\lambda(Z^t;\zeta_t)\right\|_\infty^2\right| Z^t\right] \le \sigma_\lambda^2$ *and* $\mathbb{E}\left[\left.\left\|\widehat{g}_x(Z^t;\zeta_t)\right\|_{x^t}^2\right| Z^t\right] \le D_{x,1}\sigma_x^2$ *for each step t.*

**Remark C.3.** *The reason why we bound the term* $\left\|g_x(Z^t;\zeta_t)\right\|_{x^t}$ *instead of* $\left\|g_x(Z^t;\zeta_t)\right\|_\infty$ *is that,*

$$\left\|g_x(Z^t;\zeta_t)\right\|_\infty \lesssim \frac{1}{\varphi(1-\gamma)}\frac{1}{\hat{\mu}(s_t,a_t)} \le \frac{1}{\varphi(1-\gamma)\varsigma}.$$

*Thus, we have to take* $M_{x,\infty} = \Theta\left(\frac{1}{\varphi(1-\gamma)\varsigma}\right)$ *to ensure a uniformly bound as*

$$\left\|g_x(Z^t;\zeta_t)\right\|_\infty \le M_{x,\infty}. \tag{22}$$

## D  Proof of Theorem 4.4

To bound $\mathrm{Gap}(\overline{x})$, let us denote

$$(V',\lambda') = \underset{V\in\mathcal{V},\lambda\in\Lambda}{\arg\min}\ \mathcal{L}_w(V,\lambda,\overline{x}), \qquad x' = \underset{x\in\mathcal{X}}{\arg\max}\ \underset{V\in\mathcal{V},\lambda\in\Lambda}{\min}\ \mathcal{L}_w(V,\lambda,x), \tag{23}$$

and denote $Z' = [V';\lambda';x']$. It is worth mentioning that $V',\lambda'$ are random variables that depend on $\overline{x}$ while $x'$ is deterministic. For the ease of notation, we define

$$\mathcal{G}(Z) := \begin{bmatrix} +\nabla_V\mathcal{L}_w(Z) \\ +\nabla_\lambda\mathcal{L}_w(Z) \\ -\nabla_x\mathcal{L}_w(Z) \end{bmatrix} \qquad \text{and} \qquad \widehat{g}(Z;\zeta) := \begin{bmatrix} +\widehat{g}_V(Z;\zeta) \\ +\widehat{g}_\lambda(Z;\zeta) \\ -\widehat{g}_x(Z;\zeta) \end{bmatrix}.$$

Then, by the definition of $V',\lambda',x'$ and the bi-linearity of $\mathcal{L}_w(\cdot)$, we have

$$\begin{aligned}
\mathrm{Gap}(\overline{x}) &= \underset{x\in\mathcal{X}}{\max}\ \underset{V\in\mathcal{V},\lambda\in\Lambda}{\min}\ \mathcal{L}_w(V,\lambda,x) - \underset{V\in\mathcal{V},\lambda\in\Lambda}{\min}\ \mathcal{L}_w(V,\lambda,\overline{x}) \\
&= \mathcal{L}_w(\overline{V},\overline{\lambda},x') - \mathcal{L}_w(V',\lambda',\overline{x}) \\
&= \frac{1}{T}\sum_{t=1}^T \Big(\mathcal{L}_w(V^t,\lambda^t,x') - \mathcal{L}_w(V',\lambda',x^t)\Big) \\
&= \frac{1}{T}\sum_{t=1}^T \langle \mathcal{G}(Z^t), Z^t - Z'\rangle \\
&= \underbrace{\frac{1}{T}\sum_{t=1}^T \langle \widehat{g}(Z^t;\zeta_t), Z^t - Z'\rangle}_{S_1} + \underbrace{\frac{1}{T}\sum_{t=1}^T \langle \mathcal{G}(Z^t) - \widehat{g}(Z^t;\zeta_t), Z^t - Z'\rangle}_{S_2}.
\end{aligned} \tag{24}$$

Then with the estimations in Appendix C, the $S_1$ and $S_2$ terms can be bounded by

$$\begin{aligned}
S_1 &\lesssim \frac{\alpha_V D_V^2 + \alpha_\lambda D_\lambda + \alpha_x D_x}{\eta T} + \eta\left(\frac{\sigma_V^2}{\alpha_V} + \frac{\sigma_\lambda^2 D_{\lambda,1}}{\alpha_\lambda} + \frac{\sigma_x^2 D_{x,1}}{\alpha_x}\right) \\
&\quad + \frac{\eta\iota}{T}\left(\frac{M_V^2}{\alpha_V} + \frac{M_\lambda^2 D_{\lambda,1}}{\alpha_\lambda} + \frac{M_x^2 D_{x,1}}{\alpha_x}\right)
\end{aligned} \tag{25}$$

and

$$S_2 \lesssim (D_V\sigma_V + D_{\lambda,1}\sigma_\lambda + D_{x,1}\sigma_x)\sqrt{\frac{\iota}{T}} + (D_V M_V + D_{\lambda,1}M_\lambda + D_{x,1}M_x)\frac{\iota}{T} \tag{26}$$

with probability at least $1 - \delta/10$ respectively, as long as the stepsize satisfies

$$\eta \le \frac{1}{2}\min\left(\frac{\alpha_\lambda}{M_\lambda}, \frac{\alpha_x}{M_{x,\infty}}\right). \tag{27}$$

Due to the sophistication of the proof, we move the analysis of (25) and (26) to Appendix D.2 and D.3 respectively.

Finally, combining the inequalities (24), (25) and (26), and requiring that (27) holds true for $\eta = 1/\sqrt{T}$, we have with probability at least $1 - \delta/3$

$$\mathrm{Gap}(\overline{x}) \lesssim \frac{\alpha_V D_V^2 + \alpha_\lambda D_\lambda + \alpha_x D_x}{\eta T} + \eta \left( \frac{\sigma_V^2}{\alpha_V} + \frac{\sigma_\lambda^2 D_{\lambda,1}}{\alpha_\lambda} + \frac{\sigma_x^2 D_{x,1}}{\alpha_x} \right)$$

$$+ \sqrt{\frac{\iota}{T}} \left( D_V \sigma_V + D_{\lambda,1} \sigma_\lambda + D_{x,1} \sigma_x \right)$$

$$+ \frac{\iota}{T} \left( D_V M_V + D_{\lambda,1} M_\lambda + D_{x,1} M_x \right)$$

$$+ \frac{\eta \iota}{T} \left( \frac{M_V^2}{\alpha_V} + \frac{M_\lambda^2 D_{\lambda,1}}{\alpha_\lambda} + \frac{M_x^2 D_{x,1}}{\alpha_x} \right).$$

Note that the normalizing constants are chosen as $\alpha_V = \varphi \sqrt{\frac{\psi}{|\mathcal{S}|}} = \Theta\left( \frac{\sigma_V}{D_V} \right)$, $\alpha_\lambda = \frac{1}{1-\gamma} \sqrt{\frac{\psi}{\log I}} = \Theta\left( \sigma_\lambda \sqrt{\frac{D_{\lambda,1}}{D_\lambda}} \right)$, $\alpha_x = \frac{1}{\varphi(1-\gamma)} \sqrt{\frac{\mathcal{N}\psi}{\log \psi}} = \Theta\left( \sigma_x \sqrt{\frac{D_{x,1}}{D_x}} \right)$. Then (27) holds true for the stepsize $\eta = \frac{1}{\sqrt{T}}$ with $T \gtrsim \frac{\mathcal{N}\psi\iota}{\varphi^2(1-\gamma)^4 \epsilon_e^2}$, and we can plug in the values of the constants $\alpha, D, M$, then with probability at least $1 - \delta/3$ it holds that

$$\mathrm{Gap}(\overline{x}) \lesssim \sqrt{\frac{\mathcal{N}\psi\iota}{\varphi^2(1-\gamma)^4 T}} \left( 1 + \frac{\iota}{T} \cdot \frac{\psi}{\varphi(1-\gamma)^2 \epsilon_e} \right) \lesssim \sqrt{\frac{\mathcal{N}\psi\iota}{\varphi^2(1-\gamma)^4 T}}.$$

Choosing $c_o$ to ensure $\mathrm{Gap}(\overline{x}) \le \frac{\epsilon}{2}$ completes the proof of Theorem 4.4.

### D.1 A few supporting lemmas

For the proof in the following parts of Appendix D, we introduce a few supporting lemmas.

**Lemma D.1.** *Let $\{Y^k\}_{k=1}^T$ be generated by $Y^{k+1} = \mathrm{argmin}_{Y \in \mathcal{Y}} \left( \eta \langle Y - Y^k, g^k \rangle + \mathrm{KL}(Y \parallel Y^k) \right)$, where $\eta \le \frac{1}{2 \max_k \|g_k\|_\infty}$ and $\mathcal{Y}$ is some convex set. Then for all $Y' \in \mathcal{Y}$, it holds that*

$$\frac{1}{T} \sum_{t=1}^T \langle Y^t - Y', g^t \rangle \le \frac{\mathrm{KL}(Y' \parallel Y^1)}{\eta T} + \frac{4\eta}{T} \sum_{t=1}^T \|g^k\|_{Y^k}^2$$

$$\le \frac{\mathrm{KL}(Y' \parallel Y^1)}{\eta T} + \frac{4\eta D_{Y,1}}{T} \sum_{t=1}^T \|g^k\|_\infty^2.$$

*where $D_{Y,1}$ can be any upper bound of $\max_{Y \in \mathcal{Y}} \|Y\|_1$.*

The proof of Lemma D.1 is presented in Appendix D.4.

**Lemma D.2.** *Let $\{Y^k\}_{k=1}^T$ be generated by $Y^{k+1} = \mathrm{Proj}_\mathcal{Y}\left(Y^k - \eta g^k\right)$, where $\mathcal{Y}$ is some convex set. Then for all $Y' \in \mathcal{Y}$, it holds that*

$$\frac{1}{T} \sum_{t=1}^T \langle Y^t - Y', g^t \rangle \le \frac{\|Y' - Y^1\|^2}{2\eta T} + \frac{\eta}{T} \sum_{t=1}^T \|g^k\|^2.$$

The proof of Lemma D.2 is similar but a lot simpler than that of Lemma D.1, and is hence omitted.

**Proposition D.3** (Corollary of Bernstein's inequality). *For a sequence of random variables $X_1, \cdots, X_N$ adapted to $(\mathcal{F}_n)$, and $\mathbb{E}\left[|X_i| \big| \mathcal{F}_{i-1}\right] \le c$, $|X_i| \le M$, we have with probability at least $1 - \delta$,*

$$\left| \frac{1}{N} \sum_{i=1}^N X_i \right| \le 2c + 3M \frac{\log(2/\delta)}{N}.$$

*Proof.* Notice that $\mathbb{E}\left[X_i^2 \big| \mathcal{F}_{i-1}\right] \le cM$, and by Bernstein's inequality

$$\left| \frac{1}{N} \sum_{i=1}^N \left( X_i - \mathbb{E}\left[X_i \big| \mathcal{F}_{i-1}\right] \right) \right| \le \sqrt{\frac{2cM \log(2/\delta)}{N}} + 2M \frac{\log(2/\delta)}{N},$$

$$\Rightarrow \left| \sum_{i=1}^{N} X_i \right| \leq cN + \sqrt{2cMN\log(2/\delta)} + 2M\log(2/\delta),$$

holds with probability at least $1 - \delta$. By the AM-GM inequality, $\sqrt{2cMN\log(2/\delta)} \leq \frac{1}{2}cN + M\log(2/\delta)$, which completes the proof. $\qquad\square$

### D.2 Bounding the term $S_1$

First, by definition of $\widehat{g}(\cdot)$, we have

$$S_1 = \underbrace{\frac{1}{T}\sum_{t=1}^{T}\left\langle \widehat{g}_V(Z^t;\zeta_t), V^t - V'\right\rangle}_{S_{1,V}} + \underbrace{\frac{1}{T}\sum_{t=1}^{T}\left\langle \widehat{g}_\lambda(Z^t;\zeta_t), \lambda^t - \lambda'\right\rangle}_{S_{1,\lambda}} + \underbrace{\frac{1}{T}\sum_{t=1}^{T}\left\langle -\widehat{g}_x(Z^t;\zeta_t), x^t - x'\right\rangle}_{S_{1,x}}.$$

Applying Lemma D.2 with $Y^t = V^t$, $g^t = \widehat{g}_V(Z^t;\zeta_t)$ yields

$$S_{1,V} \leq \frac{\alpha_V \left\| V' - V^1 \right\|^2}{2\eta T} + \frac{\eta}{\alpha_V T}\sum_{t=1}^{T}\left\| \widehat{g}_V(Z^t;\zeta_t)\right\|^2.$$

Applying Lemma D.1 with $Y^t = \lambda^t$, $g^t = \widehat{g}_\lambda(Z^t;\zeta_t)$, we have

$$S_{1,\lambda} \leq \frac{\alpha_\lambda \,\mathrm{KL}(\lambda' \,\|\, \lambda^1)}{\eta T} + \frac{4\eta D_{\lambda,1}}{\alpha_\lambda T}\sum_{t=1}^{T}\left\| \widehat{g}_\lambda(Z^t;\zeta_t)\right\|_\infty^2,$$

as long as $\left\| \widehat{g}_\lambda(Z^t;\zeta_t)\right\|_\infty \leq \frac{\alpha_\lambda}{2\eta}$ holds for all $t$, and $\frac{1}{\eta} \geq \frac{2M_\lambda}{\alpha_\lambda}$ suffices.

Finally, applying Lemma D.1 with $Y^t = x^t$, $g^t = -\widehat{g}_x(Z^t;\zeta_t)$, we obtain

$$S_{1,x} \leq \frac{\alpha_x \,\mathrm{KL}(x' \,\|\, x^1)}{\eta T} + \frac{4\eta}{\alpha_x T}\sum_{t=1}^{T}\left\| \widehat{g}_x(Z^t;\zeta_t)\right\|_{x^t}^2,$$

as long as $\left\| \widehat{g}_x(Z^t;\zeta_t)\right\|_\infty \leq \frac{\alpha_x}{2\eta}$ holds for all $t$, and $\frac{1}{\eta} \geq \frac{2M_{x,\infty}}{\alpha_x}$ suffices.

Combining all the estimations above, as long as the stepsize $\eta$ satisfies (27), we have

$$\begin{aligned}
S_1 \leq & \frac{\alpha_V \left\| V' - V^1 \right\|^2 + \alpha_\lambda \,\mathrm{KL}(\lambda' \,\|\, \lambda^1) + \alpha_x \,\mathrm{KL}(x' \,\|\, x^1)}{\eta T} \\
& + \frac{4\eta}{T}\sum_{t=1}^{T}\left( \frac{\left\| \widehat{g}_V(Z^t;\zeta_t)\right\|^2}{\alpha_V} + \frac{D_{\lambda,1}\left\| \widehat{g}_\lambda(Z^t;\zeta_t)\right\|_\infty^2}{\alpha_\lambda} + \frac{\left\| \widehat{g}_x(Z^t;\zeta_t)\right\|_{x^t}^2}{\alpha_x}\right).
\end{aligned} \tag{28}$$

For the second term of $S_1$ in (28), with the variance and magnitude bounds provided in Proposition C.1, applying Proposition D.3 to the sequences $\{\|\widehat{g}_V(Z^t;\zeta_t)\|^2\}_{t=1}^T$, $\{\|\widehat{g}_\lambda(Z^t;\zeta_t)\|_\infty^2\}_{t=1}^T$ and $\{\|\widehat{g}_x(Z^t;\zeta_t)\|_{x^t}^2\}_{t=1}^T$ proves the inequality (25) with probability at least $1 - \delta/10$.

### D.3 Bounding the term $S_2$

For the term $S_2$, we introduce the martingale difference sequences

$$\begin{aligned}
\Delta_V^t &:= \widehat{g}_V(Z^t;\zeta_t) - \nabla_V \mathcal{L}_w(V^t, \lambda^t, x^t), \\
\Delta_\lambda^t &:= \widehat{g}_\lambda(Z^t;\zeta_t) - \nabla_\lambda \mathcal{L}_w(V^t, \lambda^t, x^t), \\
\Delta_x^t &:= \widehat{g}_x(Z^t;\zeta_t) - \nabla_x \mathcal{L}_w(V^t, \lambda^t, x^t),
\end{aligned}$$

Then $S_2$ can be decomposed as

$$S_2 = \underbrace{\frac{1}{T} \sum_{t=1}^{T} \left( \langle \Delta_V^t, V' - V^1 \rangle + \langle \Delta_\lambda^t, \lambda' - \lambda^1 \rangle \right)}_{S_{2,c}}$$

$$+ \underbrace{\frac{1}{T} \sum_{t=1}^{T} \left( \langle \Delta_V^t, V^1 - V^t \rangle + \langle \Delta_\lambda^t, \lambda^1 - \lambda^t \rangle + \langle -\Delta_x^t, x' - x^t \rangle \right)}_{S_{2,m}}.$$

Note that the martingale part $S_{2,m}$ has expectation zero. However, for the first part, $V'$ and $\lambda'$ are random variables depending on $\bar{x}$. Thus the correlated part $S_{2,c}$ may not have zero mean.

**Bounding the term $S_{2,c}$** For the correlated part $S_{2,c}$, the sequence $\Delta_V^t$ and $\Delta_\lambda^t$ are (vector-valued) martingale difference sequences, and hence

$$S_{2,c} = \left\langle \frac{1}{T} \sum_{t=1}^{T} \Delta_V^t, V' - V^1 \right\rangle + \left\langle \frac{1}{T} \sum_{t=1}^{T} \Delta_\lambda^t, \lambda' - \lambda^1 \right\rangle$$

$$\leq \|V' - V^1\| \cdot \frac{1}{T} \left\| \sum_{t=1}^{T} \Delta_V^t \right\| + \|\lambda' - \lambda^1\|_1 \cdot \frac{1}{T} \left\| \sum_{t=1}^{T} \Delta_\lambda^t \right\|_\infty.$$

The quantity $\left\| \sum_{t=1}^{T} \Delta_V^t \right\|$ and $\left\| \sum_{t=1}^{T} \Delta_\lambda^t \right\|_\infty$ both can be bounded by applying Lemma B.1. More specifically, with probability at least $1 - \delta/20$, it holds that

$$\left\| \frac{1}{T} \sum_{t=1}^{T} \Delta_V^t \right\| \lesssim \sigma_V \sqrt{\frac{\log(|\mathcal{S}|/\delta)}{T}} + M_V \frac{\log(|\mathcal{S}|/\delta)}{T},$$

$$\left\| \frac{1}{T} \sum_{t=1}^{T} \Delta_\lambda^t \right\|_\infty \lesssim \sigma_\lambda \sqrt{\frac{\log(I/\delta)}{T}} + M_\lambda \frac{\log(I/\delta)}{T}.$$

Therefore, we have

$$S_{2,c} \lesssim (D_V \sigma_V + D_{\lambda,1} \sigma_\lambda) \sqrt{\frac{\iota}{T}} + (D_V M_V + D_{\lambda,1} M_\lambda) \frac{\iota}{T}.$$

**Bounding the term $S_{2,m}$** In order to bound the martingale part $S_{2,m}$, we have to consider martingales difference sequences[3] $\overline{\Delta}_V^t := \langle \Delta_V^t, V^1 - V^t \rangle, \overline{\Delta}_\lambda^t := \langle \Delta_\lambda^t, \lambda^1 - \lambda^t \rangle, \overline{\Delta}_x^t := \langle \Delta_x^t, x^t - x' \rangle$. We estimate the variance and magnitude as

$$\left| \overline{\Delta}_V^t \right| \leq 2 D_V M_V, \quad \mathbb{E}\left[ \left( \overline{\Delta}_V^t \right)^2 \Big| \mathcal{F}_t \right] \leq \mathbb{E}\left[ \|V^1 - V'\|^2 \|\Delta_V^t\|^2 \Big| \mathcal{F}_t \right] \leq D_V^2 \sigma_V^2,$$

$$\left| \overline{\Delta}_\lambda^t \right| \leq 2 D_{\lambda,1} M_\lambda, \quad \mathbb{E}\left[ \left( \overline{\Delta}_\lambda^t \right)^2 \Big| \mathcal{F}_t \right] \leq \mathbb{E}\left[ \|\lambda^1 - \lambda^t\|_1^2 \|\Delta_\lambda^t\|_\infty^2 \Big| \mathcal{F}_t \right] \leq D_{\lambda,1}^2 \sigma_\lambda^2,$$

$$\left| \overline{\Delta}_x^t \right| \leq 2 D_{x,1} M_x, \quad \mathbb{E}\left[ \left( \overline{\Delta}_x^t \right)^2 \Big| \mathcal{F}_t \right] \leq \mathbb{E}\left[ \left\| \frac{x' - x^t}{\sqrt{x' + x^t}} \right\|^2 \|\Delta_x^t\|_{x'+x^t}^2 \Big| \mathcal{F}_t \right] \leq 2 D_{x,1}^2 \sigma_x^2.$$

Thus, by the Bernstein's Inequality, the following holds with probability at least $1 - \delta/20$:

$$\frac{1}{T} \sum_{t=1}^{T} \overline{\Delta}_V^t \lesssim D_V \sigma_V \sqrt{\frac{\log(1/\delta)}{T}} + \frac{D_V M_V \log(1/\delta)}{T},$$

$$\frac{1}{T} \sum_{t=1}^{T} \overline{\Delta}_\lambda^t \lesssim D_{\lambda,1} \sigma_\lambda \sqrt{\frac{\log(1/\delta)}{T}} + \frac{D_{\lambda,1} M_\lambda \log(1/\delta)}{T},$$

$$\frac{1}{T} \sum_{t=1}^{T} \overline{\Delta}_x^t \lesssim D_{x,1} \sigma_x \sqrt{\frac{\log(1/\delta)}{T}} + \frac{D_{x,1} M_x \log(1/\delta)}{T}.$$

---

[3]They are martingale difference sequences w.r.t. the filtration $(\mathcal{F}_t)$ defined by $\mathcal{F}_t = \sigma(\zeta_1, \cdots, \zeta_{t-1})$.

Therefore, with probability at least $1 - \delta/20$,

$$S_{2,m} \lesssim (D_V \sigma_V + D_{\lambda,1}\sigma_\lambda + D_{x,1}\sigma_x)\sqrt{\frac{\iota}{T}} + (D_V M_V + D_{\lambda,1}M_\lambda + D_{x,1}M_x)\frac{\iota}{T}.$$

**Bounding the term $S_2$** Finally, combining the bounds on $S_{2,m}$ and $S_{2,c}$ proves the inequality (26).

## D.4 Basics of mirror descent

Before we provide the proof of Lemma D.1, we state a basic property of the mirror descent (see e.g. [6]).

**Lemma D.4.** *Under the same assumption in Lemma D.1, it holds that for any $Y' \in \mathcal{Y}$,*

$$\eta \left\langle Y^{k+1} - Y', g^k \right\rangle \le \mathrm{KL}(Y' \parallel Y^k) - \mathrm{KL}(Y' \parallel Y^{k+1}) - \mathrm{KL}(Y^{k+1} \parallel Y^k).$$

*In particular,*

$$\mathrm{KL}(Y^k \parallel Y^{k+1}) + \mathrm{KL}(Y^{k+1} \parallel Y^k) \le \eta \left\langle Y^k - Y^{k+1}, g^k \right\rangle.$$

*Proof of Lemma D.1.* By the fact that $(x-y)\log\frac{x}{y} \ge \frac{(x-y)^2}{\max(x,y)}$, we have

$$\mathrm{KL}(Y^k \parallel Y^{k+1}) + \mathrm{KL}(Y^{k+1} \parallel Y^k) = \left\langle Y^k - Y^{k+1}, \log Y^k - \log Y^{k+1} \right\rangle \ge \sum_i \frac{(Y_i^k - Y_i^{k+1})^2}{\max(Y_i^k, Y_i^{k+1})}.$$

Together with Lemma D.4, the estimation above yields

$$\left\| \frac{Y^k - Y^{k+1}}{\sqrt{Y^k + Y^{k+1}}} \right\|^2 \le \mathrm{KL}(Y^k \parallel Y^{k+1}) + \mathrm{KL}(Y^{k+1} \parallel Y^k) \le \eta \left\langle Y^k - Y^{k+1}, g^k \right\rangle.$$

By Cauchy inequality, $\left\langle Y^k - Y^{k+1}, g^k \right\rangle \le \left\| \frac{Y^k - Y^{k+1}}{\sqrt{Y^k + Y^{k+1}}} \right\| \left\| g^k \sqrt{Y^k + Y^{k+1}} \right\|$, and hence

$$\left\| \frac{Y^k - Y^{k+1}}{\sqrt{Y^k + Y^{k+1}}} \right\| \le \eta \left\| g^k \sqrt{Y^k + Y^{k+1}} \right\| = \eta \left\| g^k \right\|_{Y^k + Y^{k+1}},$$

$$\left\langle Y^k - Y^{k+1}, g^k \right\rangle \le \left\| \frac{Y^k - Y^{k+1}}{\sqrt{Y^k + Y^{k+1}}} \right\| \left\| g^k \sqrt{Y^k + Y^{k+1}} \right\| \le \eta \left\| g^k \right\|_{Y^k + Y^{k+1}}^2.$$

To further bound $\left\| g^k \right\|_{Y^k + Y^{k+1}}$ in terms of $\left\| g^k \right\|_{Y^k}$, we estimate it as

$$\begin{aligned}
\left\| g^k \right\|_{Y^k + Y^{k+1}}^2 &= \sum_i (Y_i^k + Y_i^{k+1})(g_i^k)^2 \\
&\le 2 \left\| g^k \right\|_{Y^k}^2 + \sum_i \left| Y_i^{k+1} - Y_i^k \right| (g_i^k)^2 \\
&\le 2 \left\| g^k \right\|_{Y^k}^2 + \max_i \left| g_i^k \right| \left\| \frac{Y^k - Y^{k+1}}{\sqrt{Y^k + Y^{k+1}}} \right\| \left\| g^k \right\|_{Y^k + Y^{k+1}} \\
&\le 2 \left\| g^k \right\|_{Y^k}^2 + \eta \left\| g^k \right\|_\infty \left\| g^k \right\|_{Y^k + Y^{k+1}}^2.
\end{aligned}$$

Thus, as long as $\eta \le \frac{1}{2\|g^k\|_\infty}$, it holds that $\left\| g^k \right\|_{Y^k + Y^{k+1}} \le 2 \left\| g^k \right\|_{Y^k}$. Therefore, for all $Y' \in \mathcal{Y}$,

$$\begin{aligned}
\left\langle Y^k - Y', g^k \right\rangle &\le \frac{1}{\eta} \left[ \mathrm{KL}(Y' \parallel Y^k) - \mathrm{KL}(Y' \parallel Y^{k+1}) \right] + \left\langle Y^k - Y^{k+1}, g^k \right\rangle \\
&\le \frac{1}{\eta} \left[ \mathrm{KL}(Y' \parallel Y^k) - \mathrm{KL}(Y' \parallel Y^{k+1}) \right] + 4\eta \left\| g^k \right\|_{Y^k}^2.
\end{aligned}$$

Summing over $k = 1, \cdots, T$ completes the proof. $\qquad\square$

**Corollary D.5.** *Under the same assumption in Lemma D.1, it holds that for each $k$,*

$$\left\| \frac{Y^k - Y^{k+1}}{\sqrt{Y^k + Y^{k+1}}} \right\| \le 2\eta \left\| g^k \right\|_{Y^k},$$

$$\left\| Y^{k+1} - Y^k \right\|_1 \le 4\eta \sqrt{D_{Y,1}} \left\| g^k \right\|_{Y^k} \le 4\eta D_{Y,1} \left\| g^k \right\|_\infty.$$

*Proof.* From the proof of Lemma D.1 above, we see

$$J(Y^k, Y^{k+1}) = \mathrm{KL}(Y^k \parallel Y^{k+1}) + \mathrm{KL}(Y^{k+1} \parallel Y^k) \le \eta \left\langle Y^k - Y^{k+1}, g^k \right\rangle \le 4\eta^2 \left\| g^k \right\|_{Y^k}^2.$$

Then by Lemma D.6 we have

$$\left\| Y^{k+1} - Y^k \right\|_1 \le \left( \sqrt{\|Y^k\|_1} + \sqrt{\|Y^{k+1}\|_1} \right) \sqrt{J(Y^k, Y^{k+1})} \le 4\eta \sqrt{D_{Y,1}} \left\| g^k \right\|_{Y^k}. \qquad \square$$

**Lemma D.6** (Generalized Pinsker's Inequality). *For $y, y' \in \mathbb{R}_{>0}^n$, we consider the generalized Jeffery divergence between them:*

$$J(y, y') := \mathrm{KL}(y \parallel y') + \mathrm{KL}(y' \parallel y) = \sum_i (y_i - y_i') \log \frac{y_i}{y_i'}.$$

*Then it holds that*

$$\|y - y'\|_1 \le \left( \sqrt{\|y\|_1} + \sqrt{\|y'\|_1} \right) \sqrt{J(y, y')}.$$

*Proof.* Denote $J = J(y, y')$, $Y = \|y\|_1$, $Y' = \|y'\|_1$. We consider two (normalized) distributions $\overline{y} = \frac{y}{Y}$ and $\overline{y}' = \frac{y'}{Y'}$, then

$$
\begin{aligned}
J(y, y') &= \sum_i (y_i - y_i') \log \frac{y_i}{y_i'} \\
&= \sum_i \left( Y\overline{y}_i - Y'\overline{y}_i' \right) \left( \log \frac{\overline{y}_i}{\overline{y}_i'} + \log \frac{Y}{Y'} \right) \\
&= Y \, \mathrm{KL}(\overline{y} \parallel \overline{y}') + Y' \, \mathrm{KL}(\overline{y}' \parallel \overline{y}) + (Y - Y') \log \frac{Y}{Y'} \\
&\ge (Y + Y') \cdot \frac{1}{2} \|\overline{y} - \overline{y}'\|_1^2 + \frac{|Y - Y'|^2}{\max(Y, Y')},
\end{aligned}
$$

where the last inequality is due to Pinsker's inequality and the fact $(x - y) \log \frac{x}{y} \ge \frac{(x-y)^2}{\max(x,y)}$. Therefore, w.l.o.g. $Y < Y'$, then $|Y - Y'| \le \sqrt{Y'J}$, and

$$\sqrt{\frac{2J}{Y + Y'}} \ge \|\overline{y} - \overline{y}'\|_1 = \left\| \frac{y}{Y} - \frac{y'}{Y'} \right\|_1 = \left\| \frac{y - y'}{Y} + \frac{y'}{Y'} \left( \frac{Y'}{Y} - 1 \right) \right\|_1.$$

Hence, we have

$$
\begin{aligned}
\|y - y'\|_1 &\le \left\| \frac{y'}{Y'} (Y' - Y) \right\|_1 + Y \sqrt{\frac{J}{Y + Y'}} \\
&= |Y' - Y| + Y \sqrt{\frac{J}{Y + Y'}} \\
&\le \sqrt{Y'J} + \sqrt{YJ}. \qquad \square
\end{aligned}
$$

# E   Proof of Theorem 4.1

In this section, we provide the proof of Theorem 4.1 and Remark 4.2. We should notice that if $\psi \ge C^*$, then $\epsilon_{\mathrm{approx}}(\psi)$ reduces to 0, and the result in Remark 4.2 actually agrees with Theorem 4.1. Thus we handle them simultaneously. The key to the analysis is controlling the reward sub-optimality gap and the constraint violation in terms of the duality gap $\mathrm{Gap}(\overline{x})$ that is bounded in Theorem 4.4. Before presenting the proof, let us introduce a few notations and lemmas.

### E.1 Notations and supporting lemmas

In this proof, we will view $\nu$ as vectors in $\mathbb{R}^{|\mathcal{S}||\mathcal{A}|}$, and we define a matrix $A$ as

$$A := \left[ \mathbb{1}_{\{s'=s\}} - \gamma \mathbb{P}\left(s'|s,a\right) \right]_{(s,a),s'} \in \mathbb{R}^{|\mathcal{S}||\mathcal{A}| \times |\mathcal{S}|}. \tag{29}$$

Given the matrix $A$, we conveniently write $\sum_a (\mathbf{I} - \gamma \mathbb{P}_a)\nu_a$ as $A^\top \nu$. For the reweighted saddle point problem (7), one can easily partially minimize over $V$ and $\lambda$ since their domains are simple normal balls. Therefore, we define

$$\mathcal{J}_\kappa(x) := \min_{V \in \mathcal{V}, \lambda \in \Lambda} \mathcal{L}_w(V,\lambda,x) = r^T W x - R_\mathcal{V} \left\| A^\top W x - \rho_0 \right\|_1 - R_\Lambda \left\| [U_\kappa W x]_- \right\|_\infty, \tag{30}$$

where we denote $R_\mathcal{V} = \frac{8}{1-\gamma}\left(1 + \frac{2}{\varphi}\right), R_\Lambda = \frac{8}{\varphi}$. We also define

$$j(\psi) := \min_{V \in \mathcal{V}, \lambda \in \Lambda} \mathcal{L}_w(V,\lambda,x) = \max_{x \in \mathcal{X}} \mathcal{J}_\kappa(x) \tag{31}$$

as the optimal value of problem (7). Then $j(\psi)$ has an implicit dependence on $\kappa$ due to the term $\left\| [U_\kappa W x]_- \right\|_\infty$. In particular, we will write $j_0(\psi)$ for the case where $\kappa = 0$. Finally, we define $\pi_\kappa^*$ as the optimal policy with $\kappa$ conservative constraints. That is,

$$\pi_\kappa^* = \arg\max_\pi J(\pi) \text{ s.t. } J_i^u(\pi) \geq \kappa, \ \forall i \in [I].$$

Then the following lemmas hold true.

**Lemma E.1.** *Let $\pi^*$ be the optimal policy, and let $\pi_\kappa^*$ be defined above, then it holds that*

$$J(\pi^*) \geq J(\pi_\kappa^*) \geq J(\pi^*) - \frac{2\kappa}{\varphi}.$$

*Proof.* The inequality $J(\pi^*) \geq J(\pi_\kappa^*)$ follows from definition. For the other inequality, we fix a "baseline" policy $\tilde{\pi}$ satisfying the Slater's condition, namely $J_i^u(\tilde{\pi}) \geq \frac{\varphi}{1-\gamma}$. Let $s = \frac{(1-\gamma)\kappa}{\varphi}$, we interpolate $\nu_s := s\nu^{\pi^*} + (1-s)\nu^{\tilde{\pi}}$. $\nu_s$ is still an occupancy measure such that $\langle u_i, \nu_s \rangle \geq s\langle u_i, \nu^{\tilde{\pi}} \rangle \geq \kappa$ for $\forall i \in [I]$, and

$$\langle r, \nu_s \rangle = \left\langle r, \nu^{\pi^*} \right\rangle - s(\left\langle r, \nu^{\pi^*} \right\rangle - \langle r, \nu^{\tilde{\pi}} \rangle) \geq \left\langle r, \nu^{\pi^*} \right\rangle - \frac{2s}{1-\gamma} = J(\pi^*) - \frac{2\kappa}{\varphi}.$$

We complete the proof by noticing $J(\pi_\kappa^*) \geq \langle r, \nu_s \rangle$. $\qquad\square$

The next lemma discusses the property of $j(\cdot)$.

**Lemma E.2.** *Suppose the policy class $\Pi(\psi)$ satisfies Slater's condition, then it holds that*

$$j(\psi) \geq \max_{\pi \in \Pi(\psi) \cap \mathfrak{S}} J(\pi) - \frac{2\kappa}{\varphi} = J(\pi^*) - \epsilon_{\mathrm{approx}}(\psi) - \frac{2\kappa}{\varphi}.$$

*Proof.* Similar to the proof of Lemma E.1, we fix a $\hat{\pi} = \arg\max_{\pi \in \Pi(\psi) \cap \mathfrak{S}} J(\pi)$ and a "baseline" policy $\tilde{\pi} \in \Pi(\psi)$ satisfying the Slater's condition. Let $\hat{\nu} := \nu^{\hat{\pi}}$ and $\tilde{\nu} := \nu^{\tilde{\pi}}$ be the corresponding occupancy measures. Let $s = \frac{(1-\gamma)\kappa}{\varphi}$, then $\nu_s := s\tilde{\nu} + (1-s)\hat{\nu}$ is still an occupancy measure for which the corresponding policy belongs to $\Pi(\psi)$. For $i \in [I]$, $\langle u_i, \nu_s \rangle \geq s\langle u_i, \tilde{\nu} \rangle \geq \kappa$, and

$$\langle r, \nu_s \rangle = \langle r, \hat{\nu} \rangle - s(\langle r, \hat{\nu} \rangle - \langle r, \tilde{\nu} \rangle) \geq \langle r, \hat{\nu} \rangle - \frac{2s}{1-\gamma} = \langle r, \tilde{\nu} \rangle - \frac{2\kappa}{\varphi}.$$

Now $W^{-1}\nu_s \in \mathcal{X}$ by Proposition 4.3, and

$$j(\psi) \geq \mathcal{J}_\kappa(W^{-1}\nu_s) = \langle r, \nu_s \rangle \geq \langle r, \hat{\nu} \rangle - \frac{2\kappa}{\varphi} = \max_{\pi \in \Pi(\psi) \cap \mathfrak{S}} J(\pi) - \frac{2\kappa}{\varphi} = J(\pi^*) - \epsilon_{\mathrm{approx}}(\psi) - \frac{2\kappa}{\varphi}. \ \square$$

The following result is obtained from [4, Lemma 3], by replacing $\lambda$ and $(v,u)$ in [4, Lemma 3] with our notation $(1-\gamma)\nu$ and $(V,\lambda)$, respectively.

**Lemma E.3.** *For any dual optimal solution $(V_\kappa^*, \lambda_\kappa^*)$ of the problem* (4), *where the constraint utilities $u_i$ is replaced with the shifted utilities $u_i^\kappa$, we have*

$$\|\lambda_\kappa^*\|_1 \leq \frac{2}{\varphi} \quad \text{and} \quad \|V_\kappa^*\| \leq \frac{1}{1-\gamma}\left(1 + \frac{2}{\varphi}\right).$$

*For any $\nu \in \mathbb{R}_{\geq 0}^{|\mathcal{S}||\mathcal{A}|}$ and any $\Delta > 0$, the inequality $J(\pi_\kappa^*) - \mathcal{J}(W^{-1}\nu) \leq \Delta$ immediately implies that*

$$J(\pi_\kappa^*) - \langle r, \nu \rangle \leq \Delta, \quad \left\|A^\top \nu - \rho_0\right\|_1 \leq \frac{2\Delta}{R_\mathcal{V}}, \quad \text{and} \quad \|[U_\kappa \nu]_-\|_\infty \leq \frac{2\Delta}{R_\Lambda}$$

*as long as $R_\mathcal{V} \geq 2\|V_\kappa^*\|_\infty$, $R_\Lambda \geq 2\|\lambda_\kappa^*\|_1$.*

Finally, we introduce the last lemma that is needed in this proof.

**Lemma E.4.** *For any vector $\tilde{\nu} \in \mathbb{R}_{\geq 0}^{|\mathcal{S}||\mathcal{A}|}$ that is an approximate visitation measure, consider its associate policy $\tilde{\pi}$ defined by $\tilde{\pi}(a|s) = \frac{\tilde{\nu}(s,a)}{\sum_{a'} \tilde{\nu}(s,a')}$. Let $\nu^{\tilde{\pi}}$ be the true visitation measure of $\tilde{\pi}$, then*

$$\left\|\tilde{\nu} - \nu^{\tilde{\pi}}\right\|_1 \leq \frac{1}{1-\gamma}\left\|A^\top \tilde{\nu} - \rho_0\right\|_1.$$

*Proof.* For policy $\pi$, we consider its state visitation measure $\nu_\pi$ defined by $\nu_\pi(s) = \sum_a \nu^\pi(s,a)$. Then $\nu^\pi(s,a) = \pi(a|s)\nu_\pi(s)$. With the transition matrix $\mathbb{P}_\pi(s'|s) = \sum_a \pi(a|s)\mathbb{P}(s'|s,a)$, then the constraint $A^\top \nu^\pi = \rho_0$ is equivalent to $(I - \gamma\mathbb{P}_\pi)\nu_\pi = \rho_0$.

Let $\tilde{\pi}$ induced by $\tilde{\nu}$, then $\nu_{\tilde{\pi}}$ satisfies $(I - \gamma\mathbb{P}_{\tilde{\pi}})\nu_{\tilde{\pi}} = \rho_0$. Let $\tilde{\nu}'$ be defined by $\tilde{\nu}'(s) = \sum_a \tilde{\nu}(s,a)$, then $\tilde{\nu}(s,a) = \tilde{\pi}(a|s)\tilde{\nu}'(s)$, and hence $(I - \gamma\mathbb{P}_{\tilde{\pi}})\tilde{\nu}' = A\tilde{\nu}$. Therefore,

$$\|\nu_{\tilde{\pi}} - \tilde{\nu}'\|_1 = \left\|(I - \gamma\mathbb{P}_{\tilde{\pi}})^{-1}(\rho_0 - A^\top\tilde{\nu})\right\|_1 \leq \left\|(I - \gamma\mathbb{P}_{\tilde{\pi}})^{-1}\right\|_1 \left\|A^\top\tilde{\nu} - \rho_0\right\|_1 \leq \frac{1}{1-\gamma}\left\|A^\top\tilde{\nu} - \rho_0\right\|_1.$$

We finalize the proof by the following equality

$$\left\|\nu^{\tilde{\pi}} - \tilde{\nu}\right\|_1 = \sum_{s,a}\left|\tilde{\pi}(a|s)(\nu_{\tilde{\pi}}(s) - \tilde{\nu}'(s))\right| = \sum_s \left(\sum_a \tilde{\pi}(a|s)\right)|\nu_{\tilde{\pi}}(s) - \tilde{\nu}'(s)| = \|\nu_{\tilde{\pi}} - \tilde{\nu}'\|_1. \quad \square$$

## E.2 Analysis

Now we are ready to present the proof of Remark 4.2 and Theorem 4.1.

*Proof.* By definition of $\mathrm{Gap}(\overline{x})$, we have

$$\mathrm{Gap}(\overline{x}) = \max_{x \in \mathcal{X}} \min_{V \in \mathcal{V}, \lambda \in \Lambda} \mathcal{L}_w(V, \lambda, x) - \min_{V \in \mathcal{V}, \lambda \in \Lambda} \mathcal{L}_w(V, \lambda, \overline{x}) = j(\psi) - \mathcal{J}_\kappa(\overline{x}). \tag{32}$$

Define $\overline{\nu} = W\overline{x}$, and define $\Delta := J(\pi_\kappa^*) - \mathcal{J}_\kappa(W^{-1}\overline{\nu})$, then we have

$$\Delta = \mathrm{Gap}(\overline{x}) + J(\pi_\kappa^*) - j(\psi) \overset{(i)}{\leq} \mathrm{Gap}(\overline{x}) + J(\pi^*) - j(\psi) \overset{(ii)}{\leq} \mathrm{Gap}(\overline{x}) + \epsilon_{\mathrm{approx}}(\psi) + \frac{2\kappa}{\varphi}, \tag{33}$$

where (i) is because $J(\pi_\kappa^*) \leq J(\pi^*)$ and (ii) is due to Lemma E.2. Now, let $\nu^{\overline{\pi}}$ be the true visitation measure of $\overline{\pi}$, where $\overline{\pi}(a|s) := \frac{\overline{x}(s,a)}{\sum_{a'} \overline{x}(s,a')}$. Then Lemma E.4 immediately indicates that

$$
\begin{aligned}
\left\|\nu^{\overline{\pi}} - \overline{x}\right\|_1 &\leq \frac{1}{1-\gamma}\left\|A^\top\overline{x} - \rho_0\right\|_1 \\
&\leq \frac{1}{1-\gamma}\left(\|A^\top(\overline{x} - W\overline{x})\|_1 + \|A^\top W\overline{x} - \rho_0\|_1\right) \\
&\leq \frac{1}{1-\gamma}\left(2\|\overline{x} - \overline{\nu}\|_1 + \left\|A^\top\overline{\nu} - \rho_0\right\|_1\right)
\end{aligned}
$$

which further gives

$$\left\|\nu^{\overline{\pi}} - \overline{\nu}\right\|_1 \leq \frac{1}{1-\gamma}\left(\left\|A^\top\overline{\nu} - \rho_0\right\|_1 + 3\|\overline{x} - \overline{\nu}\|_1\right). \tag{34}$$

Consequently, we have

$$J(\pi_\kappa^*) - \mathcal{J}_\kappa(W^{-1}\nu^{\overline{\pi}})$$

$$\overset{(i)}{=} \quad J(\pi_\kappa^*) - \langle r, \nu^{\overline{\pi}} \rangle + R_\Lambda \big\| \big[U_\kappa \nu^{\overline{\pi}}\big]_- \big\|_\infty$$

$$\overset{(ii)}{\leq} \quad J(\pi_\kappa^*) - \langle r, \overline{\nu} \rangle + R_\Lambda \big\| [U_\kappa \overline{\nu}]_- \big\|_\infty + \frac{1 + \frac{3}{2}R_\Lambda}{1-\gamma} \big( \big\| A^\top \overline{\nu} - \rho_0 \big\|_1 + 3 \big\| \overline{x} - \overline{\nu} \big\|_1 \big)$$

$$\overset{(iii)}{\leq} \quad J(\pi_\kappa^*) - \mathcal{J}_\kappa(W^{-1}\overline{\nu}) + \frac{5(R_\Lambda + 1)}{1-\gamma} \big\| \overline{x} - \overline{\nu} \big\|_1$$

$$\overset{(iv)}{\leq} \quad \Delta + 45\epsilon_e,$$

where (i) is because $\|A^\top \nu^{\overline{\pi}} - \rho_0\|_1 = 0$, (ii) is due to the fact that $\big| \langle r, \nu^{\overline{\pi}} \rangle - \langle r, \overline{\nu} \rangle \big| \leq \big\| \nu^{\overline{\pi}} - \overline{\nu} \big\|_1$ and $\big| \|[U_\kappa \nu^{\overline{\pi}}]_-\|_\infty - \|[U_\kappa \overline{\nu}]_-\|_\infty \big| \leq \frac{3}{2} \big\| \nu^{\overline{\pi}} - \overline{\nu} \big\|_1$, (iii) is because of $\frac{1 + \frac{3}{2}R_\Lambda}{1-\gamma} \leq R_\mathcal{V}$, and (iv) is because of $\|\overline{x} - \overline{\nu}\|_1 \leq \varphi(1-\gamma)\epsilon_e$ by Proposition 4.3. Finally, applying Lemma E.3 to $\nu^{\overline{\pi}}$ yields

$$J(\pi_\kappa^*) - \langle r, \nu^{\overline{\pi}} \rangle \leq \Delta + 45\epsilon_e, \qquad \Big\| \big[U_\kappa \nu^{\overline{\pi}}\big]_- \Big\|_\infty \leq \frac{\varphi}{4} \left( \Delta + 45\epsilon_e \right).$$

By Lemma E.2, we have

$$J(\pi^*) - \langle r, \nu^{\overline{\pi}} \rangle \leq J(\pi^*) - j(\psi) + \mathrm{Gap}(\overline{x}) + 45\epsilon_e \leq \mathrm{Gap}(\overline{x}) + \epsilon_{\mathrm{approx}}(\psi) + \frac{2\kappa}{\varphi} + 45\epsilon_e,$$

$$J_i^u(\overline{\pi}) \geq \kappa - \big\| \big[U_\kappa \nu^{\overline{\pi}}\big]_- \big\|_\infty \geq \frac{\kappa}{2} - \frac{\varphi}{4} \big( \mathrm{Gap}(\overline{x}) + \epsilon_{\mathrm{approx}}(\psi) \big) - 12\varphi\epsilon_e. \tag{35}$$

Combining the above inequality with the fact that $\epsilon_e = \frac{\epsilon}{100}$, $\kappa = 5\varphi\epsilon$, $\mathrm{Gap}(\overline{x}) \leq \epsilon/2$ completes the proof. $\square$

Finally, we point out a by-product of the above analysis, which is useful for the VERIFY method.

**Corollary E.5.** *Under the same assumption of Theorem 4.1, with probability at least $1 - 2\delta/3$, it holds that*

$$\big\| A^\top \overline{\nu} - \rho_0 \big\|_1 \leq \frac{11}{8}\varphi(1-\gamma)\epsilon, \qquad \big\| [U_\kappa \overline{\nu}]_- \big\|_\infty \leq \frac{11}{4}\varphi\epsilon \tag{36}$$

*for $\overline{\nu} := W\overline{x}$.*

*Proof.* Due to $\psi \geq C^*$ and Lemma E.2, we have

$$J(\pi_\kappa^*) - \mathcal{J}_\kappa(\overline{x}) \leq j(\psi) - \mathcal{J}_\kappa(\overline{x}) + \frac{2\kappa}{\varphi} = \mathrm{Gap}(\overline{x}) + \frac{2\kappa}{\varphi} \leq 11\epsilon.$$

Applying Lemma E.3 yields

$$\big\| A^\top \overline{\nu} - \rho_0 \big\|_1 \leq \frac{2 \cdot 11\epsilon}{R_\mathcal{V}} \leq \frac{11}{8}\varphi(1-\gamma)\epsilon, \qquad \big\| [U_\kappa \overline{\nu}]_- \big\|_\infty \leq \frac{2 \cdot 11\epsilon}{R_\Lambda} = \frac{11}{4}\varphi\epsilon. \qquad \square$$

# F   Proofs for Section 5

## F.1   Proof of Theorem 5.1

In this section, we provide the complete version of the construction illustrated in Section 5. Let us define

$$K := \min\left( \left\lfloor \frac{I}{2} \right\rfloor, \left\lfloor \frac{A-1}{2} \right\rfloor \right), \quad S_c = \min\left( \left\lfloor \frac{I}{2K} \right\rfloor, S \right), \quad S_u = \begin{cases} S - S_c, & \text{if } S_c < S - 3, \\ 0, & \text{otherwise.} \end{cases}$$

The CMDP instance $\mathcal{M}$ that we construct consists of two groups of basic blocks. The first group includes $S_c$ replicas of the basic block characterized in Fig. 1, each with actions $\{a_1, b_1, ..., a_k, b_k, e\}$ and $2K$ constraints. The second group includes $S_u$ replicas of the basic blocks characterized by Fig. 1 (a) and Fig. 1(c), each basic block only has two actions $\{a, e\}$ and no constraint. In fact the

construction of the second group ("unconstrained part") is similar to the hard MDP constructed in [21]. The transition kernel $\mathbb{P}_\theta$ of $\mathcal{M}$ is parametrized by $\theta = (\theta_c, \theta_u) \in \Theta := \{-1, +1\}^{S_c K} \times \{-1, +1\}^{S_u}$ and $\varpi_c, \varpi_u \in (0, \frac{1}{2}]$. The details of $\mathcal{M}$ are listed as follows.

**States and actions** The state space $\mathcal{S}$ consists of $S_c + S_u$ 4-state basic blocks, plus an extra "null" state $s_{-1}$. The first $S_c$ basic blocks are exactly what we described in Section 5, we write $\mathcal{S}_c = \bigsqcup_{j=1}^{S_c} \{s_0^j, s_1^j, s_\oplus^j, s_\ominus^j\}$. The next $S_u$ basic blocks will be described below, we write $\mathcal{S}_u = \bigsqcup_{j=S_c+1}^{S_c+S_u} \{s_0^j, s_1^j, s_\oplus^j, s_\ominus^j\}$. By default, $\mathcal{S}_u = \emptyset$ if $S_u = 0$. Then $\mathcal{S} = \mathcal{S}_c \bigsqcup \mathcal{S}_u \bigsqcup \{s_{-1}\}$. Next, we describe the detailed information of each block $j$.

- At $s_0^j$, $s_\oplus^j$ and $s_\ominus^j$, there is no action, and the transition does not depend on $\theta$:

$$
\begin{aligned}
\mathbb{P}\left(s_0^j \middle| s_0^j\right) = p, \quad & \mathbb{P}\left(s_1^j \middle| s_0^j\right) = 1 - p, \\
\mathbb{P}\left(s_\oplus^j \middle| s_\oplus^j\right) = q, \quad & \mathbb{P}\left(s_0^j \middle| s_\oplus^j\right) = 1 - q, \\
\mathbb{P}\left(s_\ominus^j \middle| s_\ominus^j\right) = q, \quad & \mathbb{P}\left(s_0^j \middle| s_\ominus^j\right) = 1 - q,
\end{aligned}
\tag{37}
$$

  where $p = \frac{1}{2-\gamma}$ and $q = 2 - \frac{1}{\gamma}$. We assign reward as $r(s_\oplus^j) = 1$, $r(s_\ominus^j) = -1$.

- **Constrained state** At $s_1^j \in \mathcal{S}_c$, there are $2K + 1$ actions $a_1, b_1, \cdots, a_K, b_K, e$ such that

$$
\begin{aligned}
\mathbb{P}_\theta\left(s_\oplus^j \middle| s_1^j, a_i\right) = \frac{1 + \varpi_c \theta_{i,j}}{2}, \quad & \mathbb{P}_\theta\left(s_\ominus^j \middle| s_1^j, a_i\right) = \frac{1 - \varpi_c \theta_{i,j}}{2}, \\
\mathbb{P}\left(s_\oplus^j \middle| s_1^j, b_i\right) = \frac{1}{2}\left(1 - \frac{\varpi_c}{2}\right), \quad & \mathbb{P}\left(s_\ominus^j \middle| s_1^j, a_i\right) = \frac{1}{2}\left(1 + \frac{\varpi_c}{2}\right), \\
\mathbb{P}\left(s_\oplus^j \middle| s_1^j, e\right) = \frac{1}{2}, \quad & \mathbb{P}\left(s_\ominus^j \middle| s_1^j, e\right) = \frac{1}{2}.
\end{aligned}
$$

  Here we use subscript $\theta$ to emphasize the dependency of $\mathbb{P}_\theta$ on $\theta$.[4]

- **Unconstrained state** At $s_1^j \in \mathcal{S}_u$, there are two actions $a, e$ such that

$$
\begin{aligned}
\mathbb{P}_\theta\left(s_\oplus^j \middle| s_1^j, a\right) = \frac{1 + \varpi_u \theta_j}{2}, \quad & \mathbb{P}_\theta\left(s_\ominus^j \middle| s_1^j, a\right) = \frac{1 - \varpi_u \theta_j}{2}, \\
\mathbb{P}\left(s_\oplus^j \middle| s_1^j, e\right) = \frac{1}{2}, \quad & \mathbb{P}\left(s_\ominus^j \middle| s_1^j, e\right) = \frac{1}{2}.
\end{aligned}
$$

- The null state $s_{-1}$ has no action or reward, and it always transits to itself.

**Initial distribution** In the initial distribution, $\rho_0(s_{-1}) = \rho_0(s_1^j) = \rho_0(s_\oplus^j) = \rho_0(s_\ominus^j) = 0, \forall j$. The nonzero probabilities only spread across the $\{s_0^j\}$. In the case $S_u > 0$, we choose $\rho_0$ to be

$$
\rho_0(s_0^j) = \begin{cases} \frac{\mathbb{I}\{s_0^j \in \mathcal{S}_c\}}{2S_c} + \frac{\mathbb{I}\{s_0^j \in \mathcal{S}_u\}}{2S_u}, & \text{if } \mathcal{S}_u \neq \emptyset, \\ \frac{1}{S_c}, & \text{otherwise.} \end{cases}
$$

Without loss of generality, we will only deal with the case where $\mathcal{S}_u \neq \emptyset$.

**Constraints** At each constrained block in $\mathcal{S}_c$, for each pair of actions $(a_i, b_i)$ at the state $s_0^j \in \mathcal{S}_c$, we introduce two constraints defined by the utilities

$$
u_{i,j}(s_1^j, a_i) = -1, \qquad u_{i,j}(s_1^j, b_i) = 1, \qquad \tilde{u}_{i,j}(s_1^j, b_i) = -1.
$$

At all the other state and actions, $u_{i,j}$ and $\tilde{u}_{i,j}$ returns 0. Then we set the constraints to be

$$
J_{i,j}^u(\pi) := \langle \nu^\pi, u_{i,j} \rangle \geq 0, \qquad \text{and} \qquad \tilde{J}_{i,j}^u(\pi) := \langle \nu^\pi, \tilde{u}_{i,j} \rangle \geq -\frac{\rho_c v_1}{4K},
$$

---

[4]Here we view $\theta_c \in \{-1, 1\}^{S_c K}$ as a vector indexed by $(i, j) \in [K] \times [S_c]$, and $\theta_{i,j}$ stands for the $(i,j)$-th component of $\theta_c$. Similarly, we view $\theta_u \in \{-1, 1\}^{S_u}$ as a vector indexed by $j$ with $S_c + 1 \leq j \leq S_c + S_u$, and $\theta_j$ stands for the $j$-th component of $\theta_u$.

where $\rho_c$ and $v_1$ are constants specified later in (38). After suitable shifting we can make sure that each constraint has the form $J^u \geq 0$. Basically, these two constraints are equivalent to $\pi(a_i|s_1^j) \leq \pi(b_i|s_1^j) \leq \frac{1}{4K}$. We remark that there are in total $S_c K \leq I$ constraints.

**Optimal policy** First, let us calculate the visitation measure of any given policy $\pi$. According to the proof of Lemma E.4, we set $\nu_\pi$ be the state visitation measure and let $\mathbb{P}_\pi$ be the state transition matrix under policy $\pi$, then $\nu_\pi$ will be the unique solution to $(I - \gamma \mathbb{P}_\pi)\nu_\pi = \rho_0$. Note that the $S_c + S_u$ basic blocks are in fact independent blocks, i.e., there are no transitions between different blocks. The matrix $(I - \gamma \mathbb{P}_\pi)$ is in fact a block-diagonal with $S_c + S_u$ 4 by 4 blocks and a 1 by 1 block, and we can solve the $\nu_\pi$ block by block. Define the constants

$$v_0 = \frac{2}{(2+\gamma)}, \quad v_1 = \frac{2\gamma}{(2+\gamma)(2-\gamma)}, \quad v = \frac{\gamma^2}{(2+\gamma)(2-\gamma)}, \quad \rho_c = \frac{1}{2S_c}, \quad \rho_u = \frac{1}{2S_u}, \quad (38)$$

and we consider

$$r_j(\pi) = \begin{cases} \sum_i \left( \theta_{i,j}\pi(a_i|s_1^j) - \frac{1}{2}\pi(b_i|s_1^j) \right), & s_1^j \in \mathcal{S}_c, \\ \theta_j \pi(a|s_1^j), & s_1^j \in \mathcal{S}_u. \end{cases}$$

By a direct computation, the state visitation measure of $\pi$ is given by

$$\nu_\pi(s_\oplus^j) = \frac{v}{1-\gamma}\frac{1 + \varpi_\diamond r_j(\pi)}{2}, \qquad \nu_\pi(s_0^j) = \frac{\rho_\diamond v_0}{1-\gamma},$$

$$\nu_\pi(s_\ominus^j) = \frac{v}{1-\gamma}\frac{1 - \varpi_\diamond r_j(\pi)}{2}, \qquad \nu_\pi(s_1^j) = \frac{\rho_\diamond v_1}{1-\gamma},$$

where $\diamond$ stands for $c$ if the block $j$ belongs to $\mathcal{S}_c$, and $\diamond$ stands for $u$ if the block $j$ belongs to $\mathcal{S}_u$. Consequently, the cumulative reward and the utilities are

$$J(\pi; \theta) = \sum_j \left( \nu_\pi(s_\oplus^j) - \nu_\pi(s_\ominus^j) \right) = \frac{v}{1-\gamma}\left( \rho_c \varpi_c \sum_{j:s_1^j \in \mathcal{S}_c} r_j(\pi) + \rho_u \varpi_u \sum_{j:s_1^j \in \mathcal{S}_u} r_j(\pi) \right),$$

$$J_{i,j}(\pi; \theta) = \nu^\pi(s_1^j, b_i) - \nu^\pi(s_1^j, a_i) = \rho_c v_1 \left( \pi(b_i|s_1^j) - \pi(a_i|s_1^j) \right), \tag{39}$$

$$\tilde{J}_{i,j}(\pi; \theta) = -\nu^\pi(s_1^j, b_i) = -\rho_c v_1 \pi(b_i|s_1^j).$$

Therefore, $\pi$ being safe is equivalent to requiring $\pi(a_i|s_1^j) \leq \pi(b_i|s_1^j) \leq \frac{1}{4K}$ for all the constrained block $j$ in $\mathcal{S}_c$, and any $1 \leq i \leq K$. With the above explicit expression of $J(\pi; \theta)$, we know that the (unique) optimal policy $\pi^{*,\theta}$ under the transition dynamic $\mathbb{P}_\theta$ is

$$\pi^{*,\theta}(a_i|s_1^j) = \pi^{*,\theta}(b_i|s_1^j) = \frac{\mathbb{I}\{\theta_{i,j} = 1\}}{4K}, \qquad s_1^j \in \mathcal{S}_c,$$

$$\pi^{*,\theta}(a|s_1^j) = \mathbb{I}\{\theta_j = 1\}, \qquad s_1^j \in \mathcal{S}_u. \tag{40}$$

Denote $J_\theta^* := J(\pi^{*,\theta}; \theta)$ the optimal safe reward and $\tilde{\theta} = \frac{\theta+1}{2}$, then

$$J_\theta^* = J(\pi^{*,\theta}; \theta) = \frac{v}{1-\gamma}\left( \varpi_c \rho_c \sum_{j:s_1^j \in \mathcal{S}_c} \sum_{i=1}^{K} \frac{\tilde{\theta}_{i,j}}{8K} + \varpi_u \rho_u \sum_{j:s_1^j \in \mathcal{S}_u} \tilde{\theta}_j \right). \tag{41}$$

**Reference distribution** Finally, we set the reference distribution $\mu$ as

$$\mu(s_0^j) = \frac{v_0}{C}\rho_\diamond, \qquad \mu(s_\oplus^j) = \frac{3}{4}\frac{v}{C}\rho_\diamond, \qquad \mu(s_\ominus^j) = \frac{1}{2}\frac{v}{C}\rho_\diamond, \qquad \mu(s_1^j, e) = \frac{v_1(1-\gamma)}{C}\rho_\diamond,$$

$$\begin{cases} \mu(s_1^j, a_i) = \mu(s_1^j, b_i) = \frac{\rho_c v_1(1-\gamma)}{4KC}, \ i \in [I] & s_1^j \in \mathcal{S}_c, \\ \mu(s_1^j, a) = \frac{\rho_u v_1(1-\gamma)}{C}, & s_1^j \in \mathcal{S}_u, \end{cases}$$

$$\mu(s_{-1}) = 1 - \sum_j \left( \mu(s_0^j) + \mu(s_1^j) + \mu(s_\oplus^j) + \mu(s_\ominus^j) \right).$$

As long as $C \geq 2$, $\mu(s_{-1})$ defined above is positive. Also, for any $\theta$, it holds that

$$\max_{s,a} \frac{\nu^{\pi^{*,\theta}}(s,a)}{\mu(s,a)} \leq \frac{C}{1-\gamma}, \quad \sum_{s,a} \frac{\nu^{\pi^{*,\theta}}(s,a)}{\mu(s,a)} \leq \frac{(|\mathcal{S}| + I)C}{1-\gamma}.$$

We denote $\mu_\theta = \mu \otimes \mathbb{P}_\theta$ as the probability measures of the transition pair $\zeta = (s, a, s')$ generated from the reference distribution $\mu$.

**Output policy as an estimator of $\theta$** Assume that an algorithm $\mathfrak{A}$ consumes $N$ samples generated from $\mu_\theta$, and outputs a policy $\hat{\pi}$ that is possibly dependent on the internal randomness of $\mathfrak{A}$. Consider the corresponding random vector $\hat{\pi}_c := \left(\hat{\pi}(a_i|s_1^j)\right)_{i,j}$ and $\hat{\pi}_u := \left(\hat{\pi}(a|s_1^j)\right)_j$. Then, $4K\hat{\pi}_c$ can be viewed as an estimator of $\tilde{\theta}_c$, and $\hat{\pi}_u$ can be viewed as an estimator of $\tilde{\theta}_u$. We establish the following lemma to characterize the error for "misspecifying" the parameter $\theta$.

**Lemma F.1.** *For any policy $\pi$, we define*

$$\mathcal{L}(\pi; \theta) := [J_\theta^* - J(\hat{\pi}; \theta)]_+ + \frac{\gamma \varpi_c}{1 - \gamma} \sum_{i,j} \left([J_{i,j}(\hat{\pi}; \theta)]_- + \left[\tilde{J}_{i,j}(\hat{\pi}; \theta) - \frac{v_1}{4SK}\right]_-\right) \tag{42}$$

$$= [J_\theta^* - J(\hat{\pi}; \theta)]_+ + \frac{\gamma \varpi_c}{1 - \gamma}\text{violation}(\hat{\pi}; \theta).$$

*Then it holds that*

$$\mathcal{L}(\pi; \theta) \geq \frac{v\rho_c\varpi_c}{8K(1 - \gamma)} \left\|4K\hat{\pi}_c - \tilde{\theta}_c\right\|_1 + \frac{v\rho_u\varpi_u}{1 - \gamma} \left\|\hat{\pi}_u - \tilde{\theta}_u\right\|. \tag{43}$$

*Proof.* The description of $J_\theta^*$ in (41) gives

$$\mathcal{L}(\hat{\pi}; \theta) = \frac{v}{1 - \gamma} \left[\rho_c\varpi_c \sum_{i,j} \left(\frac{\tilde{\theta}_{i,j}}{8K} - \theta_{i,j}\pi(a_i|s_1^j) + \frac{\pi(b_i|s_1^j)}{2}\right) + \rho_u\varpi_u \sum_j \left(\tilde{\theta}_j - \theta_j\pi(a|s_1^j)\right)\right]_+$$

$$+ \frac{\gamma v_1 \rho_c \varpi_c}{1 - \gamma} \sum_{i,j} \left(\left[\pi(b_i|s_1^j) - \pi(a_i|s_1^j)\right]_- + \left[\frac{1}{4K} - \pi(b_i|s_1^j)\right]_-\right)$$

$$\geq \frac{v}{1 - \gamma} \left(\rho_c\varpi_c \sum_{i,j} \delta_{i,j} + \rho_u\varpi_u \sum_j \left(\tilde{\theta}_j - \theta_j\pi(a|s_1^j)\right)\right),$$

where we use the fact $\gamma v_1 = 2v$, and denote

$$\delta_{i,j} = \frac{\tilde{\theta}_{i,j}}{8K} - \theta_{i,j}\pi(a_i|s_1^j) + \frac{\pi(b_i|s_1^j)}{2} + 2\left[\pi(b_i|s_1^j) - \pi(a_i|s_1^j)\right]_- + 2\left[\frac{1}{4K} - \pi(b_i|s_1^j)\right]_-.$$

Clearly $\tilde{\theta}_j - \theta_j\pi(a|s_1^j) \geq \left|\tilde{\theta}_j - \pi(a|s_1^j)\right|$ for all $s_1^j \in \mathcal{S}_u$. As for $s_1^j \in \mathcal{S}_c$, we consider the case $\theta_{i,j} = 1$ and $\theta_{i,j} = -1$ separately.

Case 1, $\theta_{i,j} = -1$. Directly $\delta_{i,j} \geq \pi(a_i|s_1^j) = \left|\pi(a_i|s_1^j) - \frac{\tilde{\theta}_{i,j}}{4K}\right|$.

Case 2, $\theta_{i,j} = 1$. By the fact that

$$\frac{z}{2} - x + \frac{y}{2} + 2\left[y - x\right]_- + 2\left[z - y\right]_- \geq \frac{z - x}{2} + \frac{3}{2}\left[z - x\right]_- \geq \frac{|z - x|}{2} \quad \forall x, y, z,$$

we can plug in $x = \pi(a_i|s_1^j)$, $y = \pi(b_i|s_1^j)$ and $z = \frac{1}{4K}$ and derive

$$\delta_{i,j} \geq \frac{1}{2}\left|\frac{1}{4K} - \pi(a_i|s_1^j)\right|.$$

Consequently, (43) is established by combining the above inequalities. $\square$

We now invoke the following lemma due to [9] and [24].

**Lemma F.2.** *For any integer $n \geq 1$, there exists a subset $\Theta_n$ of $\{-1, 1\}^n$ such that $|\Theta_n| \geq \exp(n/8)$, and for any pair of different $\theta, \theta' \in \Theta_n$, one has $\|\theta - \theta'\|_1 \geq \frac{n}{2}$.*

Fix a $\Theta_c$ with $n = S_cK$ and a $\Theta_u$ with $n = S_u$, we consider the family of CMDPs $\mathfrak{M} := \{\mathcal{M}_\theta\}_{\theta \in \Theta_c \times \Theta_u}$. Intuitively, CMDPs from this family are hard to distinguish according to samples. This idea can be shown mathematically by the following generalized version of Fano's inequality from [3, Lemma 3].

**Lemma F.3** (Generalized Fano's inequality). *Let $r \geq 2$ be an integer and let $\mathcal{P}$ be a set of $r$ probability measures on $(\Omega, \mathcal{F})$. Assume that $\theta(\mathbb{P})$ is the parameter of interest with values in a pseudo-metric space $(\mathcal{D}, d)$. Let $\hat{\theta} = \hat{\theta}(X)$ be an estimator of $\theta(\mathbb{P})$ based on a sample $X$ from a distribution $\mathbb{P} \in \mathcal{P}$. Assume that*

$$d\left(\theta(\mathbb{P}), \theta(\mathbb{P}')\right) \geq \alpha, \quad \forall \mathbb{P}, \mathbb{P}' \in \mathcal{P},$$

*and*

$$\mathrm{KL}(\mathbb{P} \parallel \mathbb{P}') = \int_\Omega \log\left(\frac{d\mathbb{P}}{d\mathbb{P}'}\right) d\mathbb{P} \leq \beta.$$

*Then it holds that*

$$\max_{\mathbb{P} \in \mathcal{P}} \mathbb{E}_\mathbb{P} d\left(\hat{\theta}, \theta(\mathbb{P})\right) \geq \frac{\alpha}{2}\left(1 - \frac{\beta + \log 2}{\log r}\right).$$

It is worth noting that the estimator needs not to belong to $\{\theta(\mathbb{P})\}_{\mathbb{P} \in \mathcal{P}}$. In our problem, the underlying space $(\Omega, \mathcal{F})$ depends on the internal randomness of $\mathfrak{A}$, and the probability measure on $(\Omega, \mathcal{F})$ is the extension of $\mu_\theta^{\otimes N}$ ($\mu_\theta^{\otimes N}$ is the probability measure on $\Omega_0 = (\mathcal{S} \times \mathcal{A} \times \mathcal{S})^N$, the space of the $N$-tuple of samples $(\zeta_1, \cdots, \zeta_N)$).

**The proof of Theorem 5.1** We have already demonstrated that $4K\hat{\pi}_c$ can be viewed as an estimator of $\tilde{\theta}_c$ in Lemma F.1, and hence $8K\hat{\pi}_c - 1$ can be viewed as an estimator of $\theta_c$. We fix a $\theta_u \in \Theta_u$, then Fano's inequality (Lemma F.3) yields

$$\max_{\theta_c \in \Theta_c} \mathbb{E}_{(\theta_c, \theta_u)} \|8K\hat{\pi}_a - 1 - \theta_c\|_1 \geq \frac{S_c K}{2}\left(1 - \frac{\max_{\theta_c, \theta_c' \in \Theta_c} \mathrm{KL}(\mu_{(\theta_c, \theta_u)}^{\otimes N} \parallel \mu_{(\theta_c', \theta_u)}^{\otimes N}) + \log 2}{\log |\Theta_c|}\right)$$

$$= \frac{S_c K}{2}\left(1 - \frac{N \max_{\theta_c, \theta_c' \in \Theta_c} \mathrm{KL}(\mu_{(\theta_c, \theta_u)} \parallel \mu_{(\theta_c', \theta_u)}) + \log 2}{\log |\Theta_c|}\right).$$

For any $\theta_c, \theta_c' \in \Theta_c$, we have

$$\mathrm{KL}(\mu_{(\theta_c, \theta_u)} \parallel \mu_{(\theta_c', \theta_u)}) = \sum_{i,j} \mu(s_1^j, a_i) \mathrm{KL}\left(\frac{1 + \theta_{i,j}\varpi_c}{2} \;\middle\|\; \frac{1 + \theta_{i,j}'\varpi_c}{2}\right)$$

$$\leq \sum_{i,j} \mu(s_1^j, a_i) \frac{4\varpi_c^2}{1 - \varpi_c^2} = \frac{(1 - \gamma)v_1}{2C}\frac{\varpi_c^2}{1 - \varpi_c^2}.$$

Then, taking $\varpi_c = \min\left\{\sqrt{\frac{(S_c K - 3)C}{8(1-\gamma)N}}, \frac{1}{2}\right\}$ is enough to ensure

$$\frac{N \max_{\theta_c, \theta_c' \in \Theta_c} \mathrm{KL}(\mu_{(\theta_c, \theta_u)} \parallel \mu_{(\theta_c', \theta_u)}) + \log 2}{\log |\Theta_c|} \leq \frac{5}{6},$$

which further gives $\max_{\theta_c \in \Theta_c} \mathbb{E}_{(\theta_c, \theta_u)} \|8K\hat{\pi}_a - 1 - \theta_c\|_1 \geq \frac{S_c K}{12}$, and hence

$$\max_{\theta_c \in \Theta_c} \mathbb{E}_{(\theta_c, \theta_u)}\left[\frac{1}{S_c K}\left\|4K\hat{\pi}_a - \tilde{\theta}_c\right\|_1\right] \geq \frac{1}{24}.$$

Similarly, we can take $\varpi_u = \min\left\{\sqrt{\frac{(S_u - 3)C}{8(1-\gamma)N}}, \frac{1}{2}\right\}$ to ensure that for any fixed $\theta_c \in \Theta_c$,

$$\max_{\theta_u \in \Theta_u} \mathbb{E}_{(\theta_c, \theta_u)}\left[\frac{1}{S_u}\left\|\hat{\pi}_u - \tilde{\theta}_u\right\|_1\right] \geq \frac{1}{24}$$

Therefore, we obtain

$$\max_{\theta \in \Theta} \mathbb{E}_\theta \mathcal{L}(\hat{\pi}; \theta)$$

$$\geq \max_{\theta \in \Theta} \mathbb{E}_\theta\left[\frac{v\rho_c\varpi_c}{8K(1-\gamma)}\left\|4K\hat{\pi}_c - \tilde{\theta}_c\right\|_1 + \frac{v\rho_u\varpi_u}{1-\gamma}\left\|\hat{\pi}_u - \tilde{\theta}_u\right\|\right]$$

$$= \frac{v}{2(1-\gamma)}\max_{\theta_c \in \Theta_c}\max_{\theta_u \in \Theta_u}\left\{\frac{\varpi_c}{8}\mathbb{E}_{(\theta_c, \theta_u)}\left[\frac{1}{S_c K}\left\|4K\hat{\pi}_c - \tilde{\theta}_c\right\|_1\right] + \varpi_u \mathbb{E}_{(\theta_c, \theta_u)}\left[\frac{1}{S_u}\left\|\hat{\pi}_u - \tilde{\theta}_u\right\|\right]\right\}$$

$$\geq \frac{v}{2(1-\gamma)}\left(\frac{\varpi_c}{192}+\frac{\varpi_u}{24}\right) \gtrsim \min\left\{\frac{1}{1-\gamma}, \sqrt{\frac{S_c K + S_u}{(1-\gamma)^3 N}}\right\} \gtrsim \min\left\{\frac{1}{1-\gamma}, \sqrt{\frac{\min\{SA, S+I\}}{(1-\gamma)^3 N}}\right\}.$$

In conclusion, for a fixed algorithm $\mathfrak{A}$, there exists some $\theta \in \Theta_c \times \Theta_u$, such that for the policy $\hat{\pi}$ output by $\mathfrak{A}$ on $\mathcal{M}_\theta$, either

$$\mathbb{E}_{\mathcal{M}_\theta}[J_\theta^* - J(\hat{\pi})] \gtrsim \min\left\{\frac{1}{1-\gamma}, \sqrt{\frac{\min\{SA, S+I\}}{(1-\gamma)^3 N}}\right\},$$

or

$$\mathbb{E}_{\mathcal{M}_\theta}[\mathrm{violation}(\hat{\pi})] \gtrsim 1.$$

This completes the proof of Theorem 5.1.

**Remark F.4.** *The family $(\mathcal{M}_\theta)$ constructed here does not satisfy the Slater's condition with $\varphi = \Theta(1)$, but a small modification can be made to ensure a $\varphi$ with constant order. Namely, at each $s_0^j$ we add two extra arms $e, e'$, such that $r(s_0^j, e) = 0, r(s_0^j, e') = -1$ and all utilities of $e'$ is 1. The transition at $s_0^j$ is not affected by $e, e'$. We omit this construction in the argument above for the sake of cleanness and simplicity.*

## F.2 Proof of Theorem 5.2

We further extend the idea of construction in Appendix F.1 to show that, when the Slater's condition does not hold, no zero constraint violation can be ensured. Intuitively, we can directly include an extra constraint $J(\pi) \geq J^*$ in the previous construction. However, the subtlety in such a transfer is that, the constraint will leak information of the underlying parameters $\theta, \varpi$. Thus, rather than making ad hoc adaption from Appendix F.1, we present a more interesting construction for the case $I = 1$, as follows.

**States and actions** We take the state space $\mathcal{S} = \{s_{-1}, s_0, s_\oplus, s_\ominus\} \bigsqcup_{j=1}^{S} \{s^j\}$, with actions and transition dynamic specified as follows. Here we merge the states $s_0^j, s_\oplus^j, s_\ominus^j$ in Appendix F.1 for notational simplicity. The transition dynamic is parametrized by $\theta \in \{0,1\}^S$ and $\varpi \in (0, \frac{1}{2}]$, as follows.

- At $s_0, s_\oplus$ and $s_\ominus$, there is no action, and the transition does not depend on $\theta$:

$$\begin{aligned}
\mathbb{P}(s_0 \mid s_0) &= p, & \mathbb{P}(s^j \mid s_0) &= \frac{1-p}{S}, \ j \in [S], \\
\mathbb{P}(s_\oplus \mid s_\oplus) &= q, & \mathbb{P}(s_0 \mid s_\oplus) &= 1-q, \\
\mathbb{P}(s_\ominus \mid s_\ominus) &= q, & \mathbb{P}(s_0 \mid s_\ominus) &= 1-q,
\end{aligned} \tag{44}$$

  where $p = \frac{1}{2-\gamma}$ and $q = 2 - \frac{1}{\gamma}$.

- At $s^j$, there are two actions $a, b$ such that

$$\begin{aligned}
\mathbb{P}_\theta(s_\oplus \mid s^j, a) &= \frac{1-\varpi\theta_j}{2}, & \mathbb{P}_\theta(s_\ominus \mid s^j, a) &= \frac{1+\varpi\theta_j}{2}, \\
\mathbb{P}_\theta(s_\oplus \mid s^j, b) &= \frac{1-\varpi(1-\theta_j)}{2}, & \mathbb{P}_\theta(s_\ominus \mid s^j, b) &= \frac{1+\varpi(1-\theta_j)}{2}.
\end{aligned}$$

- The null state $s_{-1}$ always transits to itself.

**Utilities and rewards** We assign $u(s_\oplus) = +1, u(s_\ominus) = -1$, and $u(s_0) = u(s^j) = 0$. No reward is assigned to $\mathcal{M}$, namely the only goal in $\mathcal{M}$ is to fulfill the constraint: $J^u(\pi) \geq 0$. Basically, this constraint requires us to determine whether $\theta_i = 1$ for each $i$.

**Optimal policy** For any policy $\pi$, we define

$$r_j(\pi) = \theta_j \pi(a \mid s_1^j) + (1-\theta_j)\pi(b \mid s_1^j), \qquad \bar{r}(\pi) = \frac{1}{S}\sum_{j=1}^{S} r_j(\pi).$$

Then by exactly the same calculation as in Appendix F.1, we have

$$\nu_\pi(s_0) = \frac{v_0}{1-\gamma}, \qquad\qquad \nu_\pi(s^j) = \frac{v_1}{S},$$

$$\nu_\pi(s_\oplus) = \frac{1-\varpi\bar{r}(\pi)}{2}\frac{v}{1-\gamma}, \qquad\qquad \nu_\pi(s_\ominus) = \frac{1+\varpi\bar{r}(\pi)}{2}\frac{v}{1-\gamma}.$$

Therefore, it holds that

$$J^u(\pi) = -\frac{v}{1-\gamma}\bar{r}(\pi) = -\frac{v}{S(1-\gamma)}\left\|\pi_b - \theta\right\|_1, \tag{45}$$

where we denote $\pi_b = \left(\pi(b|s_1^j)\right)_j$ for a policy $\pi$. Hence, there is a unique safe policy $\pi^{*,\theta}$ in $\mathcal{M}_\theta$ that can be specified by

$$\pi^{*,\theta}(a|s^j) = 1 - \theta_j, \qquad \pi^{*,\theta}(b|s^j) = \theta_j, \qquad j \in [S].$$

The formula (45) also indicates that, for $\hat{\pi}$ outputed by an algorithm $\mathfrak{A}$ after consuming $N$ samples, the vector $\hat{\pi}_b$ can be viewed as an estimator of $\theta$.

**Reference distribution** We take $\rho_0(s_0) = 1$. The reference distribution $\mu$ is chosen similar to Appendix F.1, namely

$$\mu(s_0) = \frac{v_0}{C}, \qquad\qquad \mu(s^j, a) = \mu(s^j, b) = \frac{v_1(1-\gamma)}{SC},$$

$$\mu(s_\oplus) = \mu(s_\ominus) = \frac{v}{C}, \qquad \mu(s_{-1}) = 1 - \mu(s_0) - \mu(s_\oplus) - \mu(s_\ominus) - \sum_j \mu(s^j).$$

As long as $C \geq 2$, $\mu(s_{-1})$ defined above is positive. Also, for any $\theta$, it holds that

$$\max_{s,a} \frac{\nu^{\pi^{*,\theta}}(s,a)}{\mu(s,a)} \leq \frac{C}{1-\gamma}, \quad \sum_{s,a} \frac{\nu^{\pi^{*,\theta}}(s,a)}{\mu(s,a)} \leq \frac{(|\mathcal{S}|+1)C}{1-\gamma}.$$

**Lower bound** Still, we take a subset $\Theta$ of $\{0,1\}^S$ such that $|\Theta| \geq \exp(S/8)$, and for any pair of different $\theta, \theta' \in \Theta$ it holds $\|\theta - \theta'\|_1 \geq \frac{S}{4}$. We next consider the family of CMDPs $\mathfrak{M} := \{\mathcal{M}_\theta\}_{\theta \in \Theta}$, with the reference $\mu_\theta = \mu \otimes \mathbb{P}_\theta$.

By Fano's inequality (Lemma F.3), it holds that

$$\max_{\theta \in \Theta} \mathbb{E}_\theta \left\|\hat{\pi}_b - \theta\right\|_1 \geq \frac{S}{4}\left(1 - \frac{N\max_{\theta,\theta'\in\Theta} \mathrm{KL}(\mu_\theta \,\|\, \mu_{\theta'}) + \log 2}{\log|\Theta|}\right).$$

We also have $\max_{\theta,\theta'\in\Theta} \mathrm{KL}(\mu_\theta \,\|\, \mu_{\theta'}) \leq \frac{2\varpi^2(1-\gamma)}{C}$ by a simple calculation. Therefore, taking $\varpi = \min\left\{\sqrt{\frac{(S-3)C}{16(1-\gamma)N}}, \frac{1}{2}\right\}$ is enough to ensure $\max_{\theta\in\Theta} \mathbb{E}_\theta \left\|\hat{\pi}_b - \theta\right\|_1 \geq \frac{S}{24}$. Hence, we obtain

$$\max_{\theta\in\Theta} \mathbb{E}_\theta\left[J^u(\hat{\pi};\theta)\right]_- = \max_{\theta\in\Theta} \mathbb{E}_\theta\left[\frac{v\varpi}{S(1-\gamma)}\left\|\hat{\pi}_b - \theta\right\|_1\right] \geq \frac{v\varpi}{24(1-\gamma)} \gtrsim \min\left\{\sqrt{\frac{SC}{(1-\gamma)^3 N}}, \frac{1}{1-\gamma}\right\}.$$

## G  The Adaptive-DPDL framework

### G.1  The verification method

First, let us provide the details of the VERIFY$(\cdot)$ method that is used in Algorithm 2.

As a remark, $\widehat{\Delta}_p$ is an estimator of the residual $A^\top W\overline{x} - \rho_0$, where $A$ is defined in (29), that is $\mathbb{E}_\mathcal{D}\left[\widehat{\Delta}_p\right] = A^\top W\overline{x} - \rho_0$. By a direct computation, we also know $\mathbb{E}_\mathcal{D}\left[\widehat{J}(\overline{\pi})\right] = r^\top W\overline{x}$ and $\mathbb{E}_\mathcal{D}\left[\widehat{J}^{u^\kappa}(\pi)\right] = U_\kappa W\overline{x}$. Intuitively, when $\|\widehat{\Delta}_p\|_1$ is small, then $W\overline{x}$ is a good approximation of $\nu^{\overline{\pi}}$ and thus $\widehat{J}(\overline{\pi}), \widehat{J}^{u^\kappa}(\overline{\pi})$ are good approximations of $J(\overline{\pi}), J^{u^\kappa}(\overline{\pi})$. With this in mind, we present the following proposition that characterizes the VERIFY method, whose proof is moved to Appendix G.3.

---

**Algorithm 3:** $\text{VERIFY}(\overline{x}) = \text{VERIFY}(\overline{x}; \epsilon, \delta)$

---

**input** : The output $\overline{x}$ and the parameters $\epsilon, \delta > 0$ in Algorithm 1.

1 Obtain $N_v$ offline samples $\{(s_t, a_t, s'_t, r_t, \mathbf{u}_t)\}_{t=1}^{N_v}$ from $\mathcal{D}$;

2 Compute the estimators $\widehat{J}(\overline{\pi}), \widehat{J}^{u^\kappa}(\overline{\pi}) \in \mathbb{R}$ and $\widehat{\Delta}_p \in \mathbb{R}^{|\mathcal{S}|}$ as

$$\widehat{J}(\overline{\pi}) := \frac{1}{N_v} \sum_{t=1}^{N_v} r_t \frac{\overline{x}(s_t, a_t)}{\hat{\mu}(s_t, a_t)}, \qquad \widehat{J}^{u^\kappa}(\overline{\pi}) := \frac{1}{N_v} \sum_{t=1}^{N_v} \mathbf{u}_t^\kappa \frac{\overline{x}(s_t, a_t)}{\hat{\mu}(s_t, a_t)},$$

$$\widehat{\Delta}_p(s') := \sum_a \frac{N(s', a)}{N_v} \frac{\overline{x}(s', a)}{\hat{\mu}(s', a)} - \gamma \sum_{s,a} \frac{N(s, a, s')}{N_v} \frac{\overline{x}(s, a)}{\hat{\mu}(s, a)} - \rho_0(s'), \quad \forall s' \in \mathcal{S},$$

where $N(s,a), N(s,a,s')$ are the times that $(s,a)$ and $(s,a,s')$ are observed in the $N_v$ samples.

3 **if** $\left\| \widehat{\Delta}_p \right\|_1 \leq \frac{3}{2}\varphi(1-\gamma)\epsilon$ && $\left\| \widehat{J}^{u^\kappa}(\overline{\pi}) \right\|_\infty \leq 3\varphi\epsilon$ **then**

4 $\quad$ Return $\text{VERIFY}(\overline{x}) = \text{TRUE}$, and return $\widehat{J}(\overline{\pi})$ as an estimate of $J(\overline{\pi})$;

5 **else**

6 $\quad$ Return $\text{VERIFY}(x) = \text{FALSE}$;

---

**Proposition G.1.** *For the VERIFY method, if we choose $N_v \geq \frac{64|\mathcal{S}|\psi\ell}{\varphi^2(1-\gamma)^4\epsilon_{\mathrm{ver}}^2}$, with $\ell = 4\log\left(\frac{40|\mathcal{S}|I}{\delta}\right)$ and $\epsilon_{\mathrm{ver}} = \frac{\epsilon}{10}$, then with probability at least $1 - \delta$, it holds that:*

*(1). If VERIFY($\overline{x}$) = FALSE, then $\psi < C^*$.*

*(2). If VERIFY($\overline{x}$) = TRUE, then $J^{u^\kappa}(\overline{\pi}) \geq 0$, and $j_0(\psi) - 400\epsilon \leq \widehat{J}(\overline{\pi}) \leq j_0(\psi) + 100\epsilon$.*

Basically, this proposition states that if VERIFY($\overline{x}$) = FALSE, then we know $\psi < C^*$ with high probability. If VERIFY($\overline{x}$) = TRUE, then we know that $\overline{\pi}$ is safe, and $j_0(\psi) = \widehat{J}(\overline{\pi}) + \mathcal{O}(\epsilon)$. We can apply Lemma 6.1 to determine whether the current policy is good enough.

### G.2 The adaptive-DPDL method

In this section we will discuss the details of Algorithm 2. The key to the analysis of this section is Lemma 6.1, whose proof is presented in Appendix G.4.

**Setting of sub-routine** We use $\epsilon'$ for the input sub-optimality of Adaptive-DPDL. At each step $K$, we call DPDL and VERIFY with $\epsilon = \frac{\epsilon'}{15}$ and $\delta_K := \frac{6\delta}{\pi^2 K^2}$. The $\delta_K$ is chosen so that $\sum_K \delta_K = \delta$.

**Exit condition** In Algorithm 2, line 4 to 6, we write the exit condition as $-\infty < J^K \leq J^{K-1} + \mathcal{O}(\epsilon)$. More specifically, the exit condition can be equivalently stated as

$$\text{VERIFY}(x^{(K)}) \ \&\& \ \text{VERIFY}(x^{(K-1)}) \ \&\& \ \widehat{J}(\pi^{(K)}) - \widehat{J}(\pi^{(K-1)}) \leq 500\epsilon. \tag{46}$$

Here the third condition only needs to be checked when both VERIFY($x^{(K)}$) and VERIFY($x^{(K-1)}$) return TRUE. The constant $500$ is chosen to ensure that Adaptive-DPDL will exit for $\psi_K > 2C^*$, as will be demonstrated in the following proposition, whose proof is presented in Appendix G.5.

**Proposition G.2.** *Suppose Algorithm 2 exits at step $K$. Then with probability at least $1-\delta$, the following results hold. (1) $\pi^{(K)}$ is safe and $\psi_K \leq 4C^*$. (2) It holds that $J^* - J(\pi^{(K)}) \leq \mathcal{O}\left(\frac{C^*}{\psi_K}\epsilon\right)$. (3) There is a constant $\epsilon_0(\mathcal{M})$ such that for $\epsilon' \leq \epsilon_0(\mathcal{M})$, $\psi_K \geq C^*$.*

As a remark, $\epsilon_0$ is (up to a scalar factor) the minimum performance improvement by increasing $\psi \to 2\psi$, and the minimum of slope of $j$ as a function of $\log\psi$ for $\psi \in [1, C^*]$. Therefore, when Adaptive-DPDL exits at some step $K$, the improvement that can be achieved by increase $\psi$ grows as at most $\frac{\epsilon_0}{\psi_K}$. If in this case $\psi_K$ is still far small from $C^*$, then the difficulty essentially comes from a prohibitively large $C^*$.

**Sample complexity of Adaptive-DPDL** At step $K$, the samples needed for DPDL are $\tilde{\mathcal{O}}\left(\frac{\mathcal{N}\psi_K}{\varphi^2(1-\gamma)^4\epsilon^2}\right)$, and the samples needed for verification are $\tilde{\mathcal{O}}\left(\frac{|\mathcal{S}|\psi_K}{(1-\gamma)^4\epsilon^2}\right)$. There are at most

$\lceil \log_2(C^*/\psi^1) \rceil + 1$ outer steps and the $\psi_K$ is twofold at each step, thus the total samples needed are $\tilde{\mathcal{O}}\left(\frac{\mathcal{N}\psi_K}{\varphi^2(1-\gamma)^4\epsilon^2}\right)$ if it exits at step $K$. Especially, as long as $\epsilon \leq \epsilon_0(\mathcal{M})$, Adaptive-DPDL ends after consuming $\tilde{\mathcal{O}}\left(\frac{\mathcal{N}C^*}{\varphi^2(1-\gamma)^4\epsilon^2}\right)$ samples and outputs a policy which is safe and $\mathcal{O}(\epsilon)$-optimal.

## G.3 Proof of Proposition G.1

*Proof.* First, we provide the following lemma for the estimators $\widehat{\Delta}_p$, $\widehat{J}(\overline{\pi})$ and $\widehat{J}^{u^\kappa}(\overline{\pi})$. The calculation of Lemma G.3 is very closed to Appendix B, and is thus omitted.

**Lemma G.3.** *Suppose that $N_v$ and $\epsilon_{ver}$ are chosen according to Proposition G.1. Denote $\overline{\nu} = W\overline{x}$, then with probability at least $1 - \delta/3$, we have*

$$\max\left\{\left\|\widehat{\Delta}_p - (A^\top\overline{\nu} - \rho_0)\right\|_1, \left|\widehat{J}(\overline{\pi}) - \langle r, \overline{\nu}\rangle\right|, \left\|\widehat{J}^{u^\kappa}(\overline{\pi}) - U_\kappa\overline{\nu}\right\|_\infty\right\} \leq \varphi(1-\gamma)\epsilon_{ver}.$$

**Proof of the case VERIFY$(\overline{x})$ = FALSE.** By Corollary E.5, it holds that when $\psi \geq C^*$,

$$\|A^\top\overline{\nu} - \rho_0\|_1 \leq \frac{11}{8}\varphi(1-\gamma)\epsilon, \quad \text{and} \quad \|[U_\kappa\overline{\nu}]_-\|_\infty \leq \frac{11}{4}\varphi\epsilon.$$

Combining the above inequality with Lemma G.3 indicates that $\|\widehat{\Delta}_p\|_1 \leq \frac{3}{2}\varphi(1-\gamma)\epsilon$ and $\|[\widehat{J}^{u^\kappa}(\overline{\pi})]_-\|_\infty \leq 3\varphi\epsilon$. This contradicts the condition for returning FALSE. Therefore, we know that $\psi < C^*$.

**Proof of the case VERIFY$(\overline{x})$ = TRUE.** By the condition for returning TRUE, we know $\|\widehat{\Delta}_p\|_1 \leq \frac{3}{2}\varphi(1-\gamma)\epsilon$ and $\|[\widehat{J}^{u^\kappa}(\overline{\pi})]_-\|_\infty \leq 3\varphi\epsilon$. Together with Lemma G.3, we have

$$\|A^\top\overline{\nu} - \rho_0\|_1 \leq 1.6\varphi(1-\gamma)\epsilon \quad \text{and} \quad \left\|[U_\kappa\overline{\nu}]_-\right\|_\infty \leq 3.1\varphi\epsilon.$$

Similar to our analysis in Appendix E, we write $\nu^{\overline{\pi}}$ the true visitation measure of $\overline{\pi}$. Then by (34),

$$\left\|\nu^{\overline{\nu}} - \overline{\nu}\right\|_1 \leq \frac{1}{1-\gamma}\left(\left\|A^\top\overline{\nu} - \rho_0\right\|_1 + 3\left\|\overline{x} - \overline{\nu}\right\|_1\right) \leq 1.63\varphi\epsilon.$$

where the term $\|\overline{x} - \overline{\nu}\|_1$ is controlled by Proposition 4.3. Due to the fact that $\left|\left\|[U_\kappa\nu^{\overline{\pi}}]_-\right\|_\infty - \left\|[U_\kappa\overline{\nu}]_-\right\|_\infty\right| \leq (1 + 5(1-\gamma)\varphi\epsilon)\left\|\nu^{\overline{\pi}} - \overline{\nu}\right\|_1 \leq 1.1\left\|\nu^{\overline{\pi}} - \overline{\nu}\right\|_1$ for small $\epsilon \leq \frac{1}{50(1-\gamma)}$, it holds that

$$\min_i J_i^u(\overline{\pi}) \geq \kappa - \left\|[U_\kappa\nu^{\overline{\pi}}]_-\right\|_\infty \geq \kappa - \left\|[U_\kappa\overline{\nu}]_-\right\|_\infty - 1.1\left\|\nu^{\overline{\pi}} - \overline{\nu}\right\|_1 \geq 0. \quad (47)$$

Moreover, the definition of Gap$(\overline{x})$ gives

$$j(\psi) - \text{Gap}(\overline{x}) = \langle r, \overline{\nu}\rangle - R_\mathcal{V}\left\|A^\top\overline{\nu} - \rho_0\right\|_1 - R_\Lambda\left\|[U_\kappa\overline{\nu}]_-\right\|_\infty \leq j(\psi) \leq j_0(\psi),$$

which yields

$$\langle r, \overline{\nu}\rangle \leq j_0(\psi) + R_\mathcal{V}\left\|A^\top\overline{\nu} - \rho_0\right\|_1 + R_\Lambda\left\|[U_\kappa\overline{\nu}]_-\right\|_\infty \leq j_0(\psi) + 100\epsilon,$$

$$\langle r, \overline{\nu}\rangle \geq j(\psi) - \text{Gap}(\overline{x}) \geq j_0(\psi) - 400\epsilon,$$

where we use the fact that $0 \leq j_0(\psi) - j_\kappa(\psi) \leq \frac{64\kappa}{\varphi} = 320\epsilon$. The same bound for $\widehat{J}(\overline{\pi})$ can be derived by Lemma G.3. $\qed$

## G.4 Proof of Lemma 6.1

*Proof.* First we show that, when $\psi < C^*$, $j_0(\psi) < J^*$. Otherwise, for $x_* = \arg\max_{x\in\mathcal{X}}\mathcal{J}_0(x)$, it holds that $\mathcal{J}_0(x_*) = j_0(\psi) \geq J^*$, i.e., for $\nu_* = Wx_*$,

$$J^* - \langle r, \nu_*\rangle + R_\mathcal{V}\left\|A^\top\nu_* - \rho_0\right\|_1 + R_\Lambda\left\|[U\nu_*]_-\right\|_\infty \leq 0.$$

Applying Lemma E.3 gives $\left\|A^\top\nu_* - \rho_0\right\|_1 \leq 0$, $\left\|[U\nu_*]_-\right\|_\infty \leq 0$, $J^* - \langle r, \nu_*\rangle \leq 0$. Thus, $\nu \in \mathfrak{V} \cap \mathfrak{S}$, and $\langle r, \nu_*\rangle \geq J^*$, which imply that $\nu_*$ is indeed an optimal solution of problem (4). However, $\nu_* \in W\mathcal{X} \Rightarrow \nu_* \in \mathfrak{V}(\psi) \Rightarrow \psi \geq C^*$, a contradiction.

Now the monotonicity is easy. We still fix an optimal $\nu_* \in \mathfrak{V}(C^*)$ and let $x_* := W^{-1}\nu_*$. For $1 \leq \psi' < \psi$, we write $x_{\psi'} = \arg\max_{x\in\mathcal{X}(\psi')}\mathcal{J}_0(x)$, $c = \frac{\psi-\psi'}{C^*-\psi'}$, and we consider $x_\psi := cx_* + (1-c)x_{\psi'} \in \mathcal{X}(\psi)$. It holds that

$$j_0(\psi) \geq \mathcal{J}_0(x_\psi) \geq (1-c)\mathcal{J}_0(x_{\psi'}) + c\mathcal{J}_0(x_*) = j_0(\psi') + c(J^* - j_0(\psi)).$$

The proof is completed by reorganizing the above inequality. $\qed$

## G.5 Proof of Proposition G.2

*Proof.* By Proposition G.1, if $\psi_K \geq 2C^*$, then $\psi_{K-1} \geq C^*$. By Proposition G.1, with probability at least $1 - \delta$ it holds that $\text{VERIFY}(x^{(K)}) = \text{VERIFY}(x^{(K-1)}) = \text{TRUE}$, and

$$\widehat{J}(\overline{\pi}^{(K)}), \widehat{J}(\overline{\pi}^{(K-1)}) \in [J(\pi^*) - 400\epsilon, J(\pi^*) + 100\epsilon],$$

$$\Rightarrow \left|\widehat{J}(\overline{\pi}^{(K)}) - \widehat{J}(\overline{\pi}^{(K-1)})\right| \leq 500\epsilon,$$

where we use the fact $j_0(\psi_K) = j_0(\psi_{K-1}) = j_0(C^*) = J(\pi^*)$ from Lemma 6.1. Therefore, if $\psi_K \geq 2C^*$, Adaptive-DPDL must exit at step $K$.

Now, we only need to consider the case that Adaptive-DPDL ends at some step $K$, but $\psi_K$ might not be greater than $C^*$. Because $\text{VERIFY}(x^{(K)}) = \text{VERIFY}(x^{(K-1)}) = \text{TRUE}$, we combine the exit condition (46) with Proposition G.1 and derive $|j_0(\psi_K) - j_0(\psi_{K-1})| \leq 1000\epsilon$. Then by Lemma 6.1, we have

$$J(\pi^*) - j_0(\psi_K) \leq \frac{2(C^* - \psi_K)}{\psi_K}\left(j_0(\psi_K) - j_0(\psi_{K-1})\right).$$

Thus $J(\pi^*) - J(\pi^{(K)}) \lesssim \frac{C^*}{\psi_K}\epsilon$. Furthermore, we can define the following quantity

$$\epsilon_0 := \min_{1 \leq \psi \leq C^*}\left(j_0(\psi) - j_0(\frac{\psi}{2})\right) > 0.$$

Here $\epsilon_0 > 0$ is due to Lemma 6.1. If $j_0(\psi) - j(\frac{\psi}{2}) < \epsilon_0$ for some $\psi \geq 1$, then immediately we have $\psi \geq C^*$. If $\epsilon' = 15\epsilon \leq \epsilon_0/100 =: \epsilon_0(\mathcal{M})$, Adaptive-DPDL must exit at step $K$ with $C^* \leq \psi_K \leq 4C^*$. By Theorem 4.1, the output policy $\pi^{(K)}$ is safe and $J(\pi^*) - J(\pi^{(K)}) \leq \epsilon'$. □

# H Convergence Analysis in Asynchronous Setting

## H.1 Mixing property of Markov chain

Under the setting of the asynchronous learning (Assumption 4.5), we can observe a sequence of state-action trajectory generated under the behavioral policy $\pi_b$, namely

$$s_1, a_1, s_2, a_2, s_3, \cdots, s_n, a_n, s_{n+1}, \cdots.$$

This sequence can be naturally viewed as a Markov chain $(X_t)_{t \geq 1}$ where $X_t = (s_t)$, plus a marginal component $a_t \in \mathcal{A}$. In the asynchronous setting, the reference distribution $\mu$ is the stationary distribution $\mu_{\pi_b}$ of this chain product with the policy $\pi_b$. As in the synchronous setting, we denote $\mathcal{F}_t$ for all the history information at time $t$. Actually, by the Markov property and our update rule, conditioning on $\mathcal{F}_t$ is equivalent to conditioning on $s_t, Z^t$. According to [13, Section 4], we define the mixing time of this Markov chain as

$$\begin{cases} \mathcal{E}(t) := \sup_{s \in \mathcal{S}} d_{\text{TV}}\left(\mu_{\pi_b}, \mathbb{P}^t_{\pi_b}(\cdot|s_0 = s)\right), \\ t_{\text{mix}} := \min\{t : \mathcal{E}(t) \leq \frac{1}{4}\}, \end{cases} \tag{48}$$

where $\mathbb{P}^t_{\pi_b}(\cdot|s_0 = s)$ denotes the distribution of $s_t$ given $s_0 = s$ and policy $\pi_b$. By [13, Remark 4.12], it holds that

$$\mathcal{E}(t) \leq 2^{-\left\lfloor \frac{t}{t_{\text{mix}}} \right\rfloor}.$$

Given the concept of the mixing time, we modify the standard Bernstein inequality for Markov chain [10, 19, etc.] to cover the non-stationary Markov chains.

**Proposition H.1.** *Suppose that $(X_t)_{t \geq 1}$ is a Markov chain with invariant distribution $\pi$ and mixing time $t_{\text{mix}} < +\infty$. Let $f$ be a measurable function such that $\mathbb{E}_\pi[f(X)] = 0$, $|f(X)| \leq M$. Denote $\sigma^2 = \mathbb{E}_\pi\left[f(X)^2\right]$, then for $\delta \in (0, 1)$, the following holds with probability at least $1 - \delta$*

$$\left|\sum_{t=1}^n f(X_t)\right| \leq \sqrt{32 t_{\text{mix}} n \sigma^2 \log \frac{4}{\delta}} + 82 t_{\text{mix}} M \log \frac{4}{\delta}.$$

The difficulty of analyzing Markovian gradients is the correlation between updates and samples. As demonstrated in Section 4.3, in our analysis, we leverage the fact that $s_{t+\tau}$ is a sample "almost" from $\mu_{\pi_b}$ and "almost" independent of $s_t$, as long as $\tau \geq t_{\mathrm{mix}} \cdot \log$ factor. We further demonstrate this idea in the following proposition, by comparing $\mathbb{E}\left[\widehat{g}(\cdot; \zeta_{t+\tau}) | \mathcal{F}_t\right]$ and $\mathcal{G}(Z^t)$.

**Proposition H.2** (Almost unbiased). *For a $\mathcal{F}_t$-measurable random variable $Z \in \mathcal{Z} := \mathcal{V} \times \Lambda \times \mathcal{X}$, it holds that*

$$\left\| \mathbb{E}\left[\widehat{g}_V(Z; \zeta_{t+\tau}) | \mathcal{F}_t\right] - \nabla_V \mathcal{L}_w(Z) \right\|_1 \leq \frac{2\psi}{1-\gamma} \mathcal{E}(\tau),$$

$$\left\| \mathbb{E}\left[\widehat{g}_\lambda(Z; \zeta_{t+\tau}) | \mathcal{F}_t\right] - \nabla_\lambda \mathcal{L}_w(Z) \right\|_\infty \leq \frac{2\psi}{1-\gamma} \mathcal{E}(\tau),$$

$$\left\| \mathbb{E}\left[\widehat{g}_x(Z; \zeta_{t+\tau}) | \mathcal{F}_t\right] - \nabla_x \mathcal{L}_w(Z) \right\|_\infty \leq \frac{64}{\varphi(1-\gamma)\varsigma} \mathcal{E}(\tau).$$

*Furthermore, for any $Z' \in \mathcal{Z}$, we have*

$$\left| \langle Z', \mathcal{G}(Z) - \mathbb{E}\left[\widehat{g}(Z; \zeta_{t+\tau}) | \mathcal{F}_t\right] \rangle \right| \leq \frac{128\psi}{\varphi(1-\gamma)^2} \mathcal{E}(\tau).$$

The following proposition indicates that, the estimator $\widehat{g}(\cdot; \zeta_{t+\tau})$ is not only "nearly unbiased" conditional on $\mathcal{F}_t$, but it also has a well bounded moment.

**Proposition H.3** (Bounded moment). *For any $\mathcal{F}_t$-measurable random variable $Z \in \mathcal{Z}$, it holds that*

$$\mathbb{E}\left[ \|\widehat{g}_V(Z; \zeta_{t+\tau})\| | \mathcal{F}_t \right] \lesssim \frac{C(\tau)}{1-\gamma},$$

$$\mathbb{E}\left[ \|\widehat{g}_\lambda(Z; \zeta_{t+\tau})\|_\infty | \mathcal{F}_t \right] \lesssim \frac{C(\tau)}{1-\gamma},$$

$$\mathbb{E}\left[ \|\widehat{g}_x(Z; \zeta_{t+\tau})\|_{x^t}^2 \Big| \mathcal{F}_t \right] \lesssim \frac{C(\tau)\mathcal{N}\psi}{\varphi^2(1-\gamma)^3},$$

*where $C(\tau) = 2 + \frac{\mathcal{E}(\tau)}{\varsigma}$.*

Therefore, there is a universal constant $c_\tau$ such that for $\tau \geq \lfloor c_\tau t_{\mathrm{mix}} \iota \rfloor$, we have $C(\tau) \leq 3$ and $\mathcal{E}(\tau) \leq \frac{1}{T}$ (the log factor $\iota$ and the range of $T$ are specified in Theorem H.5). We denote $\tau_0 = \lfloor c_\tau t_{\mathrm{mix}} \iota \rfloor$.

## H.2   Proof sketch of Theorem 4.6

Before our analysis of DPDL on $\mathcal{D}_{async}$, we have to first provide an analogue of Proposition 4.3. As in the synchronous setting, we set $\epsilon_e = \frac{\epsilon}{100}$ and $\varsigma = \frac{\varphi(1-\gamma)^2 \epsilon_e}{2\mathcal{N}\psi}$.

**Proposition H.4.** *Given $N_e \geq c_e' \frac{t_{\mathrm{mix}} \mathcal{N}\psi\iota}{\varphi^2(1-\gamma)^4 \epsilon_e^2}$ samples from a trajectory generated by $\pi_b$, the $\hat{\mu}$ constructed in (10) satisfies the following properties with probability at least $1 - \delta/3$.*
*(1) For all $s, a$, $\frac{\mu(s,a)}{\hat{\mu}(s,a)} \leq 2$, and $\hat{\mu}(s,a) \geq \varsigma$.*
*(2) For any $\pi \in \Pi(\psi)$, $W^{-1}\nu^\pi \in \mathcal{X}$.*
*(3) For any $x \in \mathcal{X}$, $\|Wx - x\|_1 \leq \varphi(1-\gamma)\epsilon_e$.*

Now, we present the convergence guarantee of the duality gap $\mathrm{Gap}(\overline{x})$.

**Theorem H.5.** *Given $\epsilon \in \left(0, \frac{1}{1-\gamma}\right]$, $\delta \in \left(0, \frac{1}{2}\right)$, we denote $\iota = \log\left(T|\mathcal{S}||\mathcal{A}|I/\delta\right)$. Then as long as $T \gtrsim \frac{\tau_0^2 \mathcal{N}\psi\iota}{\varphi^2(1-\gamma)^4 \epsilon_e^2}$, with probability at least $1 - \delta/3$ it holds*

$$\mathrm{Gap}(\overline{x}) \lesssim \frac{t_{\mathrm{mix}}}{\varphi(1-\gamma)^2} \sqrt{\frac{\mathcal{N}\psi\iota^3}{T}} \leq \epsilon.$$

Therefore, there is a universal constant $c_o'$ such that $\mathrm{Gap}(\overline{x}) \leq \frac{\epsilon}{2}$ as long as $T \geq c_o' \frac{t_{\mathrm{mix}}^2 \mathcal{N}\psi\iota^3}{\varphi^2(1-\gamma)^4 \epsilon^2}$. Then the proof in Appendix E can be applied directly. In conclusion, the number of samples needed is

$$\tilde{\mathcal{O}}\left( \frac{t_{\mathrm{mix}}^2 \mathcal{N}\psi}{\varphi^2(1-\gamma)^4 \epsilon^2} \right).$$

We sketch the proof of Theorem H.5 as follows. The detailed proofs of propositions are organized by order in the rest of this section.

**Decomposition of duality gap** We define the auxiliary variables $V', \lambda', x'$ as in Appendix D,

$$(V', \lambda') = \arg\min_{V \in \mathcal{V}, \lambda \in \Lambda} \mathcal{L}_w(V, \lambda, \overline{x}), \quad x' = \arg\max_{x \in \mathcal{X}} \min_{V \in \mathcal{V}, \lambda \in \Lambda} \mathcal{L}_w(V, \lambda, x), \quad Z' = [V'; \lambda'; x'].$$

Recall the decomposition (24), we have

$$\mathrm{Gap}(\overline{x}) = \underbrace{\frac{1}{T} \sum_{t=1}^{T} \langle \widehat{g}(Z^t; \zeta_t), Z^t - Z' \rangle}_{S_1} + \underbrace{\frac{1}{T} \sum_{t=1}^{T} \langle \mathcal{G}(Z^t) - \widehat{g}(Z^t; \zeta_t), Z^t - Z' \rangle}_{S_2}.$$

**Bounding the term $S_1$** The proof in Appendix D.2 can be applied without change. Namely, as long as $\eta \leq \frac{1}{2} \min\left(\frac{\alpha_\lambda}{M_\lambda}, \frac{\alpha_x}{M_{x,\infty}}\right)$, it holds that

$$S_1 \lesssim \frac{\alpha_V D_V^2 + \alpha_\lambda D_\lambda + \alpha_x D_x}{\eta T} + \frac{\eta}{T} \sum_{t=1}^{T} \left( \frac{\|\widehat{g}_V(Z^t; \zeta_t)\|^2}{\alpha_V} + \frac{D_{\lambda,1} \|\widehat{g}_\lambda(Z^t; \zeta_t)\|_\infty^2}{\alpha_\lambda} + \frac{\|\widehat{g}_x(Z^t; \zeta_t)\|_{x^t}^2}{\alpha_x} \right).$$

**Bounding the term $S_2$** In the asynchronous setting, $\zeta_1, \cdots, \zeta_T$ are no longer i.i.d samples. To deal with this issue, let us consider the following decomposition

$$\Gamma^t := \langle \mathcal{G}(Z^t) - \widehat{g}(Z^t; \zeta_t), Z^t - Z' \rangle = \underbrace{\langle \mathcal{G}(Z^t), Z^t - Z' \rangle - \langle \mathcal{G}(Z^{t-\tau}), Z^{t-\tau} - Z' \rangle}_{\Gamma_1^t}$$

$$+ \underbrace{\langle \mathcal{G}(Z^{t-\tau}) - \mathbb{E}\left[\widehat{g}(Z^{t-\tau}; \zeta_t) \,\middle|\, \mathcal{F}_{t-\tau}\right], Z^{t-\tau} - Z' \rangle}_{\Gamma_2^{t-\tau}}$$

$$+ \underbrace{\langle \mathbb{E}\left[\widehat{g}(Z^{t-\tau}; \zeta_t) \,\middle|\, \mathcal{F}_{t-\tau}\right] - \widehat{g}(Z^{t-\tau}; \zeta_t), Z^{t-\tau} - Z' \rangle}_{\Gamma_3^{t-\tau}}$$

$$+ \underbrace{\langle \widehat{g}(Z^{t-\tau}; \zeta_t), Z^{t-\tau} - Z' \rangle - \langle \widehat{g}(Z^t; \zeta_t), Z^t - Z' \rangle}_{\Gamma_4^t},$$

where $1 \leq \tau \leq \tau_0$ is a fixed integer. The quantity $\Gamma_2^{t-\tau}$ can be bounded by Proposition H.2, and $\Gamma_3^{t-\tau}$ can be bounded as in Appendix D.3. As of $\Gamma_1^t, \Gamma_4^t$, we bound it in terms of $Z^t - Z^{t-\tau}$. In conclusion, with probability at least $1 - \delta/5$, we have

$$\frac{1}{T} \sum_{t=1}^{T} \Gamma^t \lesssim \frac{\psi}{\varphi(1-\gamma)^2} \mathcal{E}(\tau) + \frac{1}{\varphi(1-\gamma)^2} \sqrt{\frac{\tau C(\tau) \mathcal{N} \psi \iota}{T}} + \frac{1}{\varphi(1-\gamma)} \sum_{t=\tau+1}^{T} \frac{|x^t - x^{t-\tau}|(s_t, a_t)}{\hat{\mu}(s_t, a_t)}$$

$$+ \frac{\eta}{T} \sum_{t=\tau+1}^{T} \left( \frac{\|\widehat{g}_V(Z^t; \zeta_t)\|^2}{\alpha_V} + \frac{D_{\lambda,1} \|\widehat{g}_\lambda(Z^t; \zeta_t)\|_\infty^2}{\alpha_\lambda} \right). \tag{49}$$

The detailed analysis is presented in Appendix H.7.

**Bounding the variance and magnitude of the updates** It remains to bound $\|\widehat{g}_V(Z^t; \zeta_t)\|$, $\|\widehat{g}_\lambda(Z^t; \zeta_t)\|_\infty$, $\|\widehat{g}_x(Z^t; \zeta_t)\|_{x^t}$, and the term $|x^t(s_t, a_t) - x^{t-\tau}(s_t, a_t)|$. For any $x \in \mathbb{R}_{\geq 0}^{|\mathcal{S}||\mathcal{A}|}$, and any $(s, a) \in \mathcal{S} \times \mathcal{A}$, we introduce the following abbreviation for the ease of notation

$$p(x; s, a) := \frac{x(s, a)}{\hat{\mu}(s, a)}, \quad q(x; s, a) := \frac{x(s, a)}{\hat{\mu}(s, a)^2}.$$

For any sample $\zeta = (s_0, s, a, s', r, \mathbf{u})$, we also reload the notations $p, q$ as $p(x; \zeta) := p(x; s, a)$ and $q(x; \zeta) := q(x; s, a)$.

It is not hard to see that $p(x^t; \zeta_t)$ and $q(x^t; \zeta_t)$ dominate the variance of the gradient estimators (for detailed discussion, see Appendix H.5). More specifically, we have

$$\left\|\widehat{g}_V(Z^t; \zeta_t)\right\| \lesssim p(x^t; \zeta_t), \quad \left\|\widehat{g}_\lambda(Z^t; \zeta_t)\right\|_\infty \lesssim p(x^t; \zeta_t),$$

$$\left\|\widehat{g}_x(Z^t; \zeta_t)\right\|_{x^t} \lesssim \frac{1}{\varphi(1-\gamma)} \sqrt{q(x^t; \zeta_t)}.$$

Then, we only need to bound $\sum_{t=1}^{T} q(x^t; \zeta_t)$, $\sum_{t=\tau+1}^{T} p(|x^t - x^{t-\tau}|; \zeta_t)$ and $\sum_{t=1}^{T} p(x^t; \zeta_t)^2$. By leveraging the idea of the decomposition (13), we can derive the desired estimation, as follows.

**Proposition H.6.** *There is a universal constant $c$ such that for $T \geq c \frac{\tau_0^2 \mathcal{N} \psi \iota}{\varphi^2 (1-\gamma)^4 \epsilon_e^2}$, the following holds for all $1 \leq \tau \leq \tau_0$ simultaneously, with probability at least $1 - \delta/10$:*

$$\frac{1}{T} \sum_{t=1}^{T} p(x^t; \zeta_t)^2 \lesssim \frac{\psi}{(1-\gamma)^2},$$

$$\frac{1}{T} \sum_{t=1}^{T} q(x^t; \zeta_t) \lesssim \frac{\mathcal{N} \psi}{1-\gamma},$$

$$\frac{1}{T} \sum_{t=\tau+1}^{T} p(|x^t - x^{t-\tau}|; \zeta_t) \lesssim \frac{\tau C(\tau)}{1-\gamma} \sqrt{\frac{\mathcal{N} \psi \iota}{T}}.$$

**Conclusion** Combining Proposition H.6 with the estimations of $S_1$ and $S_2$, we have with probability at least $1 - \delta/3$,

$$\mathrm{Gap}(\overline{x}) \lesssim \frac{\tau C(\tau)}{\varphi(1-\gamma)^2} \sqrt{\frac{\mathcal{N} \psi \iota}{T}} + \frac{\psi}{\varphi(1-\gamma)^2} \mathcal{E}(\tau). \tag{50}$$

Now, we can take $\tau = \tau_0 = \lfloor c_\tau t_{\mathrm{mix}} \iota \rfloor$. Then by the definition, it holds $C(\tau_0) \leq 3$ and $\epsilon(\tau_0) \leq \frac{1}{T}$, and hence with probability at least $1 - \delta/3$ we have

$$\mathrm{Gap}(\overline{x}) \lesssim \frac{t_{\mathrm{mix}}}{\varphi(1-\gamma)^2} \sqrt{\frac{\mathcal{N} \psi \iota^3}{T}}.$$

As a remark, if we have an (empirical) estimation $\hat{t}_{\mathrm{mix}}$ such that $\hat{t}_{\mathrm{mix}} \geq t_{\mathrm{mix}}$, then by taking $\eta = \frac{1}{\sqrt{\hat{t}_{\mathrm{mix}} T}}$, the final bound can be improved to $\mathrm{Gap}(\overline{x}) \lesssim \frac{1}{\varphi(1-\gamma)^2} \sqrt{\frac{\hat{t}_{\mathrm{mix}} \mathcal{N} \psi \iota^3}{T}}$, as long as $T \gtrsim \frac{t_{\mathrm{mix}}^2}{\hat{t}_{\mathrm{mix}}} \frac{\mathcal{N} \psi \iota^3}{\varphi^2 (1-\gamma)^4 \epsilon^2}$.

## H.3 Proof of Proposition H.1

In order to prove Proposition H.1, we invoke the following standard version of the Bernstein's inequality. We also leverage the idea of the proof of [14, Lemma 8].

**Theorem H.7** ([19, Theorem 3.9]). *Suppose $\{X_i\}_{i \geq 1}$ is a stationary Markov chain with invariant distribution $\pi$ and pseudo spectral gap $\gamma_{\mathrm{ps}}$. Let $f$ be a measurable function such that $\mathbb{E}_\pi [f(X)] = 0$, $|f(X)| \leq M$. Denote $\sigma^2 = \mathbb{E}_\pi [f(X)^2]$, then for all $x \geq 0$,*

$$\mathbb{P} \left( \left| \sum_{i=1}^{n} f(X_i) \right| \geq x \right) \leq 2 \exp \left( -\frac{x^2 \cdot \gamma_{\mathrm{ps}}}{8(n + 1/\gamma_{\mathrm{ps}}) \sigma^2 + 20 x M} \right).$$

*In particular, for uniformly ergodic chains with mixing time $t_{\mathrm{mix}}$, $\gamma_{\mathrm{ps}} \geq \frac{1}{2 t_{\mathrm{mix}}}$.*

*Proof of Proposition H.1.* Without loss of generality, we assume the Markov chain $(X_t)$ has a finite state space $\mathcal{X}$. We fix integer $\tau$ and $x \geq 0$ to be specified later, and let $\pi_n$ be the distribution of $X_n$. Theorem H.7 yields

$$\mathbb{P} \left( \left| \sum_{i=\tau+1}^{n} f(X_i) \right| \geq x \,\bigg|\, X_1 \sim \pi \right) \leq 2 \exp \left( -\frac{x^2}{16 t_{\mathrm{mix}} (n + 2 t_{\mathrm{mix}} - \tau) \sigma^2 + 40 x t_{\mathrm{mix}} M} \right).$$

Let $\mathcal{B}_\tau$ be the event $\{ |\sum_{i=\tau+1}^{n} f(X_i)| \geq x \}$, then

$$|\mathbb{P}(\mathcal{B}_\tau | X_1 \sim \pi) - \mathbb{P}(\mathcal{B}_\tau | X_1 \sim \pi_1)|$$

$$= \left| \sum_{x \in \mathcal{X}} \mathbb{P}(\mathcal{B}_\tau | X_{\tau+1} = x) (\mathbb{P}(X_{\tau+1} = x | X_1 \sim \pi) - \mathbb{P}(X_{\tau+1} = x | X_1 \sim \pi_1)) \right|$$

$$= \left| \sum_{x \in \mathcal{X}} \mathbb{P}\left(\mathcal{B}_\tau \middle| X_{\tau+1} = x\right) \left(\pi(x) - \pi_{\tau+1}(x)\right) \right|$$

$$\leq \max\left( \left\| [\pi - \pi_{\tau+1}]_+ \right\|_1, \left\| [\pi - \pi_{\tau+1}]_- \right\|_1 \right)$$

$$= d_{\mathrm{TV}}(\pi, \pi_{\tau+1}) \leq \mathcal{E}(\tau).$$

Therefore, we can take $x = \sqrt{32 t_{\mathrm{mix}}(n - \tau + 2 t_{\mathrm{mix}}) \log \frac{4}{\delta}} + 80 t_{\mathrm{mix}} M \log \frac{4}{\delta}$ and $\tau = \left\lceil \log_2 \frac{2}{\delta} \right\rceil t_{\mathrm{mix}}$, then

$$\mathbb{P}\left(\mathcal{B}_\tau \middle| X_1 \sim \pi_1\right) \leq \mathcal{E}(\tau) + \mathbb{P}\left(\mathcal{B}_\tau \middle| X_1 \sim \pi\right) \leq \frac{\delta}{2} + \frac{\delta}{2} = \delta.$$

Hence with probability at least $1 - \delta$, it holds that

$$\left| \sum_{i=\tau+1}^n f(X_i) \right| \leq \sqrt{32 t_{\mathrm{mix}}(n - \tau + 2 t_{\mathrm{mix}}) \log \frac{4}{\delta}} + 80 t_{\mathrm{mix}} M \log \frac{4}{\delta}.$$

The proof is completed by noticing that $\left| \sum_{i=1}^\tau f(X_i) \right| \leq \tau M \leq 2 t_{\mathrm{mix}} M \log \frac{4}{\delta}$ and $\tau \geq 2 t_{\mathrm{mix}}$.   □

## H.4   Proof of Proposition H.2

*Proof.* Recall that the gradient estimators are constructed as

$$\widehat{g}_V(Z; \zeta) := \mathbb{I}_{s_0} + \frac{x(s,a)}{\hat{\mu}(s,a)}\left(\gamma \mathbb{I}_{s'} - \mathbb{I}_s\right),$$

$$\widehat{g}_\lambda(Z; \zeta) := \frac{x(s,a)}{\hat{\mu}(s,a)}\mathbf{u}^\kappa,$$

$$\widehat{g}_x(Z; \zeta) := \frac{r + \gamma V(s) - V(s') + \langle \mathbf{u}^\kappa, \lambda \rangle}{\hat{\mu}(s,a)}\mathbb{I}_{s,a}.$$

Therefore, for $Z = [V; \lambda; x]$ that is $\mathcal{F}_t$ measurable, we have

$$\mathbb{E}\left[\widehat{g}_x(Z; \zeta_{t+\tau}) \middle| \mathcal{F}_t\right]$$

$$= \mathbb{E}\left[ \frac{r_{t+\tau} + \gamma V(s_{t+\tau}) - V(s_{t+\tau+1}) + \langle \mathbf{u}_{t+\tau}^\kappa, \lambda \rangle}{\hat{\mu}(s_{t+\tau}, a_{t+\tau})}\mathbb{I}_{s_{t+\tau}, a_{t+\tau}} \middle| s_t, Z \right]$$

$$= \mathbb{E}\left[ \frac{r(s_{t+\tau}, a_{t+\tau}) + \gamma V(s_{t+\tau}) - V(s_{t+\tau+1}) + \langle \mathbf{u}^\kappa(s_{t+\tau}, a_{t+\tau}), \lambda \rangle}{\hat{\mu}(s_{t+\tau}, a_{t+\tau})}\mathbb{I}_{s_{t+\tau}, a_{t+\tau}} \middle| s_t, Z \right]$$

$$= \sum_{s,a,s'} \mathbb{P}\left(s_{t+\tau} = s, a_{t+\tau} = a, s_{t+\tau+1} = s' \middle| s_t\right) \frac{r(s,a) + \gamma V(s) - V(s') + \langle \mathbf{u}^\kappa(s,a), \lambda \rangle}{\hat{\mu}(s,a)}\mathbb{I}_{s,a}$$

$$= \sum_{s,a} \frac{\mathbb{P}\left(s_{t+\tau} = s, a_{t+\tau} = a \middle| s_t\right)}{\hat{\mu}(s,a)} \left( r(s,a) + \gamma V(s) - \mathbb{E}_{s'|s,a}\left[V(s')\right] + \langle \mathbf{u}^\kappa(s,a), \lambda \rangle \right) \mathbb{I}_{s,a}.$$

For the sake of simplicity, we denote

$$W^{\tau, s_t} := \mathrm{diag}\left( \frac{\mathbb{P}_{\pi_b}\left(s_{t+\tau} = s, a_{t+\tau} = a \middle| s_t\right)}{\hat{\mu}(s,a)} \right) = \mathrm{diag}\left( \frac{\mathbb{P}_{\pi_b}\left(s_{t+\tau} = s \middle| s_t\right)\pi_b(a|s)}{\hat{\mu}(s,a)} \right)_{s,a}, \quad (51)$$

and we follow the matrix notation introduced in Appendix E.1. Then

$$\mathbb{E}\left[\widehat{g}_x(Z; \zeta_{t+\tau}) \middle| \mathcal{F}_t\right] = W^{\tau, s_t}\left(r - AV + U_\kappa^\mathrm{T} \lambda\right),$$

$$\mathbb{E}\left[\widehat{g}_V(Z; \zeta_{t+\tau}) \middle| \mathcal{F}_t\right] = \mathbb{E}\left[ \mathbb{I}_{s_0} + \frac{x(s_{t+\tau}, a_{t+\tau})}{\hat{\mu}(s_{t+\tau}, a_{t+\tau})}\left(\gamma \mathbb{I}_{s_{t+\tau+1}} - \mathbb{I}_{s_{t+\tau}}\right) \middle| s_t, Z \right]$$

$$= \rho_0 + \sum_{s,a} \mathbb{P}_{\pi_b}\left(s_{t+\tau} = s, a_{t+\tau} = a \middle| s_t\right) \frac{x(s,a)}{\hat{\mu}(s,a)}\left(\gamma \mathbb{E}_{s'|s,a}\left[\mathbb{I}_{s'}\right] - \mathbb{I}_s\right)$$

$$= \rho_0 - A^{\mathrm{T}} W^{\tau, s_t} x,$$

$$\mathbb{E}\left[\widehat{g}_\lambda(Z; \zeta_{t+\tau})\middle|\mathcal{F}_t\right] = \mathbb{E}\left[\frac{x(s_{t+\tau}, a_{t+\tau})}{\hat{\mu}(s_{t+\tau}, a_{t+\tau})} \mathbf{u}^\kappa \middle| s_t, Z\right]$$

$$= \sum_{s,a} \mathbb{P}_{\pi_b}\left(s_{t+\tau} = s, a_{t+\tau} = a\middle| s_t\right) \frac{x(s, a)}{\hat{\mu}(s, a)} \mathbf{u}^\kappa(s, a)$$

$$= U_\kappa W^{\tau, s_t} x.$$

Therefore, we have

$$\left\|\mathbb{E}\left[\widehat{g}_V(Z; \zeta_{t+\tau})\middle|\mathcal{F}_t\right] - \nabla_V \mathcal{L}_w(Z)\right\|_1 = \left\|A^{\mathrm{T}}\left(W^{\tau, s_t} - W\right)x\right\|_1 \le 2\left\|\left(W^{\tau, s_t} - W\right)x\right\|_1$$

$$\le 2\sum_{s,a} \left|\mathbb{P}_{\pi_b}\left(s_{t+\tau} = s\middle| s_t\right) - \mu_{\pi_b}(s)\right| \frac{\pi_b(a|s)x(s, a)}{\hat{\mu}(s, a)}$$

$$\le \frac{2\psi}{1 - \gamma} d_{\mathrm{TV}}\left(\mathbb{P}_{\pi_b}^\tau\left(\cdot|s_t\right), \mu_{\pi_b}\right) \le \frac{2\psi}{1 - \gamma}\mathcal{E}(\tau),$$

where $\mathbb{P}_{\pi_b}^\tau\left(\cdot|s_t\right)$ is the distribution of $s_{t+\tau}$ conditioning on $s_t$, and the last inequality is due to the definition of $\mathcal{E}(\cdot)$. Similarly, it holds that

$$\left\|\mathbb{E}\left[\widehat{g}_\lambda(Z; \zeta_{t+\tau})\middle|\mathcal{F}_t\right] - \nabla_\lambda \mathcal{L}_w(Z)\right\|_\infty \le \frac{2\psi}{1 - \gamma}\mathcal{E}(\tau),$$

$$\left\|\mathbb{E}\left[\widehat{g}_x(Z; \zeta_{t+\tau})\middle|\mathcal{F}_t\right] - \nabla_x \mathcal{L}_w(Z)\right\|_\infty \le \frac{64}{\varphi(1 - \gamma)\varsigma}\mathcal{E}(\tau).$$

Furthermore, for any $Z' = [V'; \lambda'; x'] \in \mathcal{Z}$, we have

$$\left|\langle Z', \mathcal{G}(Z) - \mathbb{E}\left[\widehat{g}(Z; \zeta_{t+\tau})\middle|\mathcal{F}_t\right]\rangle\right|$$

$$\le \|V'\|_\infty \left\|\mathbb{E}\left[\widehat{g}_V(Z; \zeta_{t+\tau})\middle|\mathcal{F}_t\right] - \nabla_V \mathcal{L}_w(Z)\right\|_1 + \|\lambda\|_1 \left\|\mathbb{E}\left[\widehat{g}_\lambda(Z; \zeta_{t+\tau})\middle|\mathcal{F}_t\right] - \nabla_\lambda \mathcal{L}_w(Z)\right\|_\infty$$

$$+ \left|\langle r - AV + U_\kappa^{\mathrm{T}}\lambda, \left(W^{\tau, s_t} - W\right)x\rangle\right|$$

$$\le \frac{128\psi}{\varphi(1 - \gamma)^2}\mathcal{E}(\tau). \qquad \square$$

### H.5 Proof of Proposition H.3

In fact, to prove Proposition H.3, let us prove a more general result stated as follows. Proposition H.3 will follow directly from the (2) and (3) of Proposition H.8. This proposition will also be useful for our later discussion. Recall that we introduce the notation $p(x; s, a) := \frac{x(s,a)}{\hat{\mu}(s,a)}$ and $q(x; s, a) := \frac{x(s,a)}{\hat{\mu}(s,a)^2}$, and the reloaded notation $p(x; \zeta) := p(x; s, a)$ and $q(x; \zeta) := q(x; s, a)$ for sample $\zeta = (s_0, s, a, s', r, \mathbf{u})$. Then the following proposition holds true.

**Proposition H.8.** *(1). For all $x \in \mathcal{X}$ and $\zeta$, it holds that*

$$p(x; \zeta) \le \frac{\psi}{1 - \gamma}, \qquad q(x; \zeta) \le \frac{1}{\varsigma}p(x; \zeta) \le \frac{\psi}{(1 - \gamma)\varsigma}.$$

*(2). For all $Z = [V; \lambda; x]$ and $\zeta$, it holds that*

$$\left\|\widehat{g}_V(Z; \zeta)\right\| \le 3p(x; \zeta), \qquad \left\|\widehat{g}_\lambda(Z; \zeta)\right\|_\infty \le 2p(x; \zeta),$$

$$\left\|\widehat{g}_x(Z; \zeta)\right\|_x \le \frac{64}{\varphi(1 - \gamma)}\sqrt{q(x; \zeta)}.$$

*(3). For $x \in \mathcal{X}$ a (possibly random) vector that is $\mathcal{F}_t$-measurable, the (asynchronous) moments of $p, q$ can be bounded as*

$$\mathbb{E}\left[p(x; \zeta_{t+\tau})\middle|\mathcal{F}_t\right] = \sum_{s,a} \frac{\mathbb{P}\left(s_{t+\tau} = s, a_{t+\tau} = a\middle| s_t\right)}{\hat{\mu}(s, a)}x(s, a) \le C(\tau)\frac{4}{1 - \gamma},$$

$$\mathbb{E}\left[q(x;\zeta_{t+\tau})|\,\mathcal{F}_t\right] = \sum_{s,a} \frac{\mathbb{P}\left(s_{t+\tau}=s, a_{t+\tau}=a|\,s_t\right)}{\hat{\mu}(s,a)} \frac{x(s,a)}{\hat{\mu}(s,a)} \le C(\tau)\frac{\mathcal{N}\psi}{1-\gamma},$$

$$\mathbb{E}\left[p(x^t;\zeta_{t+\tau})^2\big|\,\mathcal{F}_t\right] \le \frac{\psi}{1-\gamma}\mathbb{E}\left[p(x^t;\zeta_{t+\tau})\big|\,\mathcal{F}_t\right] \le C(\tau)\frac{4\psi}{(1-\gamma)^2}.$$

Since each step of this proposition can be proved by a direct computation similar to the one in Appendix C, we omit the proof for succinctness.

## H.6    Proof of Proposition H.4

Similar to the proof of Proposition 4.3, we consider $\hat{\mu}_0(s,a) = \frac{N(s,a)}{N_e}$ and the "failure event"

$$\Omega := \bigcup_{s,a}\left\{|\mu(s,a)-\hat{\mu}_0(s,a)| > \sqrt{\mu(s,a)\frac{\ell}{N_e}} + \frac{\ell}{N_e}\right\},$$

where $\ell = 100t_{\mathrm{mix}}\log\left(\frac{12|\mathcal{S}||\mathcal{A}|}{\delta}\right)$. Then by the Bernstein's inequality (Proposition H.1), it holds that

$$\mathbb{P}\left(|\mu(s,a)-\hat{\mu}_0(s,a)| > \sqrt{\mu(s,a)\frac{\ell}{N_e}} + \frac{\ell}{N_e}\right) \le \frac{\delta}{3|\mathcal{S}||\mathcal{A}|}, \quad \forall(s,a)\in\mathcal{S}\times\mathcal{A},$$

which further gives $\mathbb{P}(\Omega) \le \frac{\delta}{3}$. The proof is completed by exactly repeating the estimations in the proof of Proposition 4.3, conditioning on $\Omega^c$.

## H.7    Bounding the term $S_2$

By separately considering each term in the decomposition

$$\Gamma^t = \Gamma_1^t + \Gamma_2^{t-\tau} + \Gamma_3^{t-\tau} + \Gamma_4^t,$$

the following inequalities hold true. The detailed derivations are placed at the end of Appendix H.7.

$$\sum_{t=1}^{\tau}\Gamma^t + \sum_{t=\tau+1}^{T}\Gamma_1^t \lesssim \frac{\tau\psi}{\varphi(1-\gamma)^2}, \tag{52}$$

$$\sum_{t=1}^{T-\tau}\Gamma_3^t \lesssim \frac{1}{\varphi(1-\gamma)^2}\sqrt{T\tau C(\tau)\mathcal{N}\psi\iota} \quad \textit{with probability at least } 1-\frac{\delta}{10}, \tag{53}$$

$$|\Gamma_4^t| \lesssim \frac{1}{\varphi(1-\gamma)}\frac{|x^t - x^{t-\tau}|(s_t,a_t)}{\hat{\mu}(s_t,a_t)} + \left(1+\frac{x'(s_t,a_t)}{\hat{\mu}(s_t,a_t)}\right)\left(\|V^t-V^{t-\tau}\|_\infty + \|\lambda^t-\lambda^{t-\tau}\|_1\right). \tag{54}$$

As of $\Gamma_2^t$, by directly applying Proposition H.2 we have $|\Gamma_2^t| \lesssim \frac{\psi}{\varphi(1-\gamma)^2}\mathcal{E}(\tau)$. Thus, to estimate $S_2$, it remains to bound the sum of quantities $\|V^t - V^{t-\tau}\|_\infty$, $\|\lambda^t - \lambda^{t-\tau}\|_1$ and $\frac{x'(s_t,a_t)}{\hat{\mu}(s_t,a_t)}$. For $\|V^t - V^{t-\tau}\|_\infty$ and $\|\lambda^t - \lambda^{t-\tau}\|_1$, as long as $\eta \le \frac{\alpha_\lambda}{2M_\lambda}$, we have

$$\|V^{t+1}-V^t\|_\infty \le \|V^{t+1}-V^t\| \le \frac{\eta}{\alpha_V}\|\widehat{g}_V(Z^t;\zeta_t)\|,$$

$$\|\lambda^{t+1}-\lambda^t\|_1 \le \frac{\eta D_{\lambda,1}}{\alpha_\lambda}\|\widehat{g}_\lambda(Z^t;\zeta_t)\|_\infty,$$

due to Corollary D.5. Therefore, it holds that

$$\frac{1}{T}\sum_{t=\tau+1}^{T}\left(1+\frac{x'(s_t,a_t)}{\hat{\mu}(s_t,a_t)}\right)\left(\|V^t-V^{t-\tau}\|_\infty + \|\lambda^t-\lambda^{t-\tau}\|_1\right)$$

$$\lesssim \frac{\eta}{T}\sum_{t=\tau+1}^{T}\left(1+\frac{x'(s_t,a_t)}{\hat{\mu}(s_t,a_t)}\right)\left(\frac{\|\widehat{g}_V(Z^t;\zeta_t)\|}{\alpha_V} + \frac{D_{\lambda,1}\|\widehat{g}_\lambda(Z^t;\zeta_t)\|_\infty}{\alpha_\lambda}\right)$$

$$\lesssim \frac{\eta}{T}\sum_{t=\tau+1}^{T}\left(\frac{\|\widehat{g}_V(Z^t;\zeta_t)\|^2}{\alpha_V} + \frac{D_{\lambda,1}\|\widehat{g}_\lambda(Z^t;\zeta_t)\|_\infty^2}{\alpha_\lambda}\right) + \frac{\eta}{T}\left(\frac{1}{\alpha_V}+\frac{D_{\lambda,1}}{\alpha_\lambda}\right)\sum_{t=\tau+1}^{T}\left(1+\frac{x'(s_t,a_t)}{\hat{\mu}(s_t,a_t)}\right)^2.$$

Finally, we apply Bernstein's inequality to bound the sequence $\left(\frac{x'(s_t,a_t)^2}{\hat{\mu}(s_t,a_t)^2}\right)_t$ as follows. Due to

$$\frac{x'(s,a)}{\hat{\mu}(s,a)} \leq \frac{\psi}{1-\gamma}, \qquad \mathbb{E}_{s,a\sim\mu}\left[\frac{x'(s,a)^2}{\hat{\mu}(s,a)^2}\right] = \sum_{s,a}\frac{\mu(s,a)}{\hat{\mu}(s,a)}\frac{x'(s,a)}{\hat{\mu}(s,a)}x'(s,a) \leq \frac{8\psi}{(1-\gamma)^2},$$

and Proposition H.1, with probability at least $1 - \delta/10$, it holds that

$$\sum_{t=\tau+1}^{T}\frac{x'(s_t,a_t)^2}{\hat{\mu}(s_t,a_t)^2} \lesssim T\frac{\psi}{(1-\gamma)^2} + t_{\mathrm{mix}}\frac{\psi^2}{(1-\gamma)^2}\log\frac{1}{\delta} \lesssim \frac{T\psi}{(1-\gamma)^2}.$$

Combining all the estimations above completes the proof of (49).

### H.7.1 Derivation of inequality (52)

By definition, it holds that

$$\sum_{t=1}^{\tau}\Gamma^t + \sum_{t=\tau+1}^{T}\Gamma_1^t = \sum_{t=T-\tau+1}^{T}\left\langle\mathcal{G}(Z^t), Z^t - Z'\right\rangle - \sum_{t=1}^{\tau}\left\langle\hat{g}(Z^t;\zeta_t), Z^t - Z'\right\rangle.$$

For a sample $\zeta = (s_0, s, a, s', r, \mathbf{u})$, we denote

$$\widehat{\mathcal{L}}_\zeta(V, \lambda, x) := V(s_0) + \frac{x(s,a)}{\hat{\mu}(s,a)}\left(r - V(s) + \gamma V(s') + \langle\lambda, \mathbf{u}^\kappa\rangle\right).$$

Then, it holds that

$$\langle\hat{g}(Z;\zeta), Z - Z'\rangle = \widehat{\mathcal{L}}_\zeta(V, \lambda, x') - \widehat{\mathcal{L}}_\zeta(V', \lambda', x). \tag{55}$$

Hence we have

$$\left|\langle\hat{g}(Z^t;\zeta_t), Z^t - Z'\rangle\right| \leq \left|\widehat{\mathcal{L}}_{\zeta_t}(V^t, \lambda^t, x')\right| + \left|\widehat{\mathcal{L}}_{\zeta_t}(V', \lambda', x^t)\right| \leq \frac{100\psi}{\varphi(1-\gamma)^2}.$$

Similarly, it holds that

$$\left|\langle\mathcal{G}(Z^t), Z^t - Z'\rangle\right| \leq \left|\mathcal{L}_w(V^t, \lambda^t, x')\right| + \left|\mathcal{L}_w(V', \lambda', x^t)\right| \leq \frac{512}{\varphi(1-\gamma)^2},$$

and we complete the proof by combining the estimations above.

### H.7.2 Derivation of inequality (53)

As in Appendix D.3, we consider the sequences

$$\Delta_V^t := \hat{g}_V(Z^t;\zeta_{t+\tau}) - \mathbb{E}\left[\hat{g}_V(Z^t;\zeta_{t+\tau})\big|\mathcal{F}_t\right],$$
$$\Delta_\lambda^t := \hat{g}_\lambda(Z^t;\zeta_{t+\tau}) - \mathbb{E}\left[\hat{g}_\lambda(Z^t;\zeta_{t+\tau})\big|\mathcal{F}_t\right],$$
$$\Delta_x^t := \hat{g}_x(Z^t;\zeta_{t+\tau}) - \mathbb{E}\left[\hat{g}_x(Z^t;\zeta_{t+\tau})\big|\mathcal{F}_t\right].$$

They are no longer martingale difference sequences, because $\mathbb{E}\left[\Delta^t\big|\mathcal{F}_t\right] = 0$ but $\Delta^t$ is $\mathcal{F}_{t+\tau+1}$ measurable. Therefore, we invoke the following modified version of Bernstein's inequality.

**Lemma H.9** (Modified Bernstein's Inequality). *Assume $\{x_i\}_{i=1}^n$ is a sequence of random vectors in $\mathbb{R}^d$, such that $\mathbb{E}\left[x_t\big|\mathcal{F}_t\right] = 0$ and $x_t$ is $\mathcal{F}_{t+\tau}$ measurable. Assume that $\mathbb{E}\left[\|x_t\|^2\big|\mathcal{F}_t\right] \leq \sigma^2$ and $\|x_t\| \leq M$ a.s., then with probability at least $1 - \delta$,*

$$\left\|\sum_{i=1}^n x^i\right\| \leq 2\sigma\sqrt{n\tau\log\left(\frac{(d+1)\tau}{\delta}\right)} + 2M\tau\log\left(\frac{(d+1)\tau}{\delta}\right).$$

*When the $\ell_2$ norm is replaced by the $\ell_\infty$ norm, i.e., $\{x_i\}_{i=1}^n$ satisfies $\mathbb{E}\left[\|x_t\|_\infty^2\big|\mathcal{F}_t\right] \leq \sigma^2$, we have*

$$\left\|\sum_{i=1}^n x^i\right\|_\infty \leq 2\sigma\sqrt{n\tau\log\left(\frac{2d\tau}{\delta}\right)} + 2M\tau\log\left(\frac{2d\tau}{\delta}\right)$$

*with probability at least $1 - \delta$.*

We still decompose

$$\sum_{t=1}^{T-\tau} \Gamma_3^t = \underbrace{\sum_{t=1}^{T-\tau} \left( \langle \Delta_V^t, V' - V^1 \rangle + \langle \Delta_\lambda^t, \lambda' - \lambda^1 \rangle \right)}_{S_c}$$

$$+ \underbrace{\sum_{t=1}^{T-\tau} \left( \langle \Delta_V^t, V^1 - V^t \rangle + \langle \Delta_\lambda^t, \lambda^1 - \lambda^t \rangle + \langle -\Delta_x^t, x' - x^t \rangle \right)}_{S_m}.$$

Most of the following analysis is similar to the one in Appendix D.3.

**Correlated part** Rewrite

$$S_c = \left\langle \sum_{t=1}^{T-\tau} \Delta_V^t, V' - V^1 \right\rangle + \left\langle \sum_{t=1}^{T-\tau} \Delta_\lambda^t, \lambda' - \lambda^1 \right\rangle$$

$$\leq \|V' - V^1\| \cdot \left\| \sum_{t=1}^{T-\tau} \Delta_V^t \right\| + \|\lambda' - \lambda^1\|_1 \cdot \left\| \sum_{t=1}^{T-\tau} \Delta_\lambda^t \right\|_\infty.$$

For each $t$, by Proposition H.3 (or Proposition H.8), we have

$$\mathbb{E}\left[ \|\Delta_V^t\|^2 \Big| \mathcal{F}_t \right] \leq \mathbb{E}\left[ \|\widehat{g}_V(Z^t; \zeta_{t+\tau})\|^2 \Big| \mathcal{F}_t \right] \lesssim \frac{C(\tau)\psi}{(1-\gamma)^2}, \quad \|\Delta_V^t\| \lesssim \frac{\psi}{1-\gamma},$$

$$\mathbb{E}\left[ \|\Delta_\lambda^t\|_\infty^2 \Big| \mathcal{F}_t \right] \lesssim \mathbb{E}\left[ \|\widehat{g}_\lambda(Z^t; \zeta_{t+\tau})\|_\infty^2 \Big| \mathcal{F}_t \right] \lesssim \frac{C(\tau)\psi}{(1-\gamma)^2}, \quad \|\Delta_\lambda^t\|_\infty \lesssim \frac{\psi}{1-\gamma}.$$

Thus, we can apply Lemma H.9 to derive that, with probability at least $1 - \delta/20$,

$$\left\| \sum_{t=1}^{T-\tau} \Delta_V^t \right\| \lesssim \frac{1}{1-\gamma} \sqrt{T\tau C(\tau)\psi \log(1/\delta)} + \frac{\psi}{1-\gamma} \cdot \tau \log(1/\delta),$$

$$\left\| \sum_{t=1}^{T-\tau} \Delta_\lambda^t \right\|_\infty \lesssim \frac{1}{1-\gamma} \sqrt{T\tau C(\tau)\psi \log(I/\delta)} + \frac{\psi}{1-\gamma} \cdot \tau \log(I/\delta).$$

Therefore, it holds that with probability at least $1 - \delta/20$,

$$S_c \lesssim \frac{1}{\varphi(1-\gamma)^2} \sqrt{T\tau C(\tau)|\mathcal{S}|\psi\iota} + \frac{\tau\psi\iota}{1-\gamma} \lesssim \frac{1}{\varphi(1-\gamma)^2} \sqrt{T\tau C(\tau)|\mathcal{S}|\psi\iota}. \qquad (56)$$

**Martingale part** In order to bound $S_m$, we have to consider $\overline{\Delta}_V^t := \langle \Delta_V^t, V^1 - V^t \rangle, \overline{\Delta}_\lambda^t := \langle \Delta_\lambda^t, \lambda^1 - \lambda^t \rangle, \overline{\Delta}_x^t := \langle \Delta_x^t, x^t - x' \rangle$. By Proposition H.3, it holds that

$$\left| \overline{\Delta}_V^t \right| \lesssim \frac{\psi}{\varphi(1-\gamma)^2}, \quad \mathbb{E}\left[ \left(\overline{\Delta}_V^t\right)^2 \Big| \mathcal{F}_t \right] \leq D_V^2 \mathbb{E}\left[ \|\Delta_V^t\|^2 \Big| \mathcal{F}_t \right] \lesssim \frac{C(\tau)\psi}{\varphi^2(1-\gamma)^4},$$

$$\left| \overline{\Delta}_\lambda^t \right| \leq \frac{\psi}{\varphi(1-\gamma)}, \quad \mathbb{E}\left[ \left(\overline{\Delta}_\lambda^t\right)^2 \Big| \mathcal{F}_t \right] \leq D_{\lambda,1}^2 \mathbb{E}\left[ \|\Delta_\lambda^t\|_\infty^2 \Big| \mathcal{F}_t \right] \lesssim \frac{C(\tau)\psi}{\varphi^2(1-\gamma)^2},$$

$$\left| \overline{\Delta}_x^t \right| \leq \frac{\psi}{\varphi(1-\gamma)^2}, \quad \mathbb{E}\left[ \left(\overline{\Delta}_x^t\right)^2 \Big| \mathcal{F}_t \right] \leq \mathbb{E}\left[ \left\| \frac{x' - x^t}{\sqrt{x' + x^t}} \right\|^2 \|\Delta_x^t\|_{x'+x^t}^2 \Big| \mathcal{F}_t \right] \lesssim \frac{C(\tau)\mathcal{N}\psi}{\varphi^2(1-\gamma)^4}.$$

Thus, by applying Lemma H.9, the following three estimations hold with probability at least $1 - \delta/20$

$$\sum_{t=1}^{T-\tau} \overline{\Delta}_V^t \lesssim \frac{1}{\varphi(1-\gamma)^2} \sqrt{T\tau C(\tau)\psi \log(1/\delta)} + \frac{\psi}{\varphi(1-\gamma)^2} \cdot \tau \log(1/\delta),$$

$$\sum_{t=1}^{T-\tau} \overline{\Delta}_\lambda^t \lesssim \frac{1}{\varphi(1-\gamma)} \sqrt{T\tau C(\tau)\psi \log(1/\delta)} + \frac{\psi}{\varphi(1-\gamma)} \cdot \tau \log(1/\delta),$$

$$\sum_{t=1}^{T-\tau} \overline{\Delta}_x^t \lesssim \frac{1}{\varphi(1-\gamma)^2}\sqrt{T\tau C(\tau)\mathcal{N}\psi \log(1/\delta)} + \frac{\psi}{\varphi(1-\gamma)^2}\cdot\tau\log(1/\delta).$$

Therefore,

$$S_m \lesssim \frac{1}{\varphi(1-\gamma)^2}\sqrt{T\tau C(\tau)\mathcal{N}\psi\iota} + \frac{\tau\psi\iota}{\varphi(1-\gamma)^2} \lesssim \frac{1}{\varphi(1-\gamma)^2}\sqrt{T\tau C(\tau)\mathcal{N}\psi\iota}. \qquad (57)$$

Combining (57) with (56) completes the proof.

*Proof of Lemma H.9.* We reduce Lemma H.9 to the standard martingale Bernstein's inequality (Lemma B.1). The set $[n]$ can be decomposed into

$$[n] = \bigsqcup_{k=1}^{\tau} \mathcal{I}_k, \qquad \mathcal{I}_k := \{j \in [n] : j \equiv k \mod \tau\}.$$

For each $k$, the sequence $(X_j)_{j\in\mathcal{I}_k}$ is a martingale difference sequence w.r.t. the filtration $(\mathcal{F}_j)_{j\in\mathcal{I}_k}$. Hence by Lemma B.1, with probability at least $1 - \delta/\tau$, we have

$$\left\|\sum_{j\in\mathcal{I}_k} x^j\right\| \le 2\sigma\sqrt{|\mathcal{I}_k|\log\left(\frac{(d+1)\tau}{\delta}\right)} + 2M\log\left(\frac{(d+1)\tau}{\delta}\right).$$

Summing over $k = 1, \cdots, \tau$ yields that with probability at least $1 - \delta$

$$\left\|\sum_{j=1}^{n} x^j\right\| \le 2\sigma\sqrt{\log\left(\frac{(d+1)\tau}{\delta}\right)}\sum_{k=1}^{\tau}\sqrt{|\mathcal{I}_k|} + 2M\tau\log\left(\frac{(d+1)\tau}{\delta}\right)$$

$$\le 2\sigma\sqrt{n\tau\log\left(\frac{(d+1)\tau}{\delta}\right)} + 2M\tau\log\left(\frac{(d+1)\tau}{\delta}\right),$$

where the last inequality is due to the Cauchy inequality.

The analogous $\ell_\infty$ case can be done similarly. $\qquad\square$

### H.7.3 Derivation of inequality (54)

By (55), it holds that
$$\begin{aligned}
\Gamma_4^t &= \left\langle \widehat{g}(Z^{t-\tau};\zeta_t), Z^{t-\tau} - Z'\right\rangle - \left\langle\widehat{g}(Z^t;\zeta_t), Z^t - Z'\right\rangle \\
&= \widehat{\mathcal{L}}_{\zeta_t}(V^{t-\tau}, \lambda^{t-\tau}, x') - \widehat{\mathcal{L}}_{\zeta_t}(V', \lambda', x^{t-\tau}) + \widehat{\mathcal{L}}_{\zeta_t}(V', \lambda', x^t) - \widehat{\mathcal{L}}_{\zeta_t}(V^t, \lambda^t, x').
\end{aligned} \qquad (58)$$

Then we have

$$\begin{aligned}
&\left|\widehat{\mathcal{L}}_{\zeta_t}(V', \lambda', x^t) - \widehat{\mathcal{L}}_{\zeta_t}(V', \lambda', x^{t-\tau})\right| \\
&= \frac{|x^t - x^{t-\tau}|(s_t, a_t)}{\hat{\mu}(s_t, a_t)}\left|r_t - V'(s_t) + \gamma V'(s_{t+1}) + \langle\lambda', \mathbf{u}_t^\kappa\rangle\right| \\
&\le \frac{|x^t - x^{t-\tau}|(s_t, a_t)}{\hat{\mu}(s_t, a_t)}\left(1 + \frac{16}{1-\gamma}\left(1 + \frac{2}{\varphi}\right) + \frac{8(1+\kappa)}{\varphi}\right) \\
&\le \frac{64}{\varphi(1-\gamma)}\frac{|x^t - x^{t-\tau}|(s_t, a_t)}{\hat{\mu}(s_t, a_t)}.
\end{aligned}$$

Similarly,

$$\begin{aligned}
&\left|\widehat{\mathcal{L}}_{\zeta_t}(V^t, \lambda^t, x') - \widehat{\mathcal{L}}_{\zeta_t}(V^{t-\tau}, \lambda^{t-\tau}, x')\right| \\
&\le \left|V^t(s_{0,t}) - V^{t-\tau}(s_{0,t})\right| \\
&\quad + \frac{x'(s_t, a_t)}{\hat{\mu}(s_t, a_t)}\left(\left|V^t(s_t) - V^{t-\tau}(s_t)\right| + \gamma\left|V^t(s_{t+1}) - V^{t-\tau}(s_{t+1})\right| + \left|\langle\lambda^t - \lambda^{t-\tau}, \mathbf{u}_t^\kappa\rangle\right|\right) \\
&\le \left\|V^t - V^{t-\tau}\right\|_\infty\left(1 + 2\frac{x'(s_t, a_t)}{\hat{\mu}(s_t, a_t)}\right) + \left\|\lambda^t - \lambda^{t-\tau}\right\|_1 \cdot 128\frac{x'(s_t, a_t)}{\hat{\mu}(s_t, a_t)}.
\end{aligned}$$

The proof is completed by combining (58) with the estimations above.

## H.8 Proof of Proposition H.6

The proof of Proposition H.6 is separated into two steps.

**Step 1.** We derive bounds on $\sum p(x^t; \zeta_{t+\tau})^2$ and $\sum q(x^t; \zeta_{t+\tau})$ by directly applying Bernstein's inequality.

**Step 2.** We leverage the idea demonstrate in (13) again to bound $\sum p(x^t; \zeta_t)^2$ and $\sum q(x^t; \zeta_t)$, by bounding their difference with $\sum p(x^t; \zeta_{t+\tau})^2$ and $\sum q(x^t; \zeta_{t+\tau})$ respectively.

Then we finalize the proof by combining the results of Step 1 and Step 2.

### H.8.1 Step 1. Bounding the asynchronous sums

First, let us present the following result for the ease of discussion.
**Corollary.** *Assume $\{x_i\}_{i=1}^n$ is a sequence of random variables, such that $x_t$ is $\mathcal{F}_{t+\tau}$ measurable, and $\mathbb{E}\left[|x_t| \big| \mathcal{F}_t\right] \le c$, $|x_t| \le M$ a.s. Then with probability at least $1 - \delta$,*

$$\left| \frac{1}{n} \sum_{i=1}^n x^i \right| \le 2c\tau + 3M\tau \frac{\log(2\tau/\delta)}{n}.$$

By Proposition H.8, we have

$$q(x^t; \zeta_{t+\tau}) \le \frac{\psi}{(1-\gamma)\varsigma}, \quad \mathbb{E}\left[q(x^t; \zeta_{t+\tau}) \big| \mathcal{F}_t\right] \le C(\tau)\frac{\mathcal{N}\psi}{1-\gamma}.$$

Applying the above corollary yields that with probability at least $1 - \delta/20\tau_0$,

$$\sum_{t=1}^{T-\tau} q(x^t; \zeta_{t+\tau}) \lesssim TC(\tau)\frac{\mathcal{N}\psi}{1-\gamma} + \frac{\tau\psi}{(1-\gamma)\varsigma} \log\left(\frac{\tau_0}{\delta}\right)$$

$$\lesssim TC(\tau)\frac{\mathcal{N}\psi}{1-\gamma} + \frac{\tau_0 \mathcal{N}\psi^2 \iota}{\varphi(1-\gamma)^3 \epsilon_e}$$

$$\lesssim TC(\tau)\frac{\mathcal{N}\psi}{1-\gamma},$$

where the last inequality is due to $T \gtrsim \frac{\tau_0^2 \mathcal{N}\psi\iota^3}{\varphi^2(1-\gamma)^4 \epsilon_e^2} \ge \frac{\tau_0 \psi\iota}{\varphi(1-\gamma)^2 \epsilon_e}$.

Similarly, we have

$$p(x^t; \zeta_{t+\tau}) \le \frac{\psi}{1-\gamma}, \quad \mathbb{E}\left[p(x^t; \zeta_{t+\tau})^2 \big| \mathcal{F}_t\right] \le C(\tau)\frac{4\psi}{(1-\gamma)^2}.$$

Therefore, for each $1 \le \tau \le \tau_0$, it holds with probability at least $1 - \delta/20\tau_0$

$$\sum_{t=1}^{T-\tau} p(x^t; \zeta_{t+\tau})^2 \lesssim \frac{TC(\tau)\psi}{(1-\gamma)^2} + \frac{\tau\psi^2}{(1-\gamma)^2} \log\left(\frac{\tau_0}{\delta}\right) \lesssim \frac{TC(\tau)\psi}{(1-\gamma)^2}.$$

By taking the union bound for $1 \le \tau \le \tau_0$, we conclude that with probability at least $1 - \delta/10$,

$$\sum_{t=1}^{T-\tau} p(x^t; \zeta_{t+\tau})^2 \lesssim \frac{TC(\tau)\psi}{(1-\gamma)^2}, \qquad \sum_{t=1}^{T-\tau} q(x^t; \zeta_{t+\tau}) \lesssim \frac{TC(\tau)\mathcal{N}\psi}{1-\gamma}, \tag{59}$$

hold simultaneously and uniformly for $1 \le \tau \le \tau_0$.

### H.8.2 Step 2. Bounding the difference

Utilizing the closeness between $Z^t$ and $Z^{t+\tau}$, we bound the difference $q(x^t; \zeta_t) - q(x^t; \zeta_{t+\tau})$ as

$$\sum_{t=1}^{T} q(x^t; \zeta_t) - \sum_{t=1}^{T-\tau} q(x^t; \zeta_{t+\tau}) \le \sum_{t=1}^{\tau} q(x^t; \zeta_t) + \sum_{t=1}^{T-\tau} q(|x^t - x^{t+\tau}|; \zeta_{t+\tau})$$

$$\le \frac{\tau\psi}{(1-\gamma)\varsigma} + \frac{1}{\varsigma} \sum_{t=1}^{T-\tau} p(|x^t - x^{t+\tau}|; \zeta_{t+\tau}). \tag{60}$$

We next deal with the quantity $p(|x^t - x^{t+\tau}|; \zeta_{t+\tau})$ carefully. For any $(s,a) \in \mathcal{S} \times \mathcal{A}$, it holds that

$$p(|x^t - x^{t+\tau}|; s, a) = \frac{|x^t(s,a) - x^{t+\tau}(s,a)|}{\hat{\mu}(s,a)}$$

$$\leq \frac{1}{\hat{\mu}(s,a)} \sum_{t'=t}^{t+\tau-1} \left| x^{t'}(s,a) - x^{t'+1}(s,a) \right|$$

$$\leq \frac{1}{\hat{\mu}(s,a)} \sum_{t'=t}^{t+\tau-1} \sqrt{x^{t'}(s,a) + x^{t'+1}(s,a)} \left\| \frac{x^{t'} - x^{t'+1}}{\sqrt{x^{t'} + x^{t'+1}}} \right\|$$

$$\overset{(a)}{\lesssim} \frac{\eta}{\alpha_x} \frac{1}{\hat{\mu}(s,a)} \sum_{t'=t}^{t+\tau-1} \sqrt{x^{t'}(s,a) + x^{t'+1}(s,a)} \left\| \widehat{g}_x(Z^{t'}; \zeta_{t'}) \right\|_{x^{t'}}$$

$$\overset{(b)}{\lesssim} \frac{\eta}{\alpha_x} \frac{1}{\hat{\mu}(s,a)} \sum_{t'=t}^{t+\tau-1} \sqrt{x^{t'}(s,a) + x^{t'+1}(s,a)} \cdot \frac{1}{\varphi(1-\gamma)} \sqrt{q(x^t; \zeta_t)}$$

$$= \frac{\eta}{\alpha_x} \cdot \frac{1}{\varphi(1-\gamma)} \sum_{t'=t}^{t+\tau-1} \sqrt{q(x^{t'}; s, a) + q(x^{t'+1}; s, a)} \sqrt{q(x^t; \zeta_t)}$$

$$\overset{(c)}{\leq} \frac{\eta}{\alpha_x} \cdot \frac{1}{\varphi(1-\gamma)} \sqrt{\sum_{t'=t}^{t+\tau-1} q(x^t; \zeta_t)} \sqrt{\sum_{t'=t}^{t+\tau} q(x^{t'}; s, a)}.$$

Here the inequality (a) is due to Corollary D.5, the inequality (b) is due to Proposition H.8, and the inequality (c) comes from Cauchy inequality. Hence, we have

$$\sum_{t=1}^{T-\tau} p(|x^t - x^{t+\tau}|; \zeta_{t+\tau}) \lesssim \frac{\eta}{\alpha_x} \cdot \frac{1}{\varphi(1-\gamma)} \sum_{t=1}^{T-\tau} \sqrt{\sum_{t'=t}^{t+\tau-1} q(x^t; \zeta_t)} \sqrt{\sum_{t'=t}^{t+\tau} q(x^{t'}; \zeta_{t+\tau})}$$

$$\leq \frac{\eta}{\alpha_x} \cdot \frac{1}{\varphi(1-\gamma)} \sqrt{\sum_{t=1}^{T-\tau} \sum_{t'=t}^{t+\tau-1} q(x^t; \zeta_t)} \sqrt{\sum_{t=1}^{T-\tau} \sum_{t'=t}^{t+\tau} q(x^{t'}; \zeta_{t+\tau})}$$

$$\leq \frac{\eta}{\alpha_x} \cdot \frac{1}{\varphi(1-\gamma)} \sqrt{\tau \sum_{t=1}^{T} q(x^t; \zeta_t)} \sqrt{\sum_{j=1}^{\tau} \sum_{t=1}^{T-j} q(x^t; \zeta_{t+j})}. \qquad (61)$$

Combining (61) with (60) yields

$$\sum_{t=1}^{T-\tau} q(x^t; \zeta_t) - \sum_{t=1}^{T-\tau} q(x^t; \zeta_{t+\tau})$$

$$\lesssim \frac{\tau\psi}{(1-\gamma)\varsigma} + \frac{\eta}{\alpha_x} \frac{1}{\varphi(1-\gamma)\varsigma} \sqrt{\tau \sum_{t=1}^{T} q(x^t; \zeta_t)} \sqrt{\sum_{t=1}^{T-\tau} \sum_{t'=t}^{t+\tau} q(x^{t'}; \zeta_{t+\tau})} \qquad (62)$$

Similarly, it holds that for $0 \leq j \leq \tau$,

$$\sum_{t=1}^{T-j} q(x^t; \zeta_{t+j}) - \sum_{t=1}^{T-\tau} q(x^t; \zeta_{t+\tau})$$

$$\lesssim \frac{\tau\psi}{(1-\gamma)\varsigma} + \frac{\eta}{\alpha_x} \frac{1}{\varphi(1-\gamma)\varsigma} \sqrt{\tau \sum_{t=1}^{T} q(x^t; \zeta_t)} \sqrt{\sum_{t=1}^{T-\tau} \sum_{t'=t}^{t+\tau} q(x^{t'}; \zeta_{t+\tau})}. \qquad (63)$$

### H.8.3 Combining Step 1 and Step 2

Actually, (63) is already enough to bound $\sum_{t=1}^{T} q(x^t; \zeta_t)$. For simplicity, we denote

$$Q_1 := \frac{c_0 \tau \psi}{(1-\gamma)\varsigma} + \sum_{t=1}^{T-\tau} q(x^t; \zeta_{t+\tau}), \qquad Q_2 := \sum_{t=1}^{T} q(x^t; \zeta_t), \qquad c := c_0 \frac{\eta}{\alpha_x} \frac{1}{\varphi(1-\gamma)\varsigma},$$

$$Q_3 := \frac{1}{\tau}\sum_{t=1}^{T-\tau}\sum_{t'=t}^{t+\tau} q(x^{t'};\zeta_{t+\tau}) = \frac{1}{\tau}\sum_{j=1}^{\tau}\sum_{t=1}^{T-\tau+j} q(x^t;\zeta_{t+\tau-j}),$$

where $c_0$ is a universal constant hidden by the $\lesssim$ in (63). Now, (63) implies

$$Q_2 \le Q_1 + c\tau\sqrt{Q_2 Q_3}, \quad Q_3 \le Q_1 + c\tau\sqrt{Q_2 Q_3}, \tag{64}$$
$$\Rightarrow Q_2 + Q_3 \le Q_1 + c\tau(Q_2 + Q_3).$$

Thus, as long as $c\tau_0 \le \frac{1}{2}$, we have $Q_2 + Q_3 \le 2Q_1$. The condition $c\tau_0 \le \frac{1}{2}$ is equivalent to

$$\frac{1}{2c_0} \ge \tau_0 \frac{\eta}{\alpha_x}\frac{1}{\varphi(1-\gamma)\varsigma} = \tau_0 \cdot \sqrt{\frac{1}{T}} \cdot \left(\frac{1}{\varphi(1-\gamma)}\sqrt{\frac{\mathcal{N}\psi}{\log\psi}}\right)^{-1}\frac{1}{\varphi(1-\gamma)\varsigma} = \sqrt{\frac{4\tau_0^2\mathcal{N}\psi\log\psi}{\varphi^2(1-\gamma)^4\epsilon_e^2}}\cdot\frac{1}{T}.$$

Thus, $T \ge 16c_0^2 \frac{\tau_0^2\mathcal{N}\psi\log\psi}{\varphi^2(1-\gamma)^4\epsilon_e^2}$ is enough to ensure $Q_2 \le 2Q_1, Q_3 \le 2Q_1$ for any $\tau \le \tau_0$. Here, according to (59) we have

$$Q_1 = \frac{c_0\tau\psi}{(1-\gamma)\varsigma} + \sum_{t=1}^{T-\tau} q(x^t;\zeta_{t+\tau}) \lesssim \frac{c_0\tau_0\psi}{(1-\gamma)\varsigma} + \frac{TC(\tau)\mathcal{N}\psi}{1-\gamma} \lesssim \frac{TC(\tau)\mathcal{N}\psi}{1-\gamma}.$$

Consequently, we obtain

$$\sum_{t=1}^{T} q(x^t;\zeta_t) \lesssim \frac{TC(\tau)\mathcal{N}\psi}{1-\gamma}, \tag{65}$$
$$\sum_{t=1}^{T-\tau}\sum_{t'=t}^{t+\tau} q(x^{t'};\zeta_{t+\tau}) \lesssim \tau\frac{TC(\tau)\mathcal{N}\psi}{1-\gamma}.$$

Hence, by (61),

$$\sum_{t=\tau+1}^{T} p(|x^t - x^{t-\tau}|;\zeta_t) \lesssim \frac{\tau C(\tau)}{1-\gamma}\sqrt{T\mathcal{N}\psi\log\psi}.$$

We can further establish the bound for $\sum_{t=1}^{T} p(x^t;\zeta_t)^2$ as

$$\sum_{t=1}^{T} p(x^t;\zeta_t)^2 \lesssim \sum_{t=1}^{\tau} p(x^t;\zeta_t)^2 + \sum_{t=1}^{T-\tau}\left[p(x^t;\zeta_{t+\tau})^2 + p(|x^t - x^{t+\tau}|;\zeta_{t+\tau})^2\right]$$
$$\overset{(a)}{\lesssim} \tau\frac{\psi^2}{(1-\gamma)^2} + \sum_{t=1}^{T-\tau} p(x^t;\zeta_{t+\tau})^2 + \frac{\psi}{1-\gamma}\sum_{t=1}^{T-\tau} p(|x^t - x^{t+\tau}|;\zeta_{t+\tau}) \tag{66}$$
$$\lesssim \frac{\tau\psi^2}{(1-\gamma)^2} + \frac{TC(\tau)\psi}{(1-\gamma)^2} + \frac{\tau C(\tau)}{1-\gamma}\sqrt{T\mathcal{N}\psi\log\psi}$$
$$\overset{(b)}{\lesssim} \frac{TC(\tau)\psi}{(1-\gamma)^2},$$

where the inequality (a) is due to $p(x^t;\zeta_t) \le \frac{\psi}{1-\gamma}$, $p(|x^t - x^{t+\tau}|;\zeta_{t+\tau}) \le \frac{2\psi}{1-\gamma}$, and the inequality (b) is due to our requirement $T \gtrsim \frac{\tau_0^2\mathcal{N}\psi\iota}{\varphi^2(1-\gamma)^4\epsilon_e^2}$.

The proof is completed by taking $\tau = \tau_0$ in (66) and (65).