# OpenReview forum: "A Near-Optimal Primal-Dual Method for Off-Policy Learning in CMDP"
_NeurIPS.cc/2022/Conference — NeurIPS 2022 Accept_

### Official Review · Reviewer_73m4 · 2022-07-12

**Rating:** 6
**Confidence:** 3
**Soundness:** 3 good
**Presentation:** 4 excellent
**Contribution:** 3 good

**Summary:**

This paper studies the offline CMDP through the linear programming (LP) formulation and the primal-dual based method. An upper bound is proved under the notion of the concentrability coefficient in offline RL literature. A lower bound is established to demonstrate the near-optimality of the proposed upper bound. Finally, an adaptive algorithm is proposed to address the situation when the optimal concentrability coefficient is unknown.

**Questions:**

(1) Could the authors discuss the potential of relaxing the assumption of a finite concentrability coefficient?

(2) Could authors discuss the technical challenges and contributions of this paper compared with some key references like [14, 30] and [23]?


**Strengths And Weaknesses:**

Strengths:

(1) The paper is well-written and the presentation is very clear.

(2) Offline/off-policy CMDP is an important problem and there are not as many offline/off-policy CMDP in the literature as for online CMDP.

(3) The theoretical results are solid. Both the upper bound and lower bound are provided, and the dependencies on the key parameters are very clear. In addition, an adaptive algorithm is proposed to address the situation when the optimal concentrability coefficient is unknown, without the optimality sacrifice.

Weaknesses:

(1) The assumption of having an offline dataset with a finite concentrability coefficient seems to be too strong in practice. Most of time, we would only have an offline dataset that does not well covers all state-action pairs.

(2) The LP-based approach is very restrictive and not scalable. This further restricts the application of this work to the practical implementations.

(3) There are some other published works in offline CMDP that the author may need to compare. For example:

Wu, Runzhe, et al. "Offline Constrained Multi-Objective Reinforcement Learning via Pessimistic Dual Value Iteration." Advances in Neural Information Processing Systems 34 (2021): 25439-25451.

Xu, Haoran, Xianyuan Zhan, and Xiangyu Zhu. "Constraints penalized q-learning for safe offline reinforcement learning." Proceedings of the AAAI Conference on Artificial Intelligence. Vol. 36. No. 8. 2022.

---

> ### Author Response · Authors · 2022-07-30
> **Response to reviewer 73m4**
>
> We'd like to thank the reviewer for the insightful comments. The reviewer's concerns are addressed as follows.
>
> $\mathbf{Weakness 1}$ & $\mathbf{Q1.}$ The assumption of an offline dataset with  finite concentrability coefficient seems too strong... Could the authors discuss the potential of relaxing the assumption of a finite concentrability coefficient?
>
> Answer: There are several aspects of the assumption of a finite concentrability coefficient in our paper.
>
> (1) A finite single-policy concentrability coefficient does not require the well covering all state-action pairs; instead, it only assumes that the trajectory of optimal policy to be covered. Therefore, our assumption is much weaker compared to the uniform coverage [35, 36] or the uniform concentrability [14, 20].  For example, if the dataset is generated by a near optimal policy (imitation learning with expert data), then it is natural to expect $C^*=\mathcal{O}(1)$ even though a large fraction of state-action pairs are unvisited.
>
> (2) The guarantee of DPDL can be generalized to the case $\psi<C^*$ (which covers the possibility that $C^*=\infty$). In this case, DPDL provably outputs a policy that is comparable to the best policy in $\Pi(\psi)$, the class of policy whose deviation is controlled by $\psi$.  We also propose an adaptive deviation control framework that is adapted to unknown $C^*$.
>
> (3) Further relaxation of the finite concentrability is hard in general. Imagine an offline multi-arm bandit problem where one only has access to offline dataset of the arms. In this case, an infinite single-policy concentrability corresponds to the case where the offline dataset never touches the optimal arm. Therefore, no optimality can be guaranteed if the the finite concentrability assumption is relaxed.
>
>
> $\mathbf{Weakness2.}$ The LP-based approach is very restrictive and not scalable. This further restricts the application of this work to the practical implementations.
>
> Answer: We have to admit that the scalability of the LP-based approach is indeed a problem. Recently, there are several works on MDP that reveal the LP approach is potentially scalable: [*] considers the LP-based approach with linear features; [18] parameterizes the primal and dual variables by neural networks, and achieves performance comparable to or better than TRPO/PPO/DQN in several testing environments.
>
> On the other hand, how to make LP-based approach scalable or how to replace it is itself an important problem. This is because, currently, many algorithms for online learning CMDP that have theoretical guarantee directly use LP in the planning step.
>
> [*] Chen, Y., Li, L., and Wang, M. Scalable bilinear $\pi$ learning using state and action features.
>
> $\mathbf{Weakness 3}$ & $\mathbf{Q2.}$ There are some other works in offline CMDP that the authors may need to compare.  Could authors discuss the technical challenges and contributions of this paper compared with some key references like [14, 30] and [32]?
>
> Answer: We thank the reviewer for pointing this out, we summarize the technical challenges and contributions of this paper compared to previous works as follows:
>
> (1) The "single-policy concentrability'' condition in (tabular) MDP is first formulated in [23]; however, it is previously unclear whether this notion can be further extended. In our work, we identify this concept for CMDP, and we discover an essential difference between offline MDP and CMDP: offline MDP can be solved using $\mathcal{O}(|\mathcal{S}| C^* \epsilon^{-2})$ samples, but at least $\Omega(min (|\mathcal{S}||\mathcal{A}|,|\mathcal{S}|+I ) C^* \epsilon^{-2})$ samples are need for offline CMDP, because we have to fulfill $I$ constraints simultaneously.
>
> (2) In terms of the information theoretic lower bound for offline CMDP, compared to the hard instance of offline MDP, our construction is more intricate because we need to correlate different actions via properly designed constraints. We also establish the necessity of Slater's condition, and hence justify this commonly used assumption.
>
>
> (3) Prior to our work, the only provably efficient model-free algorithm under single-policy concentrability are variants of Q-learning [24, 33] (which do not work for CMDP), while [17, 23, 30, 31, 34] all take the model-based approach. The deviation control mechanism we develop makes the primal-dual approach work efficiently under single-policy concentrability, and it is naturally model-free.
>
> (4) We consider the asynchronous setting where the dataset is a single trajectory generated by a behavior policy. In this setting, our analysis handles the correlated gradient estimators with large variance. We believe our techniques can also benefit the analysis of other algorithms in the asynchronous setting.

---

### Official Review · Reviewer_EXgi · 2022-07-12

**Rating:** 6
**Confidence:** 4
**Soundness:** 2 fair
**Presentation:** 3 good
**Contribution:** 2 fair

**Summary:**

This paper proposes a model-free off-policy reinforcement learning algorithm DPDL that aims to solve the constrained MDP (CMDP) problems given an offline static dataset, and also provides an information theoretic lower bound on the sample complexity in the offline CMDP setting. DPDL adopts the primal-dual approach and addresses the distribution shift challenge in the offline setting via an adaptive deviation control mechanism.


**Questions:**

A large part of the analysis in this current version replies on an assumption that we know a reference distribution, while we actually don't have an access to such a distribution. Although the paper proposed to use an estimated distribution instead, this current does not provide a principled way to construct such a distribution estimation.



**Limitations:**

As mentioned in the strengths and weaknesses section, some of the concerns include a lack of experiments that can demonstrate the practice effectiveness of the proposed DPDL algorithm as well as principled ways to derive a practical algorithms based on the theoretical analysis.

**Strengths And Weaknesses:**

An innovative aspect of this work is that it considers a combination of the offline reinforcement learning (RL) and safe RL formulated as a constrained MDP. Currently, there are limited existing work on this specific combination, while recent exploration on offline RL and safe RL separately are extensive. The paper could be further improved if the authors would elaborate on the factors that motivate the authors to combine the two settings from both theoretic and practical perspectives.

In addition, while this paper analyzed the algorithm theoretically, it remains unclear whether it would be effective in solving real-world problems due to a number of issues. Towards this end, experiment results should be conducted to address this issue.

---

> ### Author Response · Authors · 2022-07-30
> **Response to reviewer EXgi**
>
> We thank the reviewer for her/his time and thoughtful feedback. We address the comments in detail as follows.
>
> $\mathbf{Q1.}$ A large part of the analysis in this current version relies on an assumption that we know a reference distribution, while we actually don't have an access to such a distribution.  Although the paper proposed to use an estimated distribution instead, this current does not provide a principled way to construct such a distribution estimation.
>
> Answer: First, all of our analysis are done with regard to the estimated distribution $\hat\mu$, we do not assume the knowledge of the reference distribution $\mu$. Second, we do provide a way to construct the $\hat\mu$, please see Eq (10) of Algorithm 1. By properly setting the parameters $N_e$ and $\varsigma$ in (10), we establish the important properties of the estimator $\hat\mu$ in Proposition 4.3, which is all we need for $\hat\mu$ in the convergence analysis.
>
> $\mathbf{Q2.}$ About the lack of numerical experiments to demonstrate the practice effectiveness of the proposed DPDL algorithm.
>
> Answer: The primary goal of our paper is to provide theoretical foundations and insights for offline learning CMDP, by studying the information theoretic lower bounds under single-policy concentrability and the algorithms to achieve almost tight sample complexity upper bound and zero constraint violation. First, through Theorem 4.1 and 5.1, we confirm the minimax optimal dependence on $|\mathcal{S}|,|\mathcal{A}|,C^*,\epsilon$ and $I$ for CMDP problem, and the dependence on $1-\gamma$ is also almost optimal. Second, with Theorem 5.2, we confirm that the Slater's condition is the necessary condition for achieving zero constraint violation.
>
> We do agree that numerical experiments are important for demonstrating the practice effectiveness. However, this is not main focus of our paper and we are not able to complete the experiments during the rebuttal period.

---

### Official Review · Reviewer_Gvdm · 2022-07-12

**Rating:** 6
**Confidence:** 3
**Soundness:** 4 excellent
**Presentation:** 2 fair
**Contribution:** 2 fair

**Summary:**

This manuscript investigates offline Constrained Markov Decision Process (CMDP), where on top of optimizing the cumulative reward r one must maintain (in a hard way) a cumulative constraint u to be non-negative. The offline setting uses only the offline data without a generative model. This manuscript proposes Deviation-controlled Primal-Dual Learning (DPDL), an algorithm that uses the saddle point formulation of MDP and a mirror descent-like update. When the concentrability coefficient of a CMDP is finite, assuming the Slater's condition, the algorithm guarantees the convergence and a proved sample complexity. An information-theoretic lower bound validates the optimality of this sample complexity up to a 1/(1-gamma) factor.

**Questions:**

N/A

**Strengths And Weaknesses:**

This paper improves previous works on CMDP by removing the generative model. Without a generative model, it is necessary to have the Slater's condition to satisfy the constraint. With the Slater's condition, the authors show a close-to-matching pair of upper and lower bounds of sample complexity (need concentrability assumption). The argument seems natural and the improvement is clear.

I would question the relevance of the manuscript as the assumptions needed to conclude the sample complexity are heavy. Despite one of them is necessary (Slater), it is not clear if this particular setting is worth investigating.

I also suggest the authors to highlight the technical contributions, if any.

---

> ### Author Response · Authors · 2022-07-31
> **Response to reviewer Gvdm**
>
> We thank the reviewer for her/his time and thoughtful feedback. We address the comments in detail as follows.
>
> $\mathbf{Weakness1.}$ I would question the relevance of the manuscript as the assumptions needed to conclude the sample complexity are heavy. Despite one of them is necessary (Slater), it is not clear if this particular setting is worth investigating.
>
> Answer: There are only 2 central assumptions in our work, we discuss the consequence of relaxing them as follows.
>
> (1) Slater's condition: We have established the necessity of Slater's condition for ensuring 0 constraint violation. When the Slater's condition is absent, our analysis of DPDL implies that it will output a policy with $\mathcal{O}(\epsilon)$ reward suboptimality gap and constraint violation.
>
> (2) Finite single-policy concentrability: On the one hand, we agree that this assumption can be a little bit strong in practice since the reference distribution may not fully cover the support of the optimal occupancy measure. However, we should notice that when this coefficient is infinity, there will definitely be no guarantee for obtaining the optimal policy. Therefore, this is an unavoidable assumption in the offline RL.  If we relax this assumption, then our DPDL will output a policy that is near optimal among the policies in $\Pi(\psi)$ for any $\psi>0$ selected by DPDL. (The optimal policy lies in $\Pi(+\infty)$.)
>
> On the other hand, among all the assumptions that can provide optimality guarantee for offline RL, our assumption is indeed very weak. This is because a finite single-policy concentrability coefficient does not require the well covering all state-action pairs; instead, it only assumes that the trajectory of optimal policy to be covered. This is much weaker compared to the uniform coverage [35, 36] or the uniform concentrability assumption [14, 20].
>
>
> $\mathbf{Weakness2.}$ I also suggest the authors to highlight the technical contributions, if any.
>
> Answer: We thank the reviewer for this suggestion. The following comments will be added under our main contribution (Line 51 - 63):
>
> "Besides the above main contributions, our construction of the worst-case instance in the lower bound derivation can be of independent interest. In order to characterize the influence of the constraints in the lower bound, we need to carefully construct the constraints so that the different actions are properly correlated, which is the key to the hardness of the worst-case instance. We believe this is an important technical contribution that can be further extended to discussing the CMDP complexity lower bound under other settings such as on-policy learning, and so on. Moreover, our analysis to the asynchronous setting is also a technically novel contribution. We believe our techniques  to handle the correlated gradient estimators with large variance can also be beneficial to other algorithms under the asynchronous setting."

---

### Official Review · Reviewer_YGXW · 2022-07-13

**Rating:** 6
**Confidence:** 2
**Soundness:** 3 good
**Presentation:** 2 fair
**Contribution:** 2 fair

**Summary:**

The paper studies the offline reinforcement learning in the framework of constrained Markov decision processes. The authors propose an offline primal-dual algorithm, prove near-optimal sample complexity under different assumptions on datasets, and propose an adaptive algorithm without prior knowledge on concentrability coefficient.

**Questions:**

- Notation $I$ is abused in Section 2.1.

- line 103: what is the sparse nature of optimal policy? LP only works for occupancy measures, not policy.

- Can you explain how to determine $C^*$ in Assumption 2.3?

- Does saddle points always exist for (6) or (7)? How does constraints (8) work for (7)?  This constraint set looks a strong restriction.

- How to compute $\mu(s,a)/\hat \mu(s,a)$ for degenerate $\hat \mu(s,a)$ when finite concentrability is absent? An issue occurs in (9) also.

- How to compute $KL(\lambda \Vert \lambda^t)$ on $\Lambda$?



**Ethics Review Area:**

["I don’t know"]

**Limitations:**

No, for example how to mitigate potential bias in offline data is not discussed.

**Strengths And Weaknesses:**

## Originality

- The proposed offline primal-dual algorithm is new in offline constrained reinforcement learning.

- The sample complexity matches a lower bound except for the dependence on the discount factor.

## Quality & Clarity

- Main results of the paper has been delivered well, except for a few concepts.

- All claims are supported by proofs, although I didn't check correctness.

## Significance

- The proposed offline primal-dual algorithm takes either independent batch dataset or single trajectory sequence, which are two important settings in offline reinforcement learning.

- The optimality of sample complexity is studied by establishing lower bound.

- An adaptive implementation of proposed algorithm is useful for practice due to the absence of prior knowledge on concentrability coefficient.

---

> ### Author Response · Authors · 2022-07-30
> **Response to reviewer YGXW**
>
> We thank the reviewer for her/his time and thoughtful feedback. We address the comments in detail as follows.
>
> $\mathbf{Q1.}$ Notation $I$ is abused in Section 2.1.
>
> Answer: We thank the reviewer for pointing out this issue. In order to distinguish the identity matrix and the number of constraints, in the revision, we will use $\mathbb{I}$ to denote the identity matrix while using $I$ to denote the number of constraints.
>
> $\mathbf{Q2.}$ Line 103: what is the sparse nature of optimal policy? LP only works for occupancy measures, not policy.
>
> Answer: Given any state-action occupancy measure $\nu$, the policy $\pi$ that generates this occupancy measure equals
> $\pi(a|s) = \frac{\nu(s,a)}{\sum_{a'}\nu(s,a')}$, see Eq. (3) of our paper. Therefore, the optimal policy $\pi^*(a|s)>0$ only when the optimal occupancy measure $\nu^{\pi^*}(s,a)>0$, for any $s,a$. That is, $\pi^*$ and $\nu^{\pi^*}$ share the same support. By Proposition 2.1, as long as $|\mathcal{S}|+I\ll |\mathcal{S}||\mathcal{A}|$, the number of nonzero entries of $\pi^*$ will be far less than its dimension, indicating the sparsity of the optimal policy.
>
> $\mathbf{Q3.}$ Can you explain how to determine $C^*$ in Assumption 2.3?
>
> Answer: The definition of concentrability coefficient $C^*$ (Assumption 2.3) involves the optimal policy of the CMDP, and hence it cannot be known a priori in general. One exception is when the offline data distribution completely covers the whole state-action spaces: $\mu(s,a)>0, \forall s,a$. In this case, a pessimistic upper bound for $C^*$ is $\frac{1}{min_{s,a}\mu(s,a)}$, which can be very loose.
>
> In fact, the issue introduced by an unknown $C^*$ appears in many previous works on offline MDP. Suppose that we are given an offline dataset of $N$ samples, then it is guaranteed in [17, 23, 24, 31, 33, 34] that a policy with suboptimality $\mathcal{O}\(\sqrt{MC^*/N})$ can be obtained for some constant $M$. Therefore, as $C^*$ is practically unknown, the suboptimality of the output policy is also unknown for these existing results.
>
> And this is exactly the reason why we propose an adaptive deviation control scheme (Section 6) to avoid the knowledge of the $C^*$.
>
> $\mathbf{Q4.}$ Does saddle points always exist for (6) or (7)? How does constraints (8) work for (7)? This constraint set looks a strong restriction.
>
> Answer: A saddle point of (6) and (7) always exists as long as the original LP problem has an optimal solution, which is indeed the case. In (8) we further restrict the primal domain and dual domain, by considering the natural relations that a saddle point $(V^*,\lambda^*,x^*)$ must satisfy. Therefore, the constraints (8) possibly exclude some suboptimal solutions, but it is guaranteed to contain the optimal solution. Such a reduction of domain size is common in the primal-dual approach of MDP/CMDP, for example [28]. The equivalence between (7) and (8) is implied by Lemma E.3.
>
> $\mathbf{Q5.}$ How to compute $\mu(s,a)/\hat{\mu}(s,a)$ for degenerate $\hat{\mu}(s,a)$ when finite concentrability is absent? An issue occurs in (9) also.
>
> Answer: Note that $\hat{\mu}$ is constructed by Eq. (10) in Algorithm 1, which does not depend on a finite concentrability. Because we truncate $\hat{\mu}$ at some pre-determined $\varsigma>0$, we know $\hat{\mu}(s,a)>\varsigma$ for all $s,a$.  Therefore, $\mu(s,a)/\hat{\mu}(s,a)$ and (9) are both well-defined.
>
> $\mathbf{Q6.}$ How to compute $\mathrm{KL}(\lambda\|\lambda^t)$ on $\Lambda$?
>
> Answer: Because $\Lambda$ is not a subset of the probability simplex, the KL divergence on it is actually the generalized KL divergence: $\mathrm{KL}(Y\|Y'):= \sum_i Y_i\log\frac{Y_i}{Y_i'} -\sum_i Y_i + \sum_i Y_i'$, see Line 168-169 under the Algorithm 1.

---

### Meta-Review · Area_Chair_Nive · 2022-08-28

**Recommendation:** Accept
**Confidence:** Certain

**Metareview:**

This paper considers offline reinforcement learning in the constrained MDP framework. It proposes an algorithm that provably obtains a near-optimal policy (under a single-policy concentrability assumption) and proves an upper bound (and a corresponding lower-bound) on the resulting sample complexity.

The reviewers found the paper well-motivated and technically sound, and unanimously recommend acceptance. Please incorporate the reviewers' feedback in the final version of the paper. In order to strengthen the final paper, it would be helpful to:
- Incorporate toy experiments and empirically validate some of the paper's claims
- Include a discussion about the tightness of the upper/lower bound.



**Award:**

No

---

### Decision · Program_Chairs · 2022-09-14

Accept